# Importance of Rose Bengal Loaded with Nanoparticles for Anti-Cancer Photodynamic Therapy

**DOI:** 10.3390/ph15091093

**Published:** 2022-08-31

**Authors:** Batoul Dhaini, Laurène Wagner, Morgane Moinard, Joël Daouk, Philippe Arnoux, Hervé Schohn, Perrine Schneller, Samir Acherar, Tayssir Hamieh, Céline Frochot

**Affiliations:** 1Reactions and Chemical Engineering Laboratory, Université de Lorraine, LRGP-CNRS, F-54000 Nancy, France; 2Laboratory of Macromolecular Physical Chemistry, Université de Lorraine, LCPM-CNRS, F-54000 Nancy, France; 3Department of Biology, Signals and Systems in Cancer and Neuroscience, Université de Lorraine, CRAN-CNRS, F-54000 Nancy, France; 4Faculty of Science and Engineering, Maastricht University, P.O. Box 616, 6200 MD Maastricht, The Netherlands; 5Laboratory of Materials, Catalysis, Environment and Analytical Methods Laboratory (MCEMA), Faculty of Sciences, Lebanese University, Hadath 6573, Lebanon

**Keywords:** photodynamic therapy, rose bengal, X-rays, nanomedicine, singlet oxygen, cancer, nanoparticle

## Abstract

Rose Bengal (RB) is a photosensitizer (PS) used in anti-cancer and anti-bacterial photodynamic therapy (PDT). The specific excitation of this PS allows the production of singlet oxygen and oxygen reactive species that kill bacteria and tumor cells. In this review, we summarize the history of the use of RB as a PS coupled by chemical or physical means to nanoparticles (NPs). The studies are divided into PDT and PDT excited by X-rays (X-PDT), and subdivided on the basis of NP type. On the basis of the papers examined, it can be noted that RB used as a PS shows remarkable cytotoxicity under the effect of light, and RB loaded onto NPs is an excellent candidate for nanomedical applications in PDT and X-PDT.

## 1. Introduction

Photodynamic therapy (PDT) for cancer presents significant advantages over chemotherapy and radiation therapy [1]. PDT does not cause any of the side effects reported for other types of cancer treatment [2]. PDT was first applied clinically on humans in 1978 [3]. PDT is based on the principle of the excitation of a drug, called a “photosensitizer” (PS), with a light beam [4]. This PS is not toxic in the dark, but induces the production of highly reactive oxygen species (ROS), such as singlet oxygen (^1^O_2_), upon light illumination [5]. Upon light illumination, the PS is activated into a singlet excited state and then into a triplet state following intersystem crossing. In its triplet state, the PS undergoes electron or proton transfer to produce superoxide anion and hydroxyl radical, or transfers its energy into oxygen to produce ^1^O_2_. Around 20 PSs have been commercialized to date, or are currently in clinical trials, including porphyrin, chlorin and phthalocyanine [6,7]. There are still some improvements to be made, such as the achievement of better absorption in the near-IR range, as well as achieving better solubility and increased selectivity. The coupling of these PSs to nanoparticles (NPs) can be used to achieve passive targeting of the PS to cancer cells, thus decreasing health cells being attacked [8]. One of the limitations of PDT is the poor penetration of light into the tissue. To overcome this problem, PDT excited by X-rays (X-PDT) can be used, owing to the fact that X-rays penetrate more deeply into the tissue [9].

Rose Bengal (4,5,6,7-tetrachloro-20,40,50,70-tetraiodofluorescein disodium, RB) is a dye belonging to the fluorescein family [10]. This amphiphilic chemical molecule is already used in medical applications to test the activity of the liver [11] and to diagnose corneal lesions [12]. Subsequently, it had been discovered by researchers that RB stimulates the immune system, and making it possible to reduce the risk of certain cancers [13].

For X-PDT applications, it is necessary to couple RB with NPs containing lanthanides [14]. Since RB absorbs within the visible range, the principle involves the excitation of lanthanides using X-rays; following the luminescence of the lanthanide, energy transfer towards RB takes place. In most cases, this is a non-radiative FRET (Förster resonance energy transfer)-type energy transfer [15]. Following this energy transfer, the RB is able to convert triplet oxygen to produce ^1^O_2_ [16]. Thanks to its high ^1^O_2_ quantum yield (Φ_∆_) and moderate fluorescence, RB can be used in anti-cancer and anti-bacterial PDT.

In this review, we collect all of the papers that describe the use of RB and NPs. These are divided in two parts: NPs excited by light (PDT), and NPs excited by X-rays (X-PDT). Then, the papers are classified on the basis of the type of NPs used. Finally, for each type of NP, the articles are presented in chronological order.

## 2. Photodynamic Therapy

### 2.1. Upconversion Nanoparticles (UCNPs)

Upconversion nanoparticles (UCNPs) are NPs that possess the ability to absorb two or more low-energy photons from infrared (IR) radiation and emit high-energy photons in the ultraviolet (UV) or visible area in between. The size of UCNPs is usually between 10 and 100 nm. A shift from high to low wavelengths is an anti-stock shift [17]. These NPs have attracted the attention of researchers due to several characteristics, including their photostability, low toxicity, no bleaching, high conversion efficiency, high light penetration into biological tissues, and long lifetime (from nanoseconds to milliseconds) [18]. UCNPs are composed of transition metals doped with rare earths (lanthanides or actinides) and incorporated deeply into cells. All of these characteristics allow their application in nanomedicine therapy, imaging and detection [19].

Liu et al. [20] were the first to synthesize and characterize hexagonal UCNPs NaYF_4_:Yb^3+^, Er^3+^(Y:Yb:Er = 78:20:2) with a size of 20 nm. To replace oleylamine ligands, they used 2-aminoethyl dihydrogen phosphate (AEP) to introduce NH_2_ functions (Figure 1). These amino groups of the NPs were coupled to the carboxylic group of the hexanoic acid ester of RB. They evaluated around 100 PS using NPs. Finally, they performed covalent coupling of folic acid (FA) via a dual functional PEG (NH_2_-PEG-COOH).

^1^O_2_ production was determined using the chemical probe 1,3-diphenylisobenzofuran (DPBF). A FRET energy transfer between this NP and RB reached 83% efficiency. ^1^O_2_ production from UCNPs^_^RB nanoconjugates after excitation at wavelengths of 980 nm and 540 nm was observed, with a low difference between them. The authors performed irradiation at a wavelength of 980 nm (1.5 W/cm^2^ for 10 min). The cell viability decreased with increasing UCNPs-PS concentration, showing the effectiveness of these biocompatible NPs (Figure 2, left). After the incubation of different concentrations of UCNPs in choriocarcinoma JAR cells (FR+) and NIH 3T3 cells (FR−) as non-cancer cells and irradiation at a wavelength of 980 nm (1.5 W/cm^2^ for 10 min), cell viability was measured (Figure 2 right). In NIH 3T3 cells, no killing was observed due to the non-incorporation of the UCNPs. In the JAR cells, 50% cell death was measured at 100 µg/mL.

Chen et al. [21] developed the multifunctional nano-platform UCNP@BSA-RB&IR825. The diameter of the NPs was around 60 nm. The UCNPs (NaGdF_4_:Yb:Er (Gd:Yb:Er = 78:20:2)) were coated with polyacrylic acid (PAA) and conjugated using bovine serum albumin (BSA) protein. The hydrophobic pockets of the BSA protein enabled the effective loading of RB (≈ 7.6%, *w*/*w*) and a NIR-absorbing dye (IR825 ≈ 22%, *w*/*w*). The UCNPs showed excellent solubility both in water and physiological solutions. The resulting UCNP@BSA-RB&IR825 had the ability to produce both the PDT effect under excitation at a wavelength of 980 nm and a photothermal therapy (PTT) effect under 808 nm laser irradiation. Moreover, the presence of Gd^3+^ metal ions conferred these UCNPs with T1-weighted MRI (Magnetic Resonance Imaging) properties. ^1^O_2_ production was confirmed using an ^1^O_2_ Sensor Green (SOSG) probe. In vitro and in vivo experiments were performed on murine breast 4T1 cancer cells and 4T1 tumor-bearing mice irradiated with an 808 nm laser (0.5 W/cm^2^ for 5 min) followed by 980 nm laser (0.4 W/cm^2^ for 10 and 30 in for in vitro and in vivo experiments, respectively, with a 1 min interval after 1 min irradiation to avoid heating). UCNP@BSA-RB&IR825 showed no in vitro cytotoxicity at various concentrations (from 0 to 0.4 mg/mL) after 24 h incubation, and 4T1 cell viability was strongly decreased by the synergistic effect of PDT/PTT (Figure 3a). This synergistic effect was also observed in vivo with the inhibition of tumor growth (Figure 3b). The relative tumor volume (*V*/*V*_0_) after 2 weeks was 7.3 for PDT, and the PTT effect was 7.5.

Wang et al. [22] designed lipid nanovesicles combined with UCNPs (NaYF_4_:Yb, Er(Y:Yb:Er = 78:20:2)) functionalized with PEG and RB (namely, RB-UPPLVs). RB-UPPLVs were modified with PEG on their surfaces. The nanosized PLVs presented high stability in blood. Once the RB-UPPLVs reached the tumor sites, they lost their PEG coating and the FA ligand was exposed under the acidic tumor environment. The average size was 62.1 nm. The biological tests were performed on HeLa cells. After treatment at 980 nm (2.5 W/cm^2^, 20 min), cell viability for RB-UPPLVs pretreated with ABS (Acetate Buffer Solution) (22%) was lower than that of RB-UPPLVs incubated with PBS (Phosphate Buffer Solution) (50%), which was not the case in the dark.

In the same year [23], the same team synthesized UCNP (NaYF_4_:Yb/Ho@NaYF_4_:Nd@NaYF_4_ (Yb/Ho(8/1%)@NaYF4:Nd(20%)) core–shell–shell nanostructures based on a thick Nd^3+^ sensitized shell. A ligand exchange approach was used to introduce NH_2_ functions with poly(allylamine). The upconversion luminescence (UCL) of the UCNPs-RB was improved, compensating for the deleterious effect on UCL resulting from the decreased Nd^3+^ sensitized layer. The covalent conjugation of RB was realized in the same way as reported in Liu et al. [20]. In vitro experiments were performed in HeLa cancer cells with different concentrations of UCNPs-RB. After irradiation at 808 nm (0.67 W/cm^2^, 10 min) with 600 µg/mL of UCNPs, only 10% of cells remained alive (Figure 4).

Lu et al. [24] designed organosilica-shelled β-NaLuF_4_:Gd/Yb/Er (Lu/Gd/Yb/Er: 50/30/18/2 in mol%) UCNPsI with a rattle structure. NPs were adsorbed either with zinc β-carboxyphthalocyanine (ZnPc-COOH, 2.7 wt%) or RB (1.9 wt%). The NP core was β-NaLuF_4_:Gd/Yb/Er and the shell was benzene-bridged organosilica. The size of these NPs was 10 nm. The rattle was used to introduce the storage of PSs and promote energy transfer. Moreover, the amino terminal fragment of urokinase plasminogen was finally conjugated in order to specifically target the urokinase plasminogen activator (UPA) receptor expressed in yeast cell. The synthesis route is shown in Figure 5.

In vitro tests were performed on H1299 cells (human lung cancer cells). After irradiation at 980 nm (0.5 W/cm^2^, 10 min), a viability of 10% was observed following incubation for 4 h with NPs (0.05 mg/mL).

Li et al. [25] developed UCNPs of luminescent lanthanide for Er doping with 20% Nd^3+^ and 10% Yb^3+^. They coupled the hexanoic acid spacer to RB. The well-known EDC (*N*-ethyl-*N*′-(3-(dimethylamino)propyl)carbodiimide)/NHS (*N*-hydroxysuccinimide) couple was used for coupling the RB-HA to polyethylene imine PEI, and then to Er@NdBF_4_^−^ (Figure 6).

^1^O_2_ production occurred as a result of the energy transfer from Nd^3+^ to Yb^3+^ to Er^3+^ to RB. The UCNPs had a size of 25.7 ± 1.4 nm, with a shell thickness of about 6 nm. In vitro tests on 4T1 cells showed the production of ^1^O_2_ by means of DPBF. The amount of ^1^O_2_ produced increased with increasing irradiation time (maximum 8 min), laser density (maximum 0.8 W/cm^2^), and concentration (maximum 200 µg/mL) (Figure 7). Moreover, Er@Nd-RB conjugates did not enter the nucleus, but remained localized in the cytoplasm, inducing cellular oxidative stress and/or destroying proteins and breaking down organelle membranes.

The in vivo tests were performed on 4T1 breast cancer in BALB/c nude mice. Following 808 nm laser irradiation (520 mW/cm^2^), Er@Nd-RB conjugates rapidly set upon the tumor region and remained at the tumor site for at least 45 min, while no signal was detected in other parts of the body. Mice treated with Er@Nd-RB NPs (100 µL, 10 mg/mL) showed a greater decrease in tumor volume compared to the control with other NPs (Figure 8a).

The liver and spleen weights of the Er@Nd-RB-treated group 4 did not significantly change after PDT, which was not the case for the PBS group, demonstrating that PDT clearly attenuated the tumor-induced increase in volume in the liver and spleen.

Liang et al. [26] designed UCNPs based on a silica layer doped with RB. The particles were a core–shell (NaYF_4_:Yb, Er@NaGdF_4_ (Yb:Er 18%:2%)) doped with RB. A modified water-in-oil microemulsion method was employed to synthesize UCNP@SiO_2_(RB). They were then functionalized with a fusion protein consisting of a binding peptide linker (L) genetically fused to *Streptococcus* Protein G, which in turn bound antibodies that were directed against cancer cells (Figure 9).

The average final size was around 43 nm. In particular, they tested antibodies against epCAM (FITC (Fluorescein isothiocyanate) labeled), which is overexpressed on the cell surface of many cancers, particularly colon adenocarcinoma cells. Antibodies bound epCAM and led to immune reaction, triggering cell death. In vitro experiments were performed on HT-29 cells with various concentrations of UCNP@SiO_2_(RB)-LPG-Ab and irradiation times using 980 nm laser irradiation (1.5 W/cm^2^). After 24 h incubation with UCNP@SiO_2_(RB)-LPG-Ab, cell viability decreased to 38% (200 µg/mL, 15 min).

Ai et al. [27] synthesized biocompatible core–shell–shell UCNPs (NaGdF_4_:Yb/Nd@NaGdF_4_:Yb/Er@NaGdF_4_ (50:50)). The UCNPs were covalently loaded with RB and Pt(IV) prodrug, in the form of *c*,*c*,*t*-[Pt(NH_3_)_2_Cl_2_(OCOCH_2_CH_2_NH_2_)_2_] (Cisplatin), and PEGylated for biodisponibility. In vitro experiments were performed on the A2780 human ovarian carcinoma cell line and A2780cisR cells (A2780 cells resistant to cisplatin), at concentrations of 16, 80 and 400 µg/mL under near-infrared (NIR) laser irradiation (808 nm, 6 W/cm^2^, 5 min). A2780 cell viability decreased to 70%, 10%, and 2%, respectively. Cell viability decreased with increasing concentration of A2780cisR (to 70%, 30%, and 5% at concentrations of 16, 80 and 400 µg/mL, respectively) (Figure 10).

Chen et al. [28] synthesized NIR-activated UCNPs to target neuroendocrine tumors (NETs) such as medullary thyroid cancer and carcinoids, which frequently hydrophobic to the liver, leading to a poor survival outcome. The size of these NPs was 29 nm, and they were composed of (NaYF_4_:Yb^3+^/Er^3+^/Tm^3+^(Y:Yb:Tm:Er(78:20:2:0.2)). On the UCNP surface, a photosensitive amphiphilic copolymer poly(4,5-dimethoxy-2-nitrobenzyl methacrylate)-polyethylene glycol (PNBMA-PEG) was covalently coupled with RB. Upon light illumination, there was a photoinduced polymer side-group cleavage leading to the release of the encapsulated drug AB3, which was loaded into the micelle. The mechanism is described in Figure 11.

The targeting agent KE108 was coupled at the PEG extremity in order to target somatostatin receptors (SSTRs), which are overexpressed in many neuroendocrine cancers. The ^1^O_2_ production under NIR irradiation was confirmed by the decrease in DPBF fluorescence. In vitro experiments were performed on NET (TT) cells incubated with T-RB-A3 (166 µg/mL) for 3 h. After NIR irradiation (980 nm, 0.5 W/cm^2^, 10 min), these cells were re-incubated for 45 h. Cell viability decreased by 90%, compared to a decrease of 40% without illumination. This decrease in viability became less important when treating cells without the targeting agent, without AB3, or without RB. In vivo experiments were performed on TT-tumor xenograft mouse model grafted onto male athymic nude mice. Irradiation was performed at 980 nm (0.5 W/cm^2^, 15 min, interval of 1 min after each 5 min of irradiation). At 7 days post injection of treatment, T-RB-AB3 showed a greater decrease in tumor volume compared to mice treated with T-RB and T-AB3 alone. These results indicated that the combination of chemotherapy with PDT was more effective than chemotherapy or PDT alone (Figure 12).

Chang et al. [29] designed core–shell (i.e., Yb^3+^, Er^3+^@ NaYF_4_ (20:2)) UCNPs coated with poly(allylamine) and co-loaded with RB and zinc phthalocyanine (ZnPc). The size of these NPs was between 24 nm and 32 nm. RB and ZnPc were coupled covalently to the surface of the NPs in the shell, allowing an efficient resonance energy transfer of around 50%. The production of ^1^O_2_, estimated on the basis of DPBF, was optimal for the co-loaded NPs (90%). The cellular uptake of the UCNPs was evaluated on A549 lung carcinoma cells, revealing an accumulation in the mitochondria after internal transport by lysosomes. After incubation for 24 h, the cell survival rate was 20% after irradiation (980 nm, 0.25 W/cm^2^, 15 min). A mitochondrial apoptosis pathway was demonstrated. Hepa1-6 tumor-bearing mice after PDT treatment (980 nm, 0.25 W/cm^2^, 15 min) showed a relative tumor inhibitory ratio of 90%.

Hou et al. [30] synthesized UCNPs (NaYF_4_:Yb, Tm)@0.6(NaYF_4_:Yb, Er (Y:Yb:Tm = 75/25/0.3)) coated with tetraethyl orthosilicate (TEOS) (UCN@mSiO_2_) coupled with azobenzene (Azo) and RB, and encapsulated with DOX. The size of these NPs was 120 nm. ^1^O_2_ production was confirmed by ABDA (9,10-anthracenediyl-bis(methylene) dimalonic acid) probe. In vitro MTT tests were performed on HeLa cells at different concentrations. Cell viability decreased significantly when the cells were treated with UCN@mSiO_2_-(Azo + RB) compared to when treated with UCN@mSiO_2_-AZo and UCN@mSiO_2_. When excited at 980 nm (2.0 W/cm^2^, 10 min), HeLa cells treated with 100 µg/mL showed a decrease in cell viability of 60% with DoxUCN@mSiO_2_-(Azo + RB) and of 40% with UCN@mSiO_2_-(Azo + RB). The NIR excitation of these NPs activated the Azo molecules, allowing the release of the drugs and the PS (DOX and RB).

Kumar et al. [31] synthesized white emitting NaYF_4_ upconversion nanocrystals (Y/Yb/Er:78/20/2). The UCNPs were composed of a hydrophobic core of oleic-acid-coated NPs and NaYF_4_:Yb/Er/Tm (OA-WENs) coated with a layer of dense silica, followed by another layer of mesoporous silica, and with primary amine groups introduced on the external surface (WE-SiO_2_). FA and RB were then alternatively covalently conjugated. The second PS (ZnPc) was loaded inside the mesoporous silica shell by physisorption. Finally, dPS-WE-SiO_2_ NPs were obtained, with a hydrodynamic size of 91 nm. The FRET effect between the WEN core and the PSs was also confirmed on the basis of fluorescence and lifetime studies. DPBF was used to detect ^1^O_2_ production. In vitro studies were conducted on HeLa cells. dPS-WE-SiO_2_ NPs were found to be localized within the cytoplasm of the HeLa cells 48 h post incubation, and most of them were internalized. In the case of cells treated with 200 μg/mL of dPS-WE-SiO_2_ NPs and irradiated with am NIR laser (980 nm, 2.5 W/cm^2^, 20 min), number of dead cells was greater than in the cells treated with the single-PS-based nano-system. Early apoptosis was observed in 43% of dPS-WE-SiO_2_-NP-treated cells, compared to 22.8% and 14.2% for RB-WE-SiO_2_-NP- and ZnPc-WE-SiO_2_-NP-treated cells, respectively. Late apoptosis and necrosis were observed in 51.5% of dPS-WE-SiO_2_-NP-treated cells, compared to 16.7% and 13.3% of RB-WE-SiO_2_-NP- and ZnPc-WE-SiO_2_-NP-treated cells, respectively.

Sabri et al. [32] synthesized NIR-excited UCNPs. Lanthanide-doped UCNPs NaGdF_4_:Yb^3+^/Er^3+^ (Gd/Er/Yb:78/2/20)(Ln-UCNPs) were coated with BSA to increase biocompatibility and stability in aqueous medium, resulting in BSA-Ln-UCNPs being obtained. RB was covalently conjugated onto the UCNPs with hexanoic acid spacer. RB-BSA-Ln-UCNPs were obtained, with a size of 78 nm. First, the energy transfer from the BSA-Ln-UCNPs to the RB was confirmed by the upconversion emission spectra, with a maximum energy transfer efficiency of 68% at a concentration of RB of 20 μM. DPBF was used to confirm the production of ^1^O_2_. In vitro studies were conducted in mammalian cancer cell line A549. MTT (3-[4,5-dimethylthiazol-2-yl]-2,5 diphenyl tetrazolium bromide) assays were conducted to evaluate the effect of various concentrations of RB-BSA-Ln-UCNPs on the viability of A549 cells after a 980 nm irradiation (13 W/cm^2^, 10 min). Dark cytotoxicity was observed, with cell viability decreasing by 15%. Cytotoxicity was observed upon irradiation, with a stronger effect at higher concentrations. At an RB-BSA-Ln-UCNP concentration of 250 μg/mL, 36% cell death was obtained after 10 min of exposure. These experiments confirmed the efficacy of these NPs when coupled to RB in PDT.

Buchner et al. [33] synthesized RB–lysine-functionalized UCNPs (UCNPs@K(RB)). RB was covalently attached to the lysine-modified UCNPs. The core–shell UCNPs were NaYF_4_:Yb,Er,Gd@NaYF_4_ (Yb/Gd/Er:20/20/2). The final solvodynamic diameter of these UCNPs was about 23.7 ± 8.4 nm, with a loading of RB estimated at 160 RB per NP. The ^1^O_2_ production of UCNPs@K(RB) was evaluated using ABDA as a probe. An in vitro study was conducted on SK-BR-3 breast cancer cells at different concentrations of UCNPs@K(RB). After 3 h incubation with UCNPs@K(RB) (20 µg/mL), PDT treatment (980 nm, 200 mW/cm^2^, 6 min) decreased the cell viability by more than 85% compared to the non-irradiated cells.

Xie et al. [34] developed a complex O_2_-loaded pH-responsive multifunctional nanodrug carrier, UC@mSiO_2_-RB@ZIF-O_2_-DOX-PEGFA (NaYF_4_:Yb/Er@NaYbF_4_:Nd@NaGdF_4_ (Yb/Er/Gd:18/2/80)), coating it onto silica in which RB was encapsulated. They added Zeolitic imidazolate framework-90 (ZIF-90) as an MOF (metal–organic framework) to act as an O_2_ reservoir, in order to release O_2_ into the tumor microenvironment to fight against hypoxia. NPs were conjugated to the chemotherapy drug DOX and PEG-FA (abbreviated URODF) to target the FA receptor (Figure 13).

These NPs emitted a green light after having absorbed photons following illumination at a wavelength of 808 nm, as a result of a multi-photon process that activated the RB. The size of the URODF was 150 nm. ^1^O_2_ production was quantified using DPBF. In vitro studies were performed on 4T1 and HeLa human cells. Eight groups were used in the study, with or without laser irradiation (808 nm, 0.5 W/cm^2^, 10 min), as follows: (1) UC@mSiO_2_@ZIF-PEG-FA (UF), (2) RB, (3) laser irradiation (+), (4) DOX, (5) UC@mSiO_2_@ZIF-DOX/PEG-FA (UDF), (6) UC@mSiO_2_-RB@ZIF-O_2_-PEG-FA with laser irradiation, (7) (UROF+) UC@mSiO_2_-RB@ZIF-DOX/PEGFA with laser irradiation (URDF+), (8) URODF with laser irradiation (URODF+). After incubation of 4T1 and HeLa cells for 14 days with URODF (100 µg/mL), and laser irradiation, the cell viability decreased from 100% to 20% (Figure 14a,b). These UCNPs were injected in vivo into H22 (murine hepatocarcinoma) tumor-bearing BABc mice with at a concentration of 4.0 mg/kg in DOX. The fluence was 0.5 W/cm^2^ for 0 to 150 s. After 15 days of treatment with URODF, tumor volume decreased significantly (50%) (Figure 14c,d).

CSGUR-MSGG/5FU NPs were designed by Kumar et al. [35] to achieve dual chemotherapy and PDT, as well as enzyme-responsive drug release for treatment of colorectal cancer. The NaYF_4_:Yb/Er (Yb/Er:20/2) core was shelled with NaYbF_4_ and loaded with RB to produce UCNPs. Then, mesoporous silica loaded with FA for colorectal cancer targeting and 5-fluorouracil (5FU) drug was shelled. Finally, a gatekeeper layer for loading drug made with a guar gum (GG) polymer was added. Glycosidase enzymes in the colon can degrade GG to produce enzyme-responsive drug release in the colon area. TEM (transmission emission microscopy) images of the NPs made it possible to measure their size (72.3 ± 7.1 nm). Under NIR irradiation (980 nm, 0.7 W/cm^2^, 10 min), an energy transfer efficiency of 94.1% from UCNPs to RB was observed with the quenching of UCNP emission at 540 nm. Drug release was evaluated under different pH conditions, but no 5FU release was observed, confirming the stability of the NPs. When NPs were incubated with colonic enzymes, 5FU was released, with 82% being released after incubation for 72 h. Under NIR irradiation without colonic enzymes, 10% 5FU release was observed. A DPBF probe was used to confirm ^1^O_2_ production under NIR irradiation. HT-29 cells were incubated with CSGUR-MSGG/5FU NPs for 48 h (200 µg/mL). A DCFH-DA (dichlorodihydrofluorescein-diacetate) probe made it possible to visualize intracellular ROS production. Under chemotherapy conditions, low green fluorescence was observed, showing a moderate production of ROS even without NIR irradiation. In PDT conditions, ROS was intensively generated, and the highest ROS production rate was observed in dual therapy conditions, demonstrating the synergistic effect of the combined treatment. In the dark, no cytotoxicity was observed. In the presence of colonic enzymes (chemotherapy conditions), CSGUR-MSGG/5FU NPs induced cell death in 22.5% of cells, and in 34.3% under NIR irradiation (PDT conditions), whereas combination therapy (enzymes + NIR) induced cell death in 74.8% of cells, confirming the synergistic effect of the dual treatment.

Sun et al. [36] synthesized Barium-Titanate NPs (BT) with RB (BT-RB NPs). The BT-RB NPs were capsulated with PAH (poly(allylamine hydrochloride). An NP size of 70 nm was determined using TEM. All in vitro experiments were performed on HeLa cells. First, they studied the cellular uptake of the NPs by comparing four different groups: RB alone, BT NPs, mixture of BT NPs with RB, and BT-RB NPs. Both SHG (second harmonic generation) and fluorescence signals were observed in the case of BT-RB NPs, confirming its cellular uptake. Intracellular ROS detection was performed using ABDA. It was shown that BT-RB NPs produced more ROS compared to RB alone, BT alone, or a mixture of RB + BT NPs. They conducted an in vitro cytotoxicity assay and demonstrated that BT-RB NPs did not induce dark cytotoxicity.

Song et al. [37] developed Nd^3+^/Yb^3+^/Er^3+^-doped UCNPs co-encapsulated with Ag_2_Se quantum dots (QDs) and RB into phosphatidylcholine (UCQRs). Ag_2_Se QDs enhanced the photoluminescence of the upconversion system, which was excited at 808 nm and was able to perform resonance energy transfer (RET) to Yb^3+^ to improve UCL. The diameter of the UCQRs was 62.57 nm. The UCLof the UCQRs was quenched by 75% with respect to NPs-QDs without RB, demonstrating the energy transfer between the NPs and the RB. The ^1^O_2_ generation capacity of the UCQRs was determined using an ABDA probe and a DCFH-DA probe in vitro. An in vitro viability assay was carried out using HeLa and HCT1116 cells incubated for 24 h with UCQRs (400 mg/mL) with or without light irradiation at a wavelength of 808 nm (1.6 W/cm^2^, 10 min). Without irradiation, a viability of 85% was observed, compared to 10% with irradiation, at a concentration of 400 μg/L. In vivo PDT efficacy was determined using two mouse models: Lewis lung carcinoma tumor-bearing mice and 4T1 tumor-bearing mice. For the first model, intratumoral injection was performed, while intravenous injection was performed for the second mouse model. There were five groups: injected with PBS and irradiated at a wavelength of 808 nm (1.6 W/cm^2^, 10 min), UCQRs, UCRs with irradiation, and UCQRs (10 mg/mL) with irradiation. Tumor growth was greatly inhibited in mice treated with UCQRs with irradiation, whereas a moderate therapeutic effect could be observed for the fourth group, injected with NPs without QDs (Figure 15a). Body weight and tumor growth were monitored for 16 days (Figure 15b).

Wawrzyńczyk et al. [38] developed smart and multifunctional nanocarriers (NCs) for theranostic applications. Co-encapsulation of Span 80 micelles containing RB and NaYF_4_:Er^3+^,Yb^3+^(Er/Yb:2/20) UCNPs in NCs was performed using a double core composed of a non-ionic surfactant Rosalfan A and poly(lactide-co-glycolide) (PLGA). A coron of cationic chitosan and PEG-HA was added to provide stealth and targeting properties. A hydrodynamic size of 154 ± 5 nm was measured for the (PEG-HA-NCs)-NaYF_4_:Er^3+^,Yb^3+^ + RB of. TEM images clearly showed NCs with a diameter of 150 nm loaded with UCNPs with an average diameter of 8 ± 3 nm. The determination of ^1^O_2_ generation was performed using ABMDMA (9,10-Anthracenediyl-bis(methylene)dimalonic acid). No in vitro or in vivo tests were performed.

Wang et al. [39] designed FA/UCNPs-RB/HCPT/PFH@Lipid (named FURH-PFH-NPs) for multimodal imaging and combination therapy. These NPs were composed of a lipid shell modified with FA to target ovarian cancer cells, loaded with 10-hydroxycamptothecin (HCPT) with 72.6% encapsulation efficiency (EE) of HCPT in FURH-PFH-NPs, a cytotoxic drug for chemotherapy and UCNPs-RB for fluorescence imaging and PDT. RB was covalently coupled to UCNP via the classical EDC/NHS method to allow efficient FRET. In the shell, the core was composed of perfluorohexane (PFH), which is able to catch and release oxygen in order to prevent hypoxia in solid tumors (Figure 16).

The size of the NPs was measured on the basis of TEM images, and was found to be 223.6 ± 68.86 nm. ^1^O_2_ formation was detected with DPBF. The low-intensity focused ultrasound (LIFU) technique was used to trigger the release of UCNPs-RB, oxygen, and HCPT. In fact, the PFH liquid enters liquid–gas transition state under irradiation, increasing the size of the particles and inducing ruptures in the NPs (2 W/cm^2^, 2 min). Laser irradiation at 980 nm for 2 min was used at 2 W/cm^2^ for DPBF, and 0.2 W/cm^2^ for other tests. In vitro assays were performed using SKOV3 cells with FURH-PFH-NPs, demonstrating killing of 69.02 ± 1.42% of the total number of cells. SKOV3 cell apoptosis was observed in the presence of FURH-PFH-NPs (100 µg/mL) irradiated with LIFU or laser (approximatively 25%). Irradiation with LIFU + laser increased cell apoptosis to 42.8%. The PDT effect of FURH-PFH-NPs in SKOV3 cells was then evaluated. DCFH-DA was used as a green fluorescence indicator of ROS production. Fluorescence was observed for cells irradiated with LIFU or laser. The fluorescence was even more intense under LIFU + laser, showing the synergistic effect of oxygen and HCPT release by LIFU and PDT using RB. In vivo assays were performed on SKOV3 tumor-bearing mice. To determine the synergistic effect of HCPT (cytotoxic drug loaded in NPs and PDT), four groups of mice were employed: (1) saline + LIFU + laser, (2) HCPT + LIFU + laser, (3) FURH-PFH-NPs + laser, and (4) FURH-PFH-NPs + LIFU + laser. LIFU and laser treatments were performed every day for 28 days. No tumor growth inhibition was observed in groups 1 and 2. Tumor growth inhibition rate of 77.31% was observed for group 3, while an inhibition rate of 92.35% was observed for group 4, demonstrating a significant synergistic antitumoral effect (Figure 17).

Li et al. [40] designed multi-shelled UCNPs (MUCNPs) for activable PDT. NaYF_4_:Gd@NaYF_4_:Er,Yb@NaYF_4_:Nd,Yb (Y/Yb/Er:78/20/2) covalently modified with antenna 800 CW for NIR irradiation luminescence enhancement, RB for ROS generation, cyanine 3 (Cy3) for fluorescence imaging and pep-QSY7 (peptide labeled with a QSY7 quencher) for ROS generation quenching. Cathespin B (CaB), a lysosomal protease overexpressed in many cancers, could hydrolyze the peptide, releasing QSY7 quencher, which could activate RB and Cy3 for PDT and fluorescence imaging. MUCNPs/800WC/RB/Cy3/pep-QSY7 had a hydrodynamic diameter of 56.1 ± 1.0 nm. Energy transfer from MUCNPs to Cy3 was demonstrated by the suppression of the MUCNPs peak at 540 nm. The luminescence lifetime of the MUCNPs decreased from 118 µs without Cy3 to 77 µs with Cy3, confirming RET. ^1^O_2_ generation was detected with DPBF (808 nm, 1.5 W/cm^2^). No ROS was produced by MUCNPs/800WC/RB/Cy3/pep-QSY7, demonstrating the efficient quenching ability of RB and Cy3. In the presence of CaB, ROS generation was almost the same as for MUCNPs/800WC/RB/Cy3, demonstrating the catalytic cleavage of pep by CaB and the activatable PDT. A cytotoxicity assay was performed with HeLa and MCF-7 cells overexpressing CaB and NIH 3T3 as a negative control (6 h incubation, 808 nm, 1.5 W/cm^2^ for 40 min). After 48 h, cell viabilities of 21.6, 26.0 and 91% were observed for HeLa, MCF-7, and NIH 3T3, respectively, demonstrating the therapeutic effect of MUCNPs/800WC/RB/Cy3/pep-QSY7. In vivo experiments were conducted using HeLa tumor-bearing BALB/c nude mice. Four groups were employed: groups 1 and 2 were used as a control with saline injection, and groups 3 and 4 were injected with MUCNPs/800WC/RB/Cy3/pep-QSY7 (300 µg/mL). Groups 2 and 4 were irradiated six times at a wavelength of 808 nm at 1.0 W/cm^2^ for 40 min during the 14 days of the experiment, and no body weight fluctuation was observed (Figure 18b). Tumor growth inhibition was clearly observed for group 4 only, demonstrating the suitability of MUCNPs/800WC/RB/Cy3/pep-QSY7 for activatable PDT (Figure 18a,c).

Han et al. [41] designed core–shell–shell NPs loaded with RB and methylene blue (MB) and DOX for dual pH-responsive chemotherapy and PDT. The UCNP core NaYF_4_:Yb/Ho/Nd@NaYF_4_:Nd was first shelled with silica containing MB, and second with boronic acid RB and DOX. Finally, cyclodextrin (CD) was conjugated onto the NP surface to obtain final NPs called DOX-RB-NH_2_-UCMS-BA-CD. In the acidic environment of cancerous cells, the CD could be dissociated from the NPs, inducing the release of DOX for chemotherapy. The hydrodynamic diameter was 130 nm. ^1^O_2_ production of NPs was observed using an ABDA probe, and NPs were irradiated at a wavelength of 808 nm, 0.5 W/cm^2^, for 5 min. The NPs without MB and the NPs without RB showed no generation of ^1^O_2_, but RB-MB-NPs showed the best ^1^O_2_ production efficacy, highlighting a synergistic effect of the two PSs. The pH-responsive release of DOX was evaluated under pH values of 7.4, 5 and 3. After 24 h incubation of DOX-RB-NH_2_-UCMS-BA-CD at different concentrations (0, 30, 60, 120 and 240 μg/mL), no significant DOX release was observed at physiological pH, whereas 61% release was observed at pH 5, confirming pH-responsive drug release. ROS generation was observed in HeLa cells incubated with NPs (100 mg/mL) under irradiation (808 nm, 0.5 W/cm^2^, 10 min) thanks to DCFH-DA probe. A cytotoxicity assay was performed on HeLa cells incubated with RB-NH_2_-UCMS-BA-CD (0.5 mg/mL). In the dark, no cell death was observed, confirming the biocompatibility of the NPs. Cell viability decreased to 64% and 55% when incubated with DOX-RB-NH_2_-UCMS-BA-CD and RB-NH_2_-UCMS-BA-CD, respectively, and irradiated at a wavelength of 808 nm, confirming their chemotherapeutic and PDT efficiency. Cell viability decreased by 95% with DOX-RB-NH_2_-UCMS-BA-CD under light irradiation, demonstrating the significant synergistic effect of dual therapy. The efficiency of dual therapy was evaluated in vivo: six groups of HeLa tumor-bearing mice were employed: (1) control, (2) 808 nm irradiation for 5 min at 0.5 W/cm^2^, (3) DOX-RB-NH_2_-UCMS-BA-CD, (4) chemotherapy, (5) PDT, (6) chemotherapy + PDT. PDT was more efficient than chemotherapy, as tumor growth inhibition was higher for PDT. Moreover, the dual chemotherapy and PDT group showed the best tumor growth inhibition, owing to the synergistic effect of the two treatments. No body weight loss was observed in the six groups, confirming the biocompatibility of the NPs and the non-toxic treatment (Figure 19a,b).

Jin et al. [42] designed SPTP@UCNP-RB-DOX NPs for dual chemotherapy and PDT. The β-NaYF_4_:Yb/Er (Y/Yb/Er: 20/78/2) UCNP core was shelled with SPTP and loaded with RB and DOX. SA-PEG-TK-PLGA (SPTP) was composed of PLGA, a thioketal group, and sialic acid, targeting E-selectin, which is overexpressed on the surface of tumor cells, linked to the thioketal group by a PEG linker. Under NIR irradiation, UCNPs were able to activate RB via RET and produce ^1^O_2_. A part of the ROS production was used to cut the thioketal group in two in order to induce self-destruction of the SPTP micelles and to release DOX into the tumor environment (Figure 20).

The hydrodynamic diameter of these NPs was 54.3 nm. DPBF probe was used to visualize the NIR (980 nm, 0.5 W/cm^2^, 5–10 min)-controlled ROS generation of SPTP@UCNP-RB-DOX, confirming the RET ability between UCNPs and RB. Regarding chemotherapy, a 20% release of DOX after 14 h was observed without irradiation, whereas DOX release of 70% was observed under NIR irradiation, confirming the good self-destruction and drug release of the NPs. Incubation of 4T1 cells with SPTP@UCNP-RB-DOX and PTP@UCNP-RB/DOX revealed better cell uptake for the NPs containing sialic acid, demonstrating the ability of these NPs to target E-selectin owing to the presence of sialic acid. SPTP@UCNP-RB and STPT@UCNP-DOX (250 µg/mL) demonstrated no cytotoxic activity without irradiation, whereas cell death of 47.2% and 24.5%, respectively, was observed under 980 nm irradiation (0.5 W/cm^2^, 6 min). For SPTP@UCNP-RB-DOX, cell death of 19.6% was observed in the dark due to uncontrolled drug release, whereas cell death of 73.1% was observed under NIR irradiation. Thus, the combination of chemotherapy and PDT was more efficient than chemotherapy or PDT alone. A higher accumulation of NPs in the tumor was observed for SPTP@UCNP-RB-DOX, which had a better tumor targeting ability. Chemo-PDT efficiency was evaluated using 4T1 tumor-bearing TNBC mice. Eight groups of mice were employed: (1) saline, (2) NIR (980 nm, 0.5 W/cm^2^ for 10 min), (3) DOX, (4) SPTP@UCNP-RB, (5) SPTP@UCNP-RB + NIR, (6) SPTP@UCNP-RB-DOX, (7) PTP@UCNP-RB-DOX + NIR, and (8) SPTP@UCNP-RB-DOX + NIR. Tumor growth was observed in groups 1, 2 and 4. Groups 3, 5 and 7 showed moderate tumor growth inhibition, and group 8 showed high tumor growth inhibition, demonstrating the high efficiency of PDT and chemotherapy, especially for dual application of chemotherapy and PDT. The survival rate of group 8 was the highest, confirming the efficiency of the treatment. (Figure 21).

Li et al. [43] also developed UCNPs based on Nd^3+^ metal ions consisting of electrostatically self-assembled core–satellite structures with single MOF NPs as the core and UCNPs (i.e., NaGdF_4_:Yb,Er@NaGdF_4_:Nd,Yb; Gd:Yb:Nd = 5:1:4) as the satellite. The high porosity of MOF NPs allowed a high co-loading (i.e., loading efficiencies in weight ≈ 11%) of chlorin e6 (Ce6) and RB, producing a CR@MU nanotheranostic platform. The biocompatibility and biostability were improved by PEGylation of CR@MU (i.e., CR@MUP, hydrodynamic size of 216.0 ± 7.3 nm). In vitro and in vivo experiments were carried out on 4T1 cells and 4T1 tumor-bearing mice irradiated by the 808 nm laser (1 W/cm^2^ and 1.25 W/cm^2^ for in vitro and in vivo, respectively, for 5 min). These experiments highlighted the ability of the non-cytotoxic CR@MUP to exert a synergistic PDT effect under excitation at a wavelength of 808 nm (i.e., with a spectral overlap between UCNP emission and Ce6 and RB absorption). The cell viability at a concentration of 200 µg/mL under 808 nm excitation was very low (on the order of 10%). In addition, this was used as a trimodal imaging platform (i.e., UCNPs as MR and UCL imaging agents, and Ce6 as a fluorescence imaging agent).

Borodziuk et al. [44] synthesized UCNPs (NaYF_4_:Er,Yb (Er/Yb: 2/20)) with a diameter of 34.2 ± 3.1 nm. These UCNPs were coated with SiO_2_, increasing the diameter of the UCNPs to 41.1 ± 4.7 nm. The UCNPs@SiO_2_ NPs were covalently coupled to RB. A FRET between the NPs and RB was confirmed. The test in solution with DPBF confirmed the production of ^1^O_2_. A Presto Blue viability assay was applied in vitro on 4T1 cells. After 1 and 24 h of incubation with UCNPs@SiO_2_-RB NPs (250 µg/mL) in 4T1 cells, and following irradiation using 2 W/cm^2^ for 10 min at a wavelength of 980 nm, cell viability decreased by 40% compared to the cells irradiated without incubation with NPs, indicating a decrease in ROS production by 60%.

Tezuka et al. [45] successfully synthesized an ultra-small biodegradable spherical UCNP (UC), NaYF_4_:Yb^3+^,Er^3+^ (Yb/Er:18/2 in mol), and a hydrocarbonized RB dye (C_18_RB). The UCNP and C_18_RB were encapsulated in the hydrophobic core of the PEG-block-poly(ε-caprolactone) (PCL) micelle. The hydrodynamic diameter was 9.0 ± 3.2 nm. ^1^O_2_ production was confirmed by DPBF probe. In vitro tests on cancer cells following 980 nm irradiation (2 W/cm^2^ for 30 min) showed an increase in cytotoxicity with increasing RB concentration. Moreover, it is quite remarkable that RB alone was much more toxic than RB and C_18_RB in HNPs (i.e., PEG-C_18_RB-UCNPs). Then, the cell viability was compared between HNPs and the controls; HNPs under irradiation were the most toxic to cancer cells. Encapsulation of C_18_RB with PEG-b-PCL in the hybrid NPs decreased the cytotoxicity of the dye without irradiation. Thus, NIR irradiation showed a PDT effect via the generation of ^1^O_2_, decreasing the viability of deep cancer cells.

Table 1 summarizes the physical–chemical properties and PDT conditions for all references dealing with UCNP@RB.

### 2.2. Silica Nanoparticles

Under standard biological conditions, suspensions of silica NPs are stable if the salt concentration is low, but precipitate when this concentration increases beyond a critical threshold. Silica NPs are therefore all the more stable when the environment contains few salts and proteins, when they are few in number and larger in diameter, and when their surface is grafted by adequate groups. The mechanism of endocytosis, although it may differ from one cell line to another, always seems to be an active process, and the internalization of silica NPs is mainly done by clathrin or caveolae [46]. In addition, silica has the ability to combine imaging and contrast agents [47]. This has been used, for example, to incorporate or chelate rare earths (lanthanides) in these NPs [48].

Uppal et al. [49] synthesized silica NPs (SiNP) covalently or electrostatically coupled to RB (i.e., RB-SiNP electrostatic or covalent complexes, respectively). SiNP and the RB-SiNP electrostatic complex had similarly sized (35 ± 5 nm), while the size of the RB-SiNP covalent complex increased to 50 ± 5 nm. ^1^O_2_ production was confirmed by SOSG probe. Cytotoxicity tests were performed on two different MCF-7 and 4451 cells at three concentrations of RB (10, 15 and 24 µM). With increasing concentration, cytotoxicity increased. Dark cytotoxicity was very low. The cytotoxicity on MCF cells at a wavelength of 480 nm for 1 h reached 87% for the RB-SiNP covalent complex NPs at a concentration of RB = 24 µM, and 94% under the same conditions in 4451 cells. The RB-SiNP electrostatic complex with a concentration of RB = 24 µM showed a cytotoxicity of 55% and 40%, respectively. The RB-SiNP covalent complex was more cytotoxic than the electrostatic complex.

Gianotti et al. [50] described the elaboration of mesoporous silica NPs (MSNs) with a size of 160–180 nm covalently coupled to RB through an amide bond. The average amount of RB per NP was about 4.20 × 10^4^. They evaluated Φ_∆_ using uric acid, and found a value of 0.74 in water. In vitro experiments were performed on melanoma cellular model (SK-MEL-28) with green light (green light, 5 min). Cell proliferation after 5 h incubation was 64.10 ± 8.79 cells/mm^2^ in tumor-bearing mice in the absence of light and 47.85 ± 13.48 in presence of light. This diminution in cell proliferation confirmed the importance of PDT in melanoma treatment.

Martins Estevão et al. [51] incorporated RB into MSNs (with an average size of 150–180 nm) to optimize the efficiency of ^1^O_2_ generation when illuminated with 540 nm green light (450 W Xenon lamp, 80 min). MSNs were functionalized using cetyltrimethylammoniumbromide (CTAB). No biological experiments were performed.

Adem et al. [52] proposed MSNs based on CTAB and RB (rMSN-ts). The length of rMSN-ts was 119 nm, and they had a width of 86 nm. In vitro experiments were performed in MCF-7 cells, and after incubation for 4 h with rMSN-ts (100 µg/mL), cell viability decreased from 100% to 25% with the application of green laser light irradiation (5 mV) for 30 min.

Liu et al. [53] designed polyglycerol MSNs with adsorbed FITC and RB (namely, FITC-PGSN-RB; average size of 100 nm) for two-photon-activated PDT. Two-photon-activated (TPA) dye-doped MSNs were synthesized using TEOS as a surfactant (Figure 22).

After 10 min irradiation with a two-photon laser (100 W xenon lamp) of HeLa cells incubated for 4 h with FITC-PGSN-RB (0.5 mg/mL), cell viability decreased to 30%.

Zhou et al. [54] synthesized NaYF_4_:Yb, Er (Y:Yb:Er = 78:20:2) as UCNPs, surrounded by a silica shell (SiO_2_) covalently coupled to RB, on which poly-(1,4-phenyleneacetone dimethylenethioketal) (PPADT) was self-assembled as a second outer shell contains the NPs and DOX. The size of UCN/SiO_2_-RB was 38.2 ± 3.6 nm without DOX and PPADT. ^1^O_2_ production was confirmed using ABDA. During NIR laser irradiation (980 nm, 1.0 W/cm^2^, 20 min), DOX release was induced, with 40% of the DOX being released after 12 h of irradiation of (A + DOX)@PPADT. The ^1^O_2_ production under NIR laser irradiation caused the biodegradation of the polymer and the release of DOX into the tumor. In vitro tests were performed on HeLa cells, and cell viability decreased to 25% and 15% for cells treated with 100 mg/mL and 200 mg/L (A + Dox)@PPADT, respectively, following NIR laser irradiation. This decrease was less significant in cells treated with A and A@PPADT, noting that cell viability was higher following treatment with A@PPADT than with A (Figure 23).

Custodio de Souza Oliveira et al. [55] compared the efficiency of SiO_2_ NPs functionalized with 3-(2-aminoethylamino)propyl groups (SiNP-AAP) either with covalently bound RB (SiNP-AAP-RB) or 9,10-anthraquinone-2-carboxylic acid (SiNP-AAP-OCAq) against human lung carcinoma A549 cells. Cell viability following incubation with SiNP-AAP-RB (50 µM) for 24 h decreased by approximately 80% under laser irradiation (410 nm, 1.6 J/cm^2^, 15 min). Although RB alone showed very high photoxity, NPs coupled to RB allowed passive targeting.

Zhan et al. [56] synthesized a magnetic mesoporous silica NP (MMSN): (poly-ethylene glycol-b-polyaspartate-modified RB-loaded magnetic mesoporous silica Fe_3_O_4_@nSiO_2_@mSiO_2_@RB@PEG-*b*-PAsp (RB-MMSNs). Fe_3_O_4_ served as the magnetic core, and could be useful for application in contrast agents. nSiO_2_ and mSiO_2_ were added, respectively, to avoid Fe_3_O_4_ oxidation and for RB loading. Polyethylene glycol-*b*-poly(aspartic acid) (PEG-*b*-PAsp) was coupled for its pH-responsive properties. RB was pre-loaded on previously prepared Fe_3_O_4_@nSiO_2_@mSiO_2_ NPs (MMSNs), before grafting of PEG-*b*-PAsp (Figure 24).

The size of the MMSNs was 190 nm. The loading efficiency of RB in MMSNs was about 35.74%. In vitro experiments were performed on B16 cells. The best photototoxicity (i.e., with a 75% reduction in cell viability) was obtained after incubation of cells with RB-MMSNs (50 µg/mL) for 12 h followed by green light irradiation (535 nm, 25 mW/cm^2^, 3 min) (Figure 25a). In vivo experiments were performed on C57BL/6J xenograft mice. After incubation for 18 days, the tumor volume decreased in all samples. The greatest decrease in survival was observed for cells treated with RB-MMSNs (Figure 25b).

Liu et al. [57] synthesized MSNs embedding carbon dots and RB in a core–shell structure with an average diameter of 104 nm. The core–shell structure prevented the self-aggregation of carbon dots and RB, enhanced the photoluminescence and ^1^O_2_ production, and allowed drug loading. The in vitro cytotoxicity of the conjugate was evaluated on H1299 cancer cells by MTT assay using laser light irradiation (540 nm, 300 mW/cm^2^, 5 min), and a reduction in cell viability of 90% was found. The conjugate presented a high capacity for imaging-guided chemo/PDT when DOX was incorporated into the NPs.

Yan et al. [58] designed MSNs conjugated with RB via an amide bond and with DOX via a pH-responsive linker (MSN-AH-DOX@RB) with a diameter of approximately 200 nm. The MSNs presented good biocompatibility and stability in an aqueous medium. DOX release in the acidic tumor microenvironment was highlighted when decreasing pH from 7.4 to 5.5. The cellular uptake of RB and DOX was demonstrated in MCF-7 cells. MSNs (0.5 mg/mL) showed minimal (~2%) dark cytotoxicity. MSN-AH-DOX@RB presented a higher cellular phototoxicity when evaluated by CCK-8 assays compared to control (RB or DOX or MSN-AH@RB or MSN-AH@DOX). ^1^O_2_ production was observed via SOSG. A laser at a wavelength of 532 nm (0.5 W/cm^2^) was used for 5 min at 1 min intervals for every 1 min of light exposure to illuminate MCF-7 incubated with MSN-AH-DOX@RB. The highest phototoxicity (90% efficiency) was observed for MSN-AH-DOX@RB plus laser irradiation for 5 min, showing a synergic chemo/PDT effect.

Jain et al. [59] reported the elaboration of silica-coated Gd_3_Al_5_O_12_:Ce^3+^ nano-platforms loaded with RB with a size of 74.13 ± 9.04 nm. ^1^O_2_ production in water was detected using DPBF, and was higher for RB-loaded NPs than for RB. In vitro experiments were performed on the MDA-MB-231 cell line. After incubation with RB-loaded NPs (200 μg/mL) followed by illumination with blue light (470 nm, 20 mW/cm^2^, 15–30–45 min), a decrease in cell viability (20%) was observed.

Jain et al. [60] synthesized magnetic-luminescent cerium-doped gadolinium aluminum NPs composed of Gd_2.98_Ce_0.02_Al_5_O_12_ (namely, GAG) coated with mesoporous silica (mSiO_2_) and loaded with RB (RB loading concentration = 20 μM), namely, the nanocomposite GAG@mSiO_2_@RB. The GAG core was synthesized using the sol–gel method and then loaded with RB. The average size of GAG was 18 nm, which increased to 110 nm following the addition of mSiO_2_. Adsorption/loading of RB increased the size of the NPs to 147 nm. The photoluminescence emission intensity was strongly dependent upon the dopant concentration within a range of [Ce^3+^] = 1–2% after excitation at 468 nm. The highest energy transfer was observed at an RB loading concentration of 20 µM. Upon excitation with X-rays (55 KV), GAG@mSiO_2_@RB continuously produced ROS, which was detected via DPBF. In vitro experiments using breast MDA-MB-231 cancer cells revealed a low dark cytotoxicity, with a cell survival rate of around 70% at concentrations of GAG@mSiO_2_@RB higher than 25 µM. The LC_50_ of GAG@mSiO_2_@RB was 6.69, 11.2 and 6.56 mg/mL at a doses of 0.16, 0.33 and 0.5 J/cm^2^, following irradiation at 15, 30 and 45 min respectively, at a wavelength of 470 nm, with 20 mW/cm^2^. Cell viability decreased with increasing blue light dose and nanocomposite concentration. At a wavelength of 470 nm and 0.495 J/cm^2^, and at a GAG@mSiO_2_@RB concentration of 50 µg/mL, cell viability decreased by 80% (Figure 26).

Hu et al. [61] synthesized drug-loaded ZIF-8@SiO_2_ NPs using a thermal-assisted microfluidic system. They used ZIF-8 as the MOF. RB, DOX, and pyrene were coated onto the ZIF-8@SiO_2_ (ZS) NPs in order to validate the universality of the microfluidic reactor system for the continuous fabrication of ZS nanocarriers, but only RB@ZS was used for cytotoxicity tests. The average size of RB@ZS particles was 93.8 ± 17.3 nm, and the drug loading was about 35.3 μg/mg. DPBF was used for the ^1^O_2_ probe. In vitro studies were conducted on 4T1 cell lines. These cells were incubated for 24 h with various concentrations of RB@ZS. Following laser irradiation (532 nm, 100 mW/cm^2^, 7 min), the viability of cells incubated with 100 µg/mL of RB@ZS decreased to 30% compared to cells treated with ZS only (~95%).

In vivo experiments (MTT) were performed on BALB/c female mice (aged 7–8 weeks). After incubation with RB@ZS (30 mg/mL), following exposure to 532 nm laser irradiation (500 mW/cm^2^) for 10 min, the relative tumor volume decreased after 2 days and then remained constant after 14 days, in contrast to controls, which showed an increase in relative tumor volume without a variation in mass, whatever the conditions (Figure 27).

Chen et al. [62] developed a pH-responsive and tumor-targeted drug delivery system for chemo-PDT. The system RB-DOX@HMSNs-N=C-HA was composed of hollow mesoporous silica NPs (HMSNs) loaded with DOX and RB and coated with oxidized hyaluronic acid (HA). The average diameter of the NPs was determined using TEM (170 ± 10 nm). The drug loadings of DOX and RB were 15.30% and 12.78% (*w*/*w*%), respectively, and the drug entrapment efficiency of DOX and RB were 76.67% and 95.85%, respectively. An in vitro drug release study was performed at different pH values (7.4, 6.0 and 5.0), and showed that, after 80 h, significant RB and DOX releases from RB-DOX@HMSNs-N=C-HA occurred at pH 5.0 and 6.0 (DOX: 39.05% and 58.60%; RB: 27.30% and 48.86% at pH 6.0 and 5.0, respectively), while the releases were negligible at pH 7.4 (DOX: 12.23%; RB: 12.27%). After excitation at a wavelength of 532 nm, ^1^O_2_ production was confirmed via DPBF. The targeted property provided by HA, which specifically recognizes the CD44 receptor, was confirmed by an in vitro 4T1 cellular uptake study. A study of in vitro cytotoxicity on 4T1cells was performed. At the maximum tested concentration (16 µg/mL, 4 h incubation), cell viability was 22.3% in the dark and 10.9% following laser irradiation (532 nm, 10 mW/cm^2^, 5 min). The IC_50_ value of RB-DOX@HMSNs-N=C-HA upon irradiation was 0.23 µg/mL, compared to 0.70 µg/mL and 8.89 µg/mL for DOX@HMSNs-N=C-HA and RB-@HMSNs-N=C-HA, respectively.

Miletto et al. [63] designed Ln:ZrO_2_@SiO_2_ (Ln = Er, Pr or Yb) NPs covalently coupled to RB for PDT treatment. The size of the NPs was approximately 20–25 nm. ^1^O_2_ production in water was determined using DPBF as the detector or using the spin tracking technique. In vitro experiments were performed on HeLa cells. After 10 min illumination (4 mW, 633 nm), a decrease in cell viability was observed (20–30%, depending on the type of lanthanide used).

Prieto-Montero et al. [64] synthesized biocompatible MSNs (with a size of 50 nm) with RB-PEG-FA covalently decorated onto their outer surface. Several sizes of PEG were tested, and the NPs that showed the highest stability in aqueous medium were the RB-PEG5000-NPs with a hydrodynamic diameter of 99 nm. The value of Φ_Δ_ in methanol was equal to 0.80–0.85. In vitro experiments on HeLa cells showed no dark cytotoxicity of RB-PEG-NPs and free RB in solution. When exposed to laser light irradiation (518 nm, 10 J/cm^2^, 5 min), RB-PEG-NPs showed a higher phototoxicity compared to RB alone in PBS at the same concentration of RB. The presence of FA on the outer surface of RB-PEG-MSN also contributed to the stability of the NPs by reducing interparticle aggregation. In vitro tests on HeLa cells with different concentrations of RB alone in PBS and RB-PEG-NPs upon exposure to laser irradiation showed that, when increasing the concentration from 0.1 µM to 10 µM, the cell viability decreased from 100% to about 10% following irradiation.

Table 2 summarizes the physical and chemical properties and PDT conditions presented in all references dealing with silica NPs@RB.

### 2.3. Organic NPs (Polymers, Micelles, Peptides)

Organic NPs are mostly biocompatible and biodegradable. They can be formed as nanospheres or nanocapsules. To formulate these NPs, different types of polymers can be used: poly(lactic acid) (PLA) [65], poly(glycolic acid) (PGA) [66] and poly(lactide-coglycolide) (PLGA) [67]. The use of hydrophilic and hydrophobic block copolymers in the formulation of these NPs improves their biodispersity. Natural or synthetic proteins are one type of polymer that can be used [68]. In addition, they can be used for coupling or encapsulating drugs, as well as as imaging and targeting agents.

#### 2.3.1. Polymer NPs

Karthikeyan et al. [69] succeeded in encapsulating RB in nanoscale dendrimers (G2.5 PAMAM (Poly(amidoamine) + RB). The size of G2.5 PAMAM was 20 nm. They demonstrated values of RB release of 35%, 50%, 74% and 83% after 12, 24, 48 and 72 h, respectively. ^1^O_2_ production was quantified by I^−^ in the presence of ammonium molybdate. In vitro experiments were performed on DLA cells, revealing that RB and G2.5 PAMAM + RB exhibited low dark cytotoxity. For cells incubated with G2.5 PAMAM + RB for 18 h, cell viability decreased by up to 24.9% at an RB concentration of 500 nM, and by up to 40% for cells incubated with RB alone at the same concentration upon exposure to a 150 W xenon lamp (540 nm, 10 min) (Figure 28).

Baumann et al. [70] developed polymer vesicles made of poly(2-methyloxazoline)–poly(dimethylsiloxane)–poly(2-methyloxazoline). These NPs were used to encapsulate RB-BSA conjugate (i.e., RB-BSA, hydrodynamic radius of 109 ± 5 nm). ^1^O_2_ production was determined using 2,2,6,6-tetramethyl-4-piperidinol (TMP–OH) as ^1^O_2_ scavenger. In vitro studies were performed on HeLa cancer cells and revealed no cytotoxicity of RB-BSA NPs at different concentrations (between 0 and 300 μg/mL) after incubation for 24 h. The light-controlled PDT effect of RB-BSA NPs was evaluated using irradiation doses of 30–135 J/cm^2^ for 5–30 min with laser irradiation at wavelengths of 405 nm, 543 nm, and 633 nm. Visualization of the HeLa cells plasma membrane using confocal microscopy indicated the formation of blebs only upon laser irradiation at a wavelength of 543 nm, while the plasma membrane remained intact upon laser irradiation at wavelengths of 405 nm and 633 nm.

Bhattacharyya et al. [71] synthesized coumarin 153 (C153)-dye-doped poly(N-vinyl carbazole) (PVK) polymer NPs. The FRET efficiency was 60%. No biological experiments were performed.

Han et al. [72] successfully synthesized a PAH-modified UCNP/hyaluronate-RB covalently coupled (UCNP/PAH/HA-RB) in the form of a hexagonal (100) crystal lattice with a size of 30.34 ± 2.10 nm. The composition of UCNPs was 78:19:3 of Y:Yb:Er. The hydrodynamic diameter of these NP was 459.9 ± 98 nm. In vitro tests on NIH 3T3 cells showed that, by increasing the concentration of UCNP/PAH, cell viability decreased more significantly following laser irradiation (980 nm, 1.5 mW/cm^2^, 30 min). Nevertheless, at a concentration of 200 µg/mL, cell viability was maintained at more than 90% (Figure 29).

Yang et al. [73] synthesized PLGA NPs with a diameter of 28 nm, where RB was grafted via an amide bond. Firefly luciferin was then conjugated to the platform at a short distance from RB to allow efficient energy transfer (Figure 30).

The NPs produced a fluorescent signal in the presence of luciferin that was able to excite RB with a bioluminescence resonance energy transfer (BRET) efficiency of around 58%. MTT assays performed on MCF-7 and HeLa cells showed no cytotoxicity of the NPs at a concentration of 50 μg/mL following incubation for 48 h. With increasing concentration of luciferin, cell viability estimated by MTT decreased to 45%, with the optimal value being reached at 60 μg/mL. In vivo experiments performed on H22 tumor-bearing mice demonstrated apparent tumor growth inhibition in the BRET-PDT group, whereas the tumors in the control group exhibited rapid growth (520 nm, 200 mW/cm^2^, 30 min) (Figure 31).

Chang et al. [74] developed photo-activated ROS-responsive nanocarriers (NCs) containing Paclitaxel (PTX) and RB for combinational chemotherapy and PDT. THe HA-BSA/CTS/PVA/bPEI-blended NCs (HBNCs) were composed of chitosan (CTS), poly(vinyl alcohol) (PVA) branched with polyethylenimine (bPEI), and BSA. The dual-functional drug carriers RB/PTX-HBNCs were loaded with PTX and RB. This core was shelled with HA as a result of electrostatic bonds. The hydrodynamic diameter of HBNCs was 220 ± 14 nm. The encapsulation efficiencies (EE%) of RB and PTX were 60.7 7 ± 2.7% and 55.2 ± 8.9%, respectively. A DCFH-DA probe was used to determined ROS generation. It was found that 30% of PTX was released after an exposure time of 6 h, whereas negligible release was observed without light irradiation. Cellular uptake of RB/PTX-HBNCs in Tramp-C1 prostate cancerous cells expressing CD44 receptor was observed, confirming the HA-specific recognition and targeting ability of HBNCs. Significant intracellular ROS generation and PTX delivery was observed in Tramp-C1 cells following incubation with RB/PTX-HBNCs for 6 h after light irradiation for 1 h at a wavelength of 632 nm (15 mW/cm^2^). A cell viability assay was performed to evaluate the antitumoral effect of RB/PTX-HBNCs. In the dark, cell death of 18% was observed, indicating that the NCs possessed a moderate toxic effect. After irradiation for 1 h at a wavelength of 632 nm (15 mW/cm^2^), cell death increased to 24% for PTX-HNCs and to 40% for RB-HBNCs. RB/PTX-HBNCs showed an improved cytotoxic effect, with cell death increasing to 61%.

Bazylińska et al. [75] engineered double-core NCs for the co-delivery of RB and trioctylphosphine oxide (TOPO)-stabilized luminescent lanthanide-doped NaYF_4_:2%Er^3+^, 20%Yb^3+^ NPs to human melanoma for theranostic applications. The double core was composed of a non-ionic surfactant, Cremophor A25, and stabilized by PLGA copolymer. The loaded NCs had a hydrodynamic diameter of 158 nm. Under 980 nm laser excitation, NaYF_4_:Er^3+^,Yb^3+^ NPs showed three emission peaks, at 520, 540, and 660 nm. The green emission bands overlapped the RB absorbance spectra, leading to efficient energy transfer by FRET at a level of 25%. In vitro cytotoxicity assays were performed on two tumorous cell lines (human melanoma granular fibroblast (MeWo), lymph node metastasis of skin melanoma (Me-45)), as well as on non-tumorous human cutaneous keratinocyte (HaCaT) as control cells. Empty NCs under dark or light conditions (520–560 nm, 10 J/cm^2^, 5 min) showed no cytotoxicity. Free RB presented moderate cytotoxicity under irradiation for the three cell types, while it presented high toxicity for MeWo (>90%) and Me-45 (55%) after incubation for 48 h, but not for HaCaT cells (25%) (Figure 32).

#### 2.3.2. Peptide NPs

Sun et al. [76] designed cationic dipeptide NPs conjugated with bis-pyrene (BP) as an energy donor and RB as an acceptor. BP and RB were enveloped in spherical NPs with a diameter of 280 nm. The overlap between BP emission and RB absorption was high, allowing efficient energy transfer that was estimated to be 46%. The generation of ^1^O_2_ was confirmed by means of an ABDA probe. Confocal laser scanning microscopy confirmed an internal uptake of the conjugate in MCF-7 cells, which presented good biocompatibility and low dark cytotoxicity. After co-culture for 48 h and irradiation at 480 nm (245 mW/cm^2^) for 20 min, cell viability decreased to 27%. Irradiation (245 mW) at a wavelength of 810 nm for 50 min induced high two-photon photo-cytotoxicity.

Liu et al. [77] developed nanocapsules (PARN) that were composed of an amphiphilic peptide and RB for use in sonodynamic therapy and PDT. The amphiphilic peptide C_18_GR_7_RGDS was composed of a carbonate chain and an RGD peptide for targeting the αvβ3 integrin receptor. This peptide was able to self-assemble owing to electrostatic interaction, and could be loaded with RB via weak interactions. The hydrodynamic diameter of PARN was 17.28 ± 0.88 nm. ^1^O_2_ generation was confirmed by means of a DPBF probe. PARN was irradiated by laser at a wavelength of 808 nm, 1.5 W/cm^2^ for 3 min, or by ultrasound (US) (50 MHz, 1 W/cm^2^, 5 min). ROS generation was observed under both laser and US irradiation. B16 (melanoma cells) and HeLa cells were incubated for 24 h with PARN or RB and irradiated using either laser or US. In vitro ROS generation was determined using CLSM (confocal laser scanning microscopy) and DCFH-DA probes. Cells incubated with PARN under US or laser irradiation induced ROS production. Cytotoxicity tests were performed with B16 and HeLa cells incubated with PARN (40 µg/mL). Without irradiation, cell viability was 30.2% and 29.7% in B16 and HeLa cells, respectively. After 5 min of US irradiation, cell viability decreased significantly to 4.7% and 4.1% in B16 and HeLa cells, respectively. Thus, both irradiation modalities were equivalent. In vivo sonodynamic therapy and PDT were evaluated using HeLa tumor-bearing mice, with five groups being used in this study, as follows: (1) control, (2) PARN, (3) PARN + laser, (4) PARN + US, and (5) PARN + laser + US (treatment for 18 days (US: frequency: 1.0 MHz; duty cycle: 50%; power density: 1.0 W/cm^2^; wavelength: 808 nm; duration: 3 min). No tumor growth inhibition was observed for group 2, whereas, under US, laser or laser + US irradiation, moderate tumor growth inhibition was observed. Thus, the efficiency of PARN irradiated with US or laser needs to be improved before proposing treatment with highly promising results (Figure 33).

#### 2.3.3. Micelle NPs

Korpusik et al. [78] reported the design of a targeted micelle-based PDT system (Sgc8 micelles, hydrodynamic diameter of 52 nm) based on azide-functionalized polymeric micelles decorated with protein tyrosine kinase 7 (PTK7)-binding DNA (Deoxyribonucleic acid) aptamer (Sgc8) via click chemistry. The covalent conjugation of RB to micelles provided quantitative and controlled RB loading. The use of ADPA revealed that Sgc8 micelles had a much higher ^1^O_2_ production than free RB, and a controllable “on-off” release of ^1^O_2_ using yellow light (590–595 nm, 0.2 mW/cm^2^, 15 min). These RB-loaded micelles also exhibited better fluorescence emission than free RB. In vitro experiments were performed on PTK7-expressing HCT116 cancer cells and K562 cells lacking PTK7 receptors under 15 min of yellow light exposure. The cell targeting capacity of Sgc8 micelles compared to unlabeled ones was clearly demonstrated. The authors showed that Sgc8 micelles could be used to achieve selective PDT, but that it is necessary to be very careful regarding the concentration used (i.e., Sgc8 micelles exhibit significant cytotoxicity at high doses (e.g., 5.4 μM)).

Table 3 summarizes the physical and chemical properties and PDT conditions presented in all references dealing with organic NPs@RB.

### 2.4. Gold Nanoparticles

Gold NPs are assemblies of 30 to 40 million gold atoms. The size of these NPs depends on their synthesis conditions, and mainly ranges between 1 and 100 nm [79]. For centuries, gold NPs have been used in clinical chemistry, laser phototherapy of cancer cells and tumors, targeted delivery of drugs, DNA and antigens, optical bioimaging, and monitoring of cells, tumors, and tissues using state-of-the-art detection systems. They are used in diagnosis, therapy, prevention, and hygiene. They can be used in these applications owing to their unique physical and chemical properties. In particular, the optical properties of gold NPs are determined by their plasmon resonance, which is associated with the collective excitation of conduction electrons and localized in the wide region, from visible to infrared (IR), depending on the size, shape, and structure of the particles. Having discovered the properties of biocompatibility and photostability possessed by gold NPs, scientists have also been attracted to their exceptional photophysical and optical properties [80,81]. These properties have led researchers to apply these NPs in cancer diagnosis and therapy [82].

Wang et al. [83] synthesized a gold nanorod with a diameter of 13 ± 2 nm and a length of 52 ± 4 nm in which RB was encapsulated (RB-GNRs) for application in PTT–PDT. The seed-mediated growth method was followed, and PAH was used to modify the surface of the CTAB-coated GNRs. RB was then adsorbed onto the surface. ^1^O_2_ production was observed in vitor via ABDA in water (ABDA absorption decreased by 92% with light and RB-GNRs, while the reduction was only 15% without light or without RB, while a decrease of 85% was observed with RB alone) and DCFH-DA. In vitro experiments were performed on Cal-27 cells and a home-made light-emitting diode at a wavelength of 530 ± 15 nm (170 mW/cm^2^ for 90 s). Total cell survival after light illumination for 12 h was 19%, and 26.3% after light illumination for 24 h. In vivo experiments were performed on Male Syrian Golden hamsters with oral carcinomas induced by DMBA (7,12-dimethylbenz(a)anthracene) in acetone. After PDT (532 nm solid-state diode laser, 1.76 W/cm^2^, for 10 min), an inhibition rate of 46.5% was observed on the 10th day, while 65.5% was observed after PTT (810 nm NIR light, 8.16 W/cm^2^, 5 min), and 95.5% after combined PTT–PDT (Figure 34).

Prasanna et al. [84] designed glutathione (GSH)-capped gold NPs. RB was covalently attached via an amide bond between RB and GSH with EADC (1-Ethyl-3-(3-dimethylaminopropyl)carbodiimide) (Figure 35) or non-covalently coupled via electrostatic interactions.

The size of the AuNPs-RB electrostatic complex was 53 nm, which is similar to the covalent complex, at 49 nm. PDT experiments were performed on Vero and HeLa cell lines using white LED light (500 nm, 5 J/cm^2^). For assays in Vero cells, free RB and electrostatic and covalent AuNPs-RB complexes presented IC_50_ values of 19.53, 16.69, and 9.18 μg/mL (3 h), respectively. These values were 16.99, 14.47 and 8.24 μg/mL in HeLa cells. The AuNPs-RB complexes were more toxic than free RB, and the covalent complex was more toxic than the electrostatic complex.

Kautzka et al. [85] synthesized a liposome of hydrogenated soy L-a-phosphatidylcholine (HSPC) and 1,2-dioleoyl-*sn*-glycero-phosphoethanolamine-*N*-(hexanoylamine) (PE-NH_2_) encapsulating gold NPs, RB, and DOX. The size of the HSPC:PE-NH_2_-gold NPs was dependent on the formulation (i.e., 130.1 ± 1.0 nm for 57:5:8.5; 124.6 ± 2.3 nm for 57:5:17; 131.1 ± 1.3 nm for 57:5:34). ^1^O_2_ production was observed by means of an SOSG probe. For the HSPC:PE-NH2-gold NP formulation of 57:5:17, the relative Φ_Δ_ was found to be the highest, with a value equal to 1.33 ± 0.27, and an enhancement factor of 1.75 compared to liposomes without gold NPs. For the HSPC:PE-NH_2_-gold NP formulation of 57:5:8.5, an enhancement factor of 1.53 was found. No enhancement was detected for the HSPC:PE-NH2-gold NP formulation of 57:5:34. Human colon adenocarcinoma (HCT116) and human breast cancer (MCF-7) cell lines were chosen for in vitro experiments following green light irradiation (532 nm, 14.3 mW/cm^2^, 6 min) with the HSPC:PE-NH_2_-gold NP formulation of 57:5:17. In HCT116, the highest cell mortality following illumination was obtained for HSPC:PE-NH_2_-gold NP +RB+DOX, with 38% of cells being killed at a lipid concentration of 15.7 µg/mL. Similar results were observed in MCF-7. No dark cytotoxicity was observed in HCT116, MCF-7, or normal human colon epithelial cells (CCD 841 CoN).

Fu et al. [86] synthesized CaTiO_3_:Yb,Er (CTO) nanofibers covalently conjugated with RB and gold nanorods (CTO-RB-AuNRs). The diameter distribution for CTO was 0.273 nm for nanofibers without AuNRs and RB. ^1^O_2_ production was detected using a DPBF probe under laser irradiation at a wavelength of 980 nm (1.5 W/cm^2^, 0–12 min). In vitro tests were performed on Hep G2 cells. The cells were incubated with CTO, CTO-RB, and CTO-RB-AuNRs and CTO-AuNRs. After incubation with CTO-RB-AuNR (200 µg/mL) for 24 h followed by illumination (980 nm, 1.5 W/cm^2^, 6 min), the cell viability decreased to 15%.

Table 4 summarizes the physical and chemical properties and PDT conditions presented in all references dealing with gold NPs@RB.

### 2.5. Polymer Dot Nanoparticles

Research has confirmed the importance of semiconducting polymer dots (Pdots) in biological applications. They are considered an essential tool for the diagnosis and treatment of certain diseases, thanks to their high luminosity, their long lifetime, their adjustable size, their narrow luminescence and their biocompatibility [87,88].

Haimov et al. [89] described the synthesis of polymer dots (Pdots) (chromophoric organic homo-polymers) using three types of phospholipids (phosphoethanolamine, PE): PEG350-PE, PEG2000-PE, and PEG5000-PE. Pdots have never before been used for PDT applications. RB or methylated RB was bound non-covalently but with high affinity. The sizes of the Pdot NPs were 43 ± 9 nm, 77 ± 0.3 nm, and 60 ± 0.15 nm for PEG350-PE, PEG2000-PE, PEG5000-PE, respectively. By using DMA (9,10-dimethylanthracene) the value of Φ_∆_ in ethanol of RB was measured as 0.38, 0.24, and 0.16 for PEG350-PE, PEG2000-PE, PEG5000-PE, respectively, while that of methylated RB was determined to be 1.0, 0.81, and 0.76 for PEG350-PE, PEG2000-PE, PEG5000-PE, respectively. In vitro experiments were performed in MCF-7 (473 nm, 0.15 mJ/cm^2^, between 1 h and 4 h). A decrease in cell viability was observed after illumination (with a reduction of 8.5% being observed after 1 h and 50% after 4 h) following incubation for 4 h with PF-PEG350 Pdots.

Hua et al. [90] developed fluorescent carbon quantum dots (CQDs) made of chitosan, ethylenediamine, and mercaptosuccinic acid, with RB being covalently grafted by means of amide bonds (CQDs-RB). Diameters of 33.1 ± 8.7 nm and 2.1 ± 0.3 nm for CQDs-RB and CQDs, respectively, were determined. ^1^O_2_ production was observed by means of SOSG probe. In MCF-7 cells, the CQDs-RB were localized in the mitochondria. After PDT (532 nm, 30 min, 20 mW/cm^2^), CQDs-RB NPs were more effective than RB alone, with a decrease in cell viability from 100% to 11% being observed when the laser power was increased from 0 to 30 mW.

Table 5 summarizes the physical and chemical properties and PDT conditions presented in all references dealing with Pdots@RB.

### 2.6. Other Nanoparticles

#### 2.6.1. Nanocapsules

A nanocapsule is composed of a core and a shell. The shell is composed of a polymer that can be coated with several types of targeting agents. The hydrophobic drug to be delivered is placed in the core. The size of the nanocapsules is between 10 and 1000 nm. The encapsulation of the drug serves to protect it from its environment during delivery. Lipid NPs have been suggested for use in cases of multidrug resistance in tumors [91]. The diffusion of drugs to the target is achieved by means of chemical, thermal or biological triggering.

Zhang et al. [92] successfully synthesized 300 nm bifunctional nanospheres (BFNS). The BFNS were a mixture of partially hydrolyzed α-lactalbumin (PHLA), RB, Gd^3+^ (for MRI), and Ca^2+^, coupled to RGD peptide. ^1^O_2_ production was confirmed by DPBF. In vitro tests were performed on HepG2 cells. After irradiation at 550 nm (100 mW/cm^2^, 15 min), cell viability decreased with increasing concentration of BFNS and BFNS-RGD (0, 6.25, 12.5, 25, 50, 100, 200 μg/mL). At a concentration of 200 µg/mL, cell viability was lower when treated with BFNS-RGD compared to with BFNS (with values of 19% and 49%, respectively, being observed).

#### 2.6.2. Nanocomplexes

A nanocomplex [93] is an NP containing a chemical complex. A complex is a polyatomic structure consisting of one or more independent interacting entities.

Cao et al. [94] synthesized an organic nanocomplex composed of carboxylated bis(pyrene) molecules (BP) and RB-PAH. The core of the NPs consisted of BP, and the shell consisted of RB-PAH. Nanoaggregates of BP coated with RB-PAH provided the desired nanocomplex BP-RB, via the self-assembly method, for a use in two-photon PDT by exploiting the FRET effect. TEM images showed a diameter of about 70 nm for the BP-RB nanocomplexes. ^1^O_2_ production was observed using ABDA. In vitro studies were conducted on MCF-7 cells. The lowest cell viability (21%) was observed for cells incubated with 36 µM for 4 h upon exposure to a 100 W xenon lamp (810 nm, 20 min).

#### 2.6.3. Magnetic Nanoparticles

Magnetic NPs [95] have a size on the order of 1–100 nm. They are generally composed of a metal core (iron, cobalt, nickel, etc.) coated with other molecules. Magnetic NPs can be used as a contrast agent in biological imaging. They can improve image resolution and information content. Superparamagnetism finds application in data analysis and medicine, in particular, ferrofluids, owing to their viscosity [96].

Yeh et al. [97] synthesized magnetic nanoclusters (MNCs) coated with a tri-polymer and encapsulating RB using an oil-in-water emulsion method. The core consisted of iron oxide NPs (Fe_3_O_4_ NPs). The shell was composed of three polymers: CTS, PVA (poly(vinyl alcohol)), and bPEI, with a weight ratio of CTS:PVA:bPEI:Fe_3_O_4_ of 10:2:4:1. RB was incorporated into the polymeric coating by means of electrostatic interaction in order to give the final RB:MNCs. The encapsulation efficiency of RB was estimated to be about 87% (*w*/*w*%). The hydrodynamic size of RB:MNCs was about 108.6 ± 11.7 nm at pH 7.4. ^1^O_2_ production was confirmed using DCFH-DA upon irradiation at a wavelength of 632 nm (15 mW/cm^2^) for 1 h. A rate of cell death of 73 ± 3% was observed following illumination with a laser at a wavelength of 532 nm (15 mW/cm^2^) for 5 min (7.5 µM of RB and 6 h). These strong PDT effects of RB:MNCs were also confirmed in mouse prostate cancer cell lines. No dark cytotoxicity was observed in Tramp-C1 cells or human ovary cancer cell line SKOV-3, but strong cell death was observed upon irradiation with a green laser (532 nm, 100 mW/cm^2^, 5 min). Furthermore, PTX was also encapsulated with RB (74.4% EE%) to prove the combined chemo/PDT effect of such nanosystems. RB/PTX:MNCs (with an RB concentration of 7.5 µM) showed a cytotoxic effect on MCF-7 cells upon irradiation with a green laser (532 nm, 100 mW/cm^2^) for 5 min, and cell viability decreased by almost 80%. In vivo studies were conducted in BALB/c nude mice. In this context, drug-resistant human breast cancer cells MCF-7/MDR were injected into nude mice. At 4 h post injection with RB:MNCs and exposure to a green laser (532 nm, 100 mW/cm^2^) for 5 min, a strong cytotoxic effect was observed. Tumor growth was significantly reduced (with an inhibition rate of 85% being observed). Moreover, RB:MNCs were not toxic to healthy tissues and organs in the 38 days following intratumoral injection. By conducting a TUNEL (terminal deoxynucleotidyl transferase dUTP nick end labeling) assay, at 4 h post injection, strong cell apoptosis was observed, induced by RB:MNCs (124 μg and 0.25 mg per mouse of RB and Fe_3_O_4_ NPs, respectively) in the treated mice 24 h after PDT treatment (Figure 36).

#### 2.6.4. pH-Sensitive Nanoparticles

pH-sensitive NPs [98] make it possible to perform therapy by achieving either protonation or deprotonation of the NP system. The change in pH enables the delivery of the drug or the triggering of its application system.

Li et al. [99] worked on improving the phototoxicity of RB and enhancing tumor targeting using hypoxia/pH dual-responsive RB-NPs. Cs-Na/RB and Cs-Na/RBD were formed by the self-assembly of nitroimidazole-modified chitosan (Cs-Na) with RB or RBD (RB ω-carboxyheptyl ester) owing to electrostatic and intramolecular interactions. Chitosan was modified by nitroimidazole in order to obtain hypoxia-responsive NPs (nitroimidazole was transformed into its hydrophilic form by nitroreductases leading to the destabilization of NPs, and the release of dyes in cells). The modification of RB was carried out in order to adjust the amphiphilicity and to change the biodistribution of RB by increasing blood circulation time and tumor tissue distribution. The hydrodynamic diameters of Cs-Na/RB and Cs-Na/RBD were determined by DLS (with the values being found to be 86.9 and 115.4 nm, respectively), while their diameters were determined using AFM (50 and 60 nm, respectively). The drug loading capacity of Cs-Na was found to be 23.5 ± 2.5% for RB and 26.4 ± 1.9% (*w*/*w*%) for RBD. In vitro drug release in PBS was determined at pH 7.4 and 5.5 with and without Na_2_S_2_O_4_ in order to simulate hypoxia conditions. Dye release was observed for both Cs-Na/RB and Cs-Na/RBD under normal conditions (pH 7.4), but significantly increases, by 22.5% and 19%, respectively, were observed under acidic condition, by 14.5% and 14%, respectively, under hypoxia condition, and by 25.5% and 30%, respectively, under acidic and hypoxic conditions. A cytotoxicity study was performed using PC9 cells. At 10 µg/mL under light irradiation (518 nm, 4.0 W/cm^2^, 10 min), RB and RBD showed moderate cell inhibition (demonstrating cell viability of 74.4% and 60.8%, respectively). Without light irradiation, Cs-Na/RB and Cs-Na/RBD exhibited moderate cell inhibition (cell viability of 80.7% and 80.6%, respectively), whereas they exhibited enhanced cell elimination upon light irradiation (with values of 68.8% and 62.2%, respectively). Under hypoxic conditions and under light irradiation, cell was enhanced even further for Cs-Na/RB and Cs-Na/RBD (60.1% and 50.2%, respectively). ROS production from RB, RBD, Cs/RBD, Cs-Na/RB and Cs-Na/RBD on PC9 cells was evaluated by means of a DCFH-DA probe. The results showed that, under the same concentration, RB, RBD, Cs/RBD, Cs- NA/RB, and Cs-NA/RBD with concentrations equivalent to 10 μg/mL of RB or RBD produced 4.1, 5.2, 7.2, 7.7 and 10.3 times more ROS compared to the control group without application of any of the samples.

#### 2.6.5. Nanohybrid Nanoparticles

Hybrid NPs have both organic and inorganic compositions [100]. Su et al. [101] synthesized a multi-functional nanohybrid system, UCNPs@MF-RB/PEG. First, PAA was coated with UCNPs (NaYF_4_,Yb/Er@NaYF_4_ (Y/Yb/Er: 78/20/2). Second, MnFe_2_O_4_ (MF) was linked through an intermediate covalent bond and coated using PEI. Finally, PEG-COOH and RB were covalently bonded to this new hybrid system by means of the formation of amide bonds. UCNPs were used to convert photons from NIR into the 521/545 nm wavelength range, which could be further used to activate RB. ^1^O_2_ production was observed by means of DPBF. In vitro studies were conducted on HeLa cells with four treatment groups: (1) RB, (2) RB+H_2_O_2_, (3) UCNPs@MF-RB/PEG, (4) UCNPs@MF-RB/PEG+H_2_O_2_. For groups 1 and 2, cell viability remained almost unchanged. Cell viability decreased significantly for groups 3 and 4, and this change was more pronounced for group 4 (Figure 37). This indicated the importance of the presence of H_2_O_2_ for the production of ROS.

#### 2.6.6. Nanogels

A nanogel is composed of a hydrogel [102], a network of cross-linked hydrophilic polymers that is most often synthetic. These polymers can also be physically or chemically cross-linked biopolymers. The size of a nanogel is usually 10–100 nm. The pores of the nanogels can be filled with small or large molecules.

Torres-Martínez et al. [103] showed that nanogels adapted for the intracellular transport of PDT agents such as RB did not show cytotoxicity in the dark. ^1^O_2_ production was confirmed in water using an ABDA probe. In vitro tests were performed on HT-29 cells incubated for 24 h with RB alone, PBS, and RB@1 (PS-loaded nanogel) ([RB] = 2 μM) and irradiated with two 11 W LEDs (λ_em_ = 400–700 nm) for 2 min. The results showed that RB@1 induced apoptosis in more than 70% of cultured cells, while this value was only 15% for free RB. Under the same conditions, RB@1 showed no dark cytotoxicity.

Table 6 summarizes the physical and chemical properties and PDT conditions presented in all references dealing with nanogels, nanohybrid NPs, pH-sensitive NPs, magnetic NPs, nanocomplexes, and nanocapsules@RB.

## 3. X-ray Photodynamic Therapy (X-PDT)

### 3.1. Silica Nanoparticles

Elmenoufy et al. [104] synthesized scintillating NPs (ScNPs) based on LaF_3_:Tb coated with layers of silica. These ScNPs were covalently coupled to RB. ^1^O_2_ production was confirmed using DPBF. No biological tests were performed.

Hsu et al. [105] designed an ScNP core–shell–shell, Eu(15%)@NaLuF_4_:Gd(40%)@NaLuF_4_:Gd(35%),Tb(15%) encapsulating RB and coupled to PEG-FA (ScNP-PAH-RB-PEG-FA). The production of ^1^O_2_ following energy transfer was confirmed by ABDA probe (0–10 Gy). In vitro tests were performed on MCF-7 and MDA-MB-231 cells incubated for 24 h with ScNP-PAH-RB-PEG-FA (50 μg/mL). Cell viability decreased by 31% and 21% when MDA and MCF-7 cells, respectively, were treated with ScNP-PAH-RB-PEG-FA before X-ray irradiation (1, 3 and 5 Gy, 75 kV, 20 mA, 20 min) (Figure 38).

Sun et al. [106] synthesized silicate nanoscintillators with controllable size and X-ray-excited optical luminescence (450–900 nm) using a general ion-incorporating silica-templating method. The NPs were conjugated to RB and RGD peptide (RGD-ZSM-RB). Both in vitro and in vivo experiments demonstrated that RGD-ZSM-RB incubated for 24 h increased the inhibitory effect on tumor progression under low-dose X-ray irradiation (1 Gy). The cell viability of U87MG decreased to half with RGD-ZSM-RB at a concentration of 80 µg/mL following X-ray irradiation of 1 Gy (Figure 39a). In vivo treatment in U87MG tumor-bearing mice induced tumor death following irradiation at 1.0 Gy, and an RGD-ZSM-RB concentration of 20 mg/kg (Figure 39b). After 48 h, the accumulation of RGD-ZSM-RB (20 mg/mL) was significant in the liver and spleen. Two hours after the injection of RGD-ZSM-RB (20 mg/kg) into the mice, strong fluorescence (ex/em: 570/585 nm) and CT imaging signals (1 Gy) of the RB were observed, and reaching their maximum values after 8 h.

Ahmad et al. [107] developed scintillation NPs of CeF_3_ co-doped with Tb^3+^ and Gd^3+^ (CeF_3_:Gd^3+^, Tb^3+^:CGTS), with Gd^3+^ (12.3 mol%) and Tb^3+^ (1.24 mol%) covered with a layer of mesoporous silica loaded with RB (CGTS-RB NPs). The size of the CGTS was 80 ± 1 nm. ^1^O_2_ production was confirmed using SOSG. In vitro studies were performed on 4T1, Renca, and Mgc89 cells under several different X-ray doses (0–8 Gy). After incubation of these cells for 24 h with 50 μg/mL of CGTS-RB, irradiation of the cells at 2 Gy revealed decreased cell viability in the three cell lines (survival rates towards 41%) (Figure 40).

In vivo tests were performed on 4T1 tumor-bearing mice. Intravenous injection of CGTS-RB (2 mg/mL, 200 μL) and their X-ray irradiation (160 kV, 25 μA, 6 Gy) showed inhibition of tumor proliferation over 14 days (Figure 41).

Table 7 summarizes the physical and chemical properties and X-PDT conditions of nanocapsule@RB presented in all references dealing with magnetic NPs@RB and silica NPs@RB.

### 3.2. Polymer Nanoparticles

Bekah et al. [108] synthesized LaF_3_ NPs doped with either Ce^3+^, Tb^3+^, or both (size average of 4 nm). These NPs were surface modified with alendronate to covalently link PEG and PSs (either Ce6 or RB). No biological experiments were performed.

Maiti et al. [109] reported polyoxomybdate monoclusters loaded (68% (*w*/*w*%)) with RB, namely POMo NCs. These NPs were functionalized with chitosan and PEG and given a macro complex, namely a strawberry shape, with an estimated size of 5.9 nm. Moreover, chitosan formed a hollow into which RB was adsorbed. The POMo NCs were designed firstly for X-ray PDT producing ^1^O_2_, and secondly for X-ray-inducible radiation generating auger electrons to induce DNA damage at low X-ray dose deposits of 2 Gy. The impact of the NPs was tested with mouse breast cancer 4T1 cells in vitro and in vivo, following the subcutaneous grafting of cells on BALB/c strain mice. In vitro, cell viability decreased with increasing amounts of NPs and RB. After treatment with 100 µg/mL POMo NCs@RB under 2 Gy/min, cell viability decreased from 100% to 20% (Figure 42a). In vivo, the application of POMo NCs@RB (25 mg of Mo/kg) in 4T1 tumor-bearing mice induced total inhibition of tumor volume proliferation (Figure 42b).

Sun et al. [110] explored the combination of PDT and radiation therapy using gadolinium-RB coordination polymer nanodots (GRDs) to treat breast tumor. The hydrodynamic diameter of GRDs was 7.7 ± 1.4 nm. RB and GRDs were excited at 525 nm and respective values of Φ_f_ of 0.11 and 0.97 were observed in ethanol. Thus, an increase in Φ_f_ by 7.7 times was observed, demonstrating that immobilization of RB in GRDs induced enhanced fluorescence due to the limitation of the rotation of the aromatic structure. Φ_Δ_ was also calculated to have values of 1.46 for GRDs and 0.75 for RB in water, showing an increase in Φ_Δ_ by 1.94 times for GRDs, proving their suitability for PDT. The ^1^O_2_ generation capacity of GRDs was determined by ABDA probe. 4T1 cells were incubated with 24 µg/mL of RB or GRDs for 24 h and irradiated at a wavelength of 532 nm for 5 min (30 mW/cm^2^), and cell mortalities of 63.5% and 36.4% were obtained for GRDs, and 36.4% for RB. After irradiation (532 nm, 30 mW/cm^2^, 10 min), cell viability was 37.4 ± 1.3%, demonstrating the synergistic effect of combined irradiations (Figure 43a). An in vivo therapy assay was performed on 4T1 tumor-bearing mice. Eight groups of mice were employed to evaluate the efficiency of PDT, radiation therapy, and combination therapy: (1) PBS, (2) PBS + light, (3) PBS + X-ray, (4) PBS + light + X-ray, (5) 10 mg/kg GRDs, (6) GRDs + light, (7) GRDs + X-ray, and (8) GRDs + light + X-ray (light = laser irradiation at 532 nm, 140 mW/cm^2^ for 15 min and X-ray = 1Gy). No body weight loss was observed in any of the groups. Groups 6 and 7 exhibited good tumor growth inhibition rates (50.5 and 43.8%, respectively), while group 8 exhibited an excellent tumor growth inhibition rate of 98.8% and possessed the smallest tumor volume at the end of the study (Figure 43b).

Table 8 summarizes the physical and chemical properties and X-PDT conditions presented in all references dealing with polymer NPs.

### 3.3. Nanocomposites

Zhang et al. [111] designed β-NaGdF_4_:Tb^3+^ NPs and RB (average size of 9 nm). RB was covalently coupled to the NPs. As the NPs were hydrophobic, they were functionalized with 2-aminoethylphosphonic. In this NPs-RB system, ^1^O_2_ production was high, due to a high FRET efficiency. In vitro and in vivo biological studies were performed on human hepatocarcinoma HepG2 cells and mice bearing HepG2 cell tumors. Compared to NPs alone at different concentrations, NPs-RB demonstrated a decrease in cell viability from 100% to 20% (1 mg/mL after 90 min irradiation, 1.5 Gy, 24 h incubation) (Figure 44). The in vivo X-PDT (1.17 Gy/h, 80 kV, 0.5 mA) efficacy of NPS-RB following 1 h irradiation with 10 mg/mL after 24 h incubation was around 90% on HepG2 tumor.

Zhang et al. [112] designed X-ray-luminescent NPs (XLNPs-RB). The XLNPs-RB were composed of a βNaLuF_4_:Tb^3+^ core with ultra-high FRET efficiency and RB covalently coupled to the core. The NPs were functionalized with AEP (2-aminoethyl dihydrogen phosphate) to increase water dispersity (Figure 45).

As RB presented a high luminescence yield, the team evaluated the innovative XLNPs-RB for X-PDT under a lower X-ray dose (0.5 Gy) than other treatment strategies reported by other research teams. The average diameter of XLNPs-RB was 25.6 nm. FRET efficiency was guaranteed by the perfect match between the excitation spectrum of βNaLuF_4_:Tb^3+^ with the absorption spectrum of RB. XLNPs-RB were X-ray irradiated at 80 kV (0.5 mA) in the presence of a DPBF probe to evaluate ^1^O_2_ production. The in vitro cytotoxicity of XLNPs and XLNPs-RB was evaluated on HepG6 cells. After X-ray irradiation for 1.5 h (80 kV, 0.5 mA for 20 min), cell death of 93% was observed at an XLNPs-RB concentration of 1 mg/mL after incubation for 10 h, with insignificant dark cytotoxicity. An in vivo X-PDT study was performed using HepG6 tumor-bearing nude mice injected with 20 mg/mL of XLNPs-RB. Following a single instance of irradiation for 20, 40 or 90 min (0.19, 0.38 and 0.87 Gy each time, respectively), significant tumor growth inhibition was observed, which was similar for the three tested conditions. Following four irradiations of 20, 60 or 90 min, significant tumor growth inhibition was observed, whereas no inhibition was observed in the control groups. Compared to a single instance of irradiation, being subjected to irradiation four times for 60 and 90 min showed better X-PDT efficiency. Nevertheless, with 60 min of irradiation, a significant loss of weight body was observed. In conclusion, successful X-PDT was achieved by performing irradiation four times for 90 min, but a single instance of irradiation of 20 min also gave good results, while requiring minimal X-ray exposure.

Polozhentsev et al. [113] synthesized a GdF_3_:Tb^3+^@RB nanocomposite with 10% chelated lanthanide. These NPs exhibited an orthorhombic structure in the form of a “spindle” with a length of 250 nm and a width of 60 nm. Following X-PDT (36 mGy/4 min), an efficient FRET energy transfer was observed between the lanthanides and the RB adsorbed onto the nanoparticle. In vivo studies were performed regarding biodistribution on intact BALB/c via CT imaging, revealing a significant improvement in the biodistribution contrast of NPs (200 µL of PEG-capped GdF_3_:Tb^3+^ quantified with 20 mg Gd), with 136 mGy after 5 min, 1 h, 2 h and 4 h injection being quantitatively estimated on the basis of CT imaging for the liver and spleen (Figure 46). This study showed enhanced contrast for the liver and spleen, but negative effects should be considered due to the prolonged accumulation times.

Table 9 summarizes the physical and chemical properties and X-PDT conditions presented in all references dealing with nanocomposites@RB.

### 3.4. Other Nanoparticles

#### 3.4.1. Mesoporous LaF_3_:Tb Scintillating Nanoparticles

Tang et al. [114] synthesized scintillating NPs (ScNPs = LaF_3_:Tb). RB was encapsulated in these NPs. DPBF was used to confirm ^1^O_2_ production. No biological experiments were performed.

#### 3.4.2. Nanophosphors

Luminophores have attracted attention from researchers due to their optical properties [115]. They have important properties of organic phosphors, like its high absorption, good photoluminescence quantum yield, fast luminescence decay time and good processability [116].

Ren et al. [117] designed Tb-doped nanophosphors with a unique core–shell–shell (CSS) structure, referred to as NaGdF_4_@NaGdF_4_:Tb@NaYF_4_ (0.85 mol of Gd and 0.15 mol of Tb). The nanoconjugate was successively covalently functionalized with polylysine, RB (namely, CSS-RB) and thiol-RGD peptide (RGD-CSS-RB), with an average size of 120 nm for CSS-RB. DPBF was used as a probe to confirm ^1^O_2_ production. In vitro tests were performed on U87MG cells, showing that cell viability after 24 h incubation with RGD-CSS-RB (200 µg/mL) and X-ray radiation (4 Gy/min) decreased by up to 25%.

#### 3.4.3. Lanthanide@MOF Nanoprobes

Zhao et al. [118] designed a new type of soft X-ray (tube voltage of 10–50 kV)-stimulated nanoprobe via in situ growth of Zr porphyrin-based MOFs on lanthanide (NaYF_4_:Gd,Tb@NaYF_4_ (Gd/Tb: 40/15 (*w*/*w*%)) scintillator NPs (SNP with an estimated size of 30 nm). The nanoprobes exhibited a porous structure, enabling adsorption of RB, as well as porphyrin. In vitro experiments were performed on 4T1 cells following incubation for 24 h with SNPs@Zr-MOF@RB at at different concentrations (0, 50, 100, 200, 500, 1000 μg/mL). In vivo were performed in 4T1 tumor-bearing mice (BALB/c mice). The soft X-ray irradiation was performed at 45 kV for various durations (0, 3, 6, 9, 12, 15 min). A highly significant response was confirmed for the use of deep PDT in tissue and in immunotherapy for cancer cells.

Table 10 summarizes the physical and chemical properties and X-PDT conditions presented in all references dealing with lanthanide@MOF@nanoprobes@RB, nanophosphors@RB and ScNPs@RB.

## 4. Conclusions

RB coupled to NPs is a good approach for PDT applications (Figure 47). RB is a xanthene dye that has interesting photo- and sono-sensitive properties. RB is already used for clinical applications, and an RB formulation for cancer known as PV-10 is currently undergoing clinical trials for use in different types of cancer (melanoma, breast cancer) or infection (clinicaltrials.gov). RB can be used alone, but can also be used as a PS in PDT applications. RB presents several advantages, including the ability to produce ^1^O_2_ upon light illumination and solubility in water. One disadvantage is its absorption spectra, which have an absorption maximum wavelength in water of 550 nm, which does not allow great penetration of light into the tissue. Moreover, RB is not selective for cancer cells.

To improve the system, the use of NPs is an interesting strategy. Indeed, all types of NPs presented in this review were very efficient (UCNPs, Silica NPs, Organic NPs, Gold NPs, Pdots NPs, Nanocapsule, Nanocomplex, Magnetic NPs, pH sensitive NPs, Hybrid NPs, Nanogel, Nanocomposite, Nanophosphor, Lanthanide@MOF Nanoprobe). The use of NPs makes it possible to target RB to the tumor as a result of the passive targeting due to EPR effect. It should be noted that, on the basis of all of the papers reviewed here, the efficiency of RB is higher when RB is coupled with NP, and the best system is that in which RB is covalently coupled to NP, rather than encapsulated. From our point of view, UCNPs present many advantages, since their size is small enough to allow passive targeting, and they can be easily functionalized by a vector, enabling active targeting, and they can be excited by both NIR and X-rays.

Another advantage of RB is that its absorption spectrum matches the emission of lanthanides such as Tb. It is therefore possible to excite Tb using X-ray, and following the energy transfer to RB, to induce the formation of ^1^O_2_. The design in which NPs are coupled with RB and the targeted unit in order to target over-expressed receptors could be a nice option for treating deep tumor or melanoma, for example.

## Figures and Tables

**Figure 1 pharmaceuticals-15-01093-f001:**
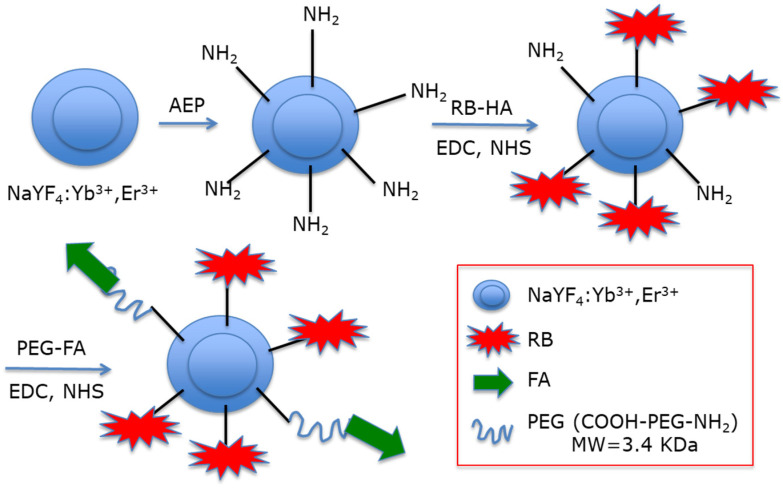
Covalent coupling of NaYF_4_:Yb^3+^ and Er^3+^UCNPs with RB and FA. Adapted from Liu et al. [20].

**Figure 2 pharmaceuticals-15-01093-f002:**
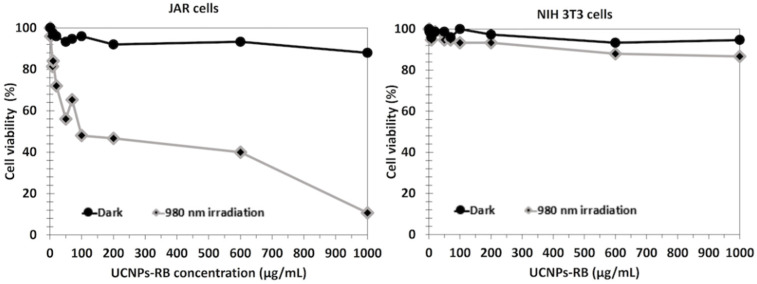
JAR and NIH 3T3 cells viability in dark and after excitation at 980 nm (1.5 W/cm^2^, 10 min) with increasing UCNPs-RB concentration. Adapted from Liu et al. [20].

**Figure 3 pharmaceuticals-15-01093-f003:**
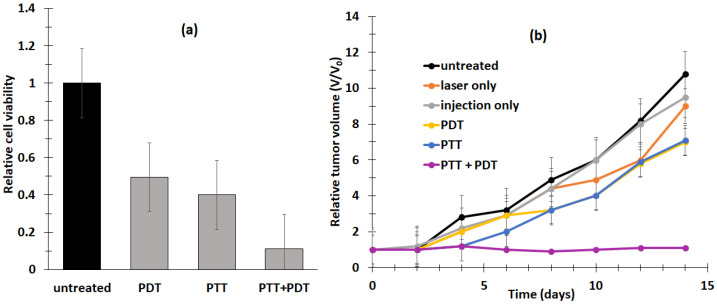
(**a**) In vitro cytotoxicity effect of UCNP@BSA-RB&IR825 for PDT, PTT and combined PTT–PDT treatment. PDT (808 nm, 0.5 W/cm^2^) and PTT (980 nm, 0.4 W/cm^2^). (**b**) In vivo tumor growth in different groups of mice after various treatments, as indicated: 1: untreated; 2: laser only (808 nm and 980 nm); 3: injection only; 4: PDT; 5: PTT, 6: PTT + PDT. Wavelengths of 808 nm (0.5 W/cm^2^, 5 min) and 980 nm (0.4 W/cm^2^, 30 min) were used to separately trigger PTT and PDT, respectively. Adapted from Chen et al. [21].

**Figure 4 pharmaceuticals-15-01093-f004:**
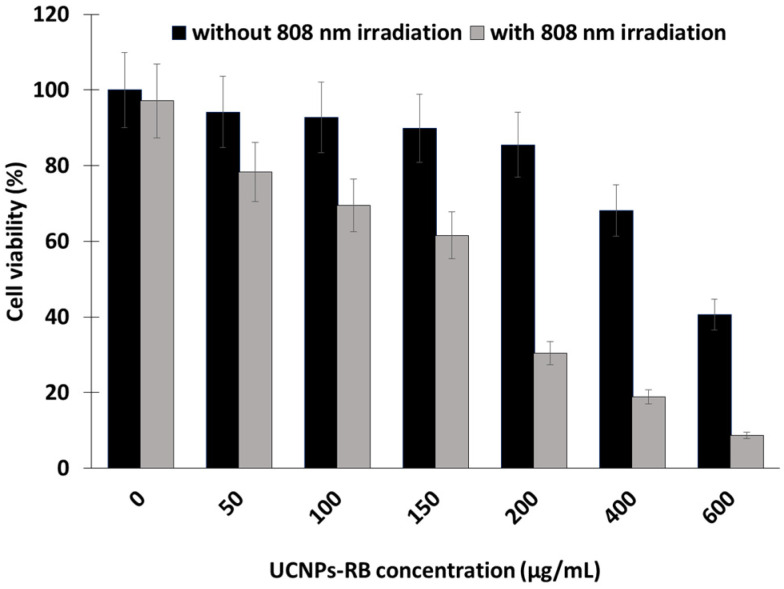
HeLa cell viability when treated with different concentrations of UCNPs-RB with and without irradiation (808 nm, 0.67 W/cm^2^, 10 min). Adapted from Wang et al. [23].

**Figure 5 pharmaceuticals-15-01093-f005:**
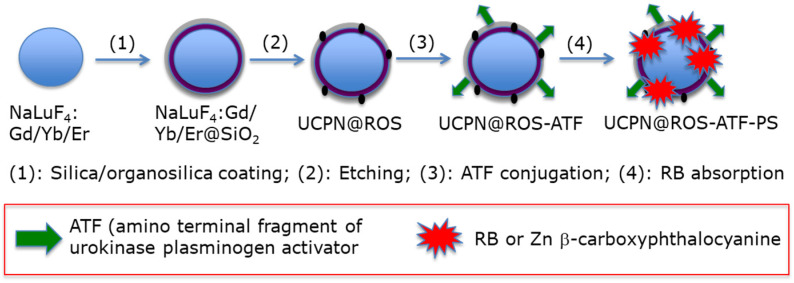
Preparation of multifunctional UCNPs. Adapted from Lu et al. [24].

**Figure 6 pharmaceuticals-15-01093-f006:**
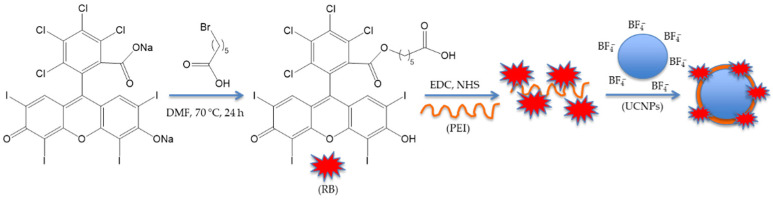
Covalent coupling of RB onto Er@Nd UCNPs. Adapted from Li et al. [25].

**Figure 7 pharmaceuticals-15-01093-f007:**
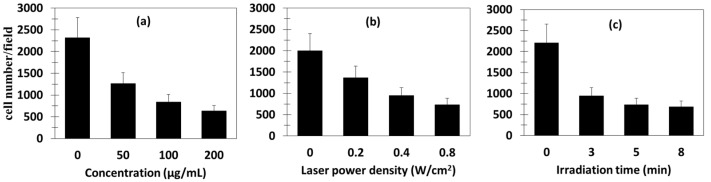
In vitro PDT of Er@Nd-RB. Cell viability of 4T1 cells with different (**a**) concentrations, (**b**) laser power densities, and (**c**) irradiation times. Adapted from Li et al. [25].

**Figure 8 pharmaceuticals-15-01093-f008:**
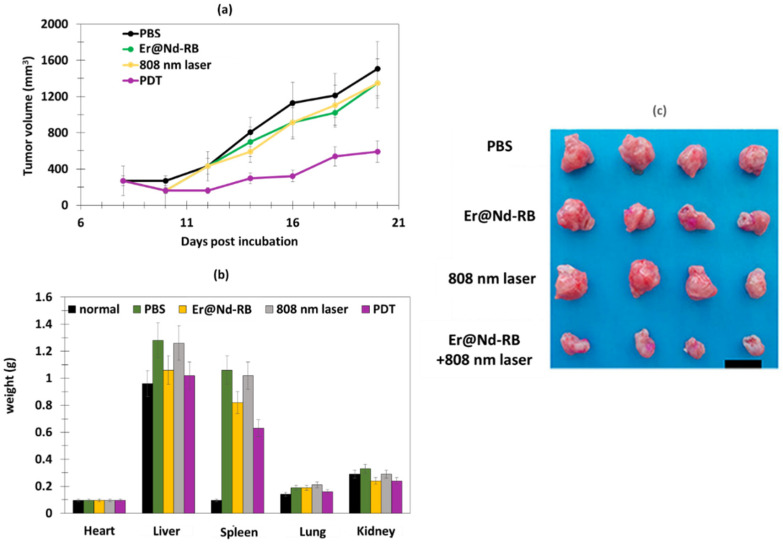
PDT on 4T1 breast murine tumor after in vivo experiment (808 nm, 520 mW/cm^2^). (**a**) Tumor volume growth as a function of number of days of treatment, and (**b**) weight of heart, liver, spleen, lung and kidney for the four mice groups. (**c**) Digital photos of tumors for the four groups of mice. The mice were sacrified 14 days after treatment (808 nm, 520 mW/cm^2^). Figures (**a**,**b**) adapted from Li et al. [25]. Figure (**c**) reprinted with permission from Li et al. [25]. Copyright 2016 American Chemical Society.

**Figure 9 pharmaceuticals-15-01093-f009:**
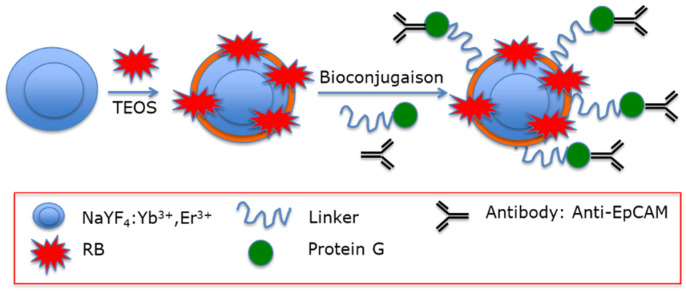
Covalent coupling of RB and antibody through a bifunctional fusion protein G. Adapted from Liang et al. [26].

**Figure 10 pharmaceuticals-15-01093-f010:**
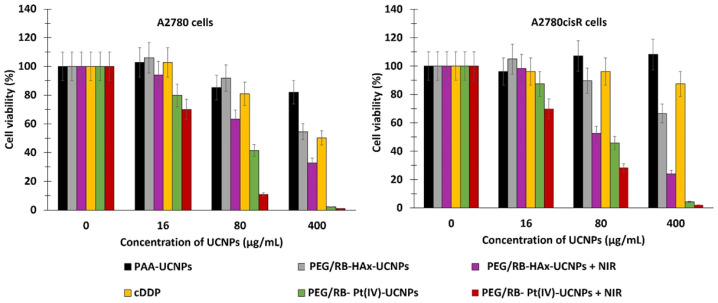
Cell viabilities of A2780 and A2780cisR cells treated with various concentrations of PEG/RB-Pt(IV)-UCNPs, PEG/RB-HA-UCNPs, PAA-UCNPs and cDDP after NIR irradiation (808 nm, 6 W/cm^2^, 5 min). Poly(acrylic acid) = PAA, Hexamethylenediamine = HAx, cisplatine or cis-diaminedichloroplatine(II) = cDDP. Adapted from Ai et al. [27].

**Figure 11 pharmaceuticals-15-01093-f011:**
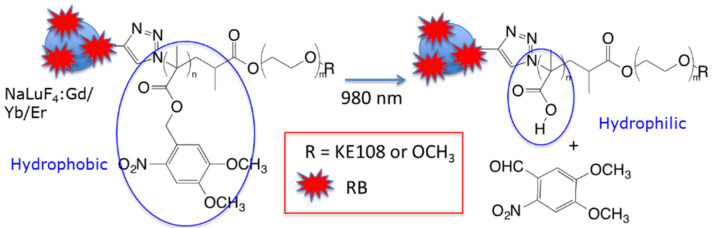
Illustration of NIR-triggered hydrophobic-to-hydrophilic transition leading to the release of hydrophobic drug. Adapted from Chen et al. [28].

**Figure 12 pharmaceuticals-15-01093-f012:**
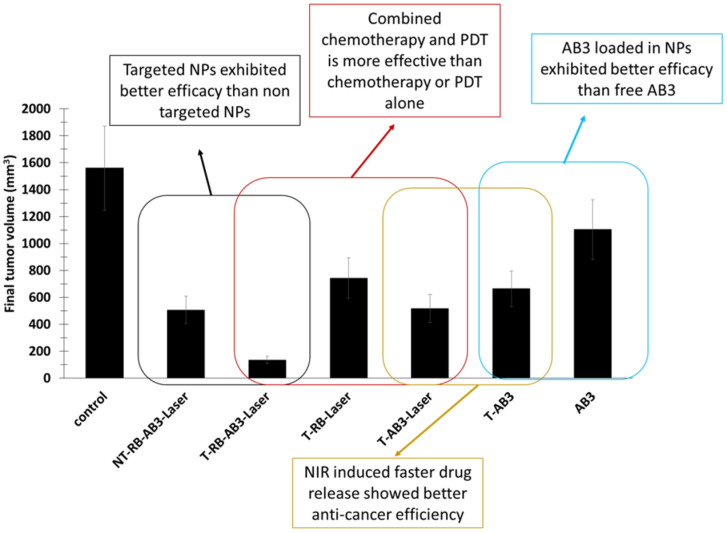
Final tumor volumes, demonstrating the in vivo anti-cancer PDT efficacy of different formulations of UCNP-based theranostic micelles in NET xenografts. Tumor-bearing mice were treated in seven groups: (1) saline (control); (2) AB3 (conventional chemotherapy); (3) T-RB-Laser (targeted NIR-controlled PDT); (4) T-AB3 (targeted chemotherapy); (5) T-AB3-Laser (targeted NIR-controlled chemotherapy); (6) NT-AB3-RB (nontargeted chemotherapy; no PDT effect as no laser illumination); (7) NT-AB3-RB-Laser (nontargeted combination NIR-controlled chemotherapy and PDT). Each mouse received two intravenous injections (30 mg/kg of AB3) over a 7-day interval. A continuous wave fiber-coupled 980 nm laser (0.5 W/cm^2^, 15 min, 1 min interval after every 5 min of irradiation) was applied at the tumor sites in the “Laser” groups 4 h post injection. Adapted from Chen et al. [28].

**Figure 13 pharmaceuticals-15-01093-f013:**
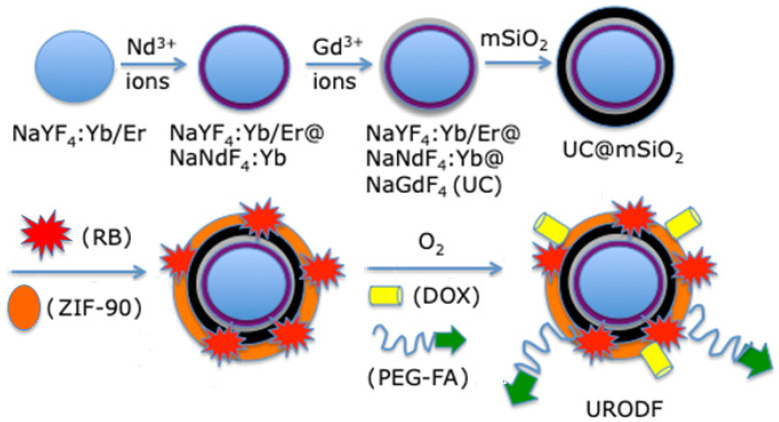
Preparation of URODF NPs. Adapted from Xie et al. [34].

**Figure 14 pharmaceuticals-15-01093-f014:**
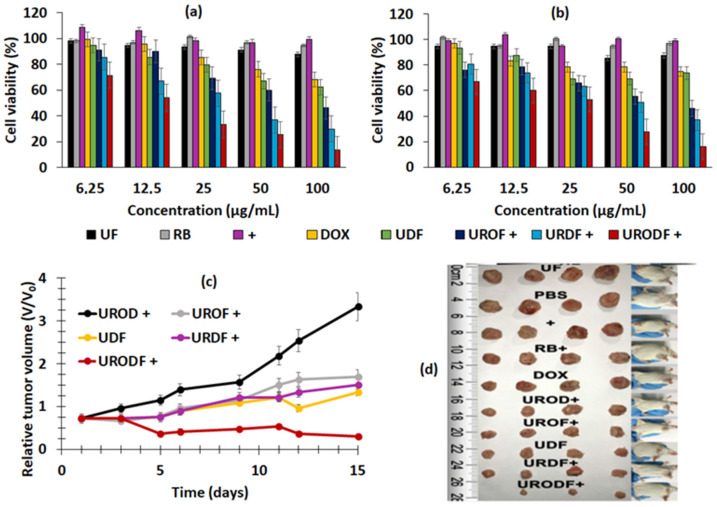
MTT assays for cell viability of (**a**) 4T1 and (**b**) HeLa cells incubated with different products and concentrations for 24 h. (**c**) Tumor volume growth curves and (**d**) dynamic body change of mice in groups receiving UROD+, UROF+, UDF, URDF+ and URODF+ treatments over a period of 14 days (+ = laser irradiation (808 nm, 0.5 W/cm^2^, 0 to 150 s)). Figures (**a**–**c**) adapted from Xie et al. [34]. Figure (**d**) reprinted with permission from Xie et al. [34]. Copyright 2019 American Chemical Society.

**Figure 15 pharmaceuticals-15-01093-f015:**
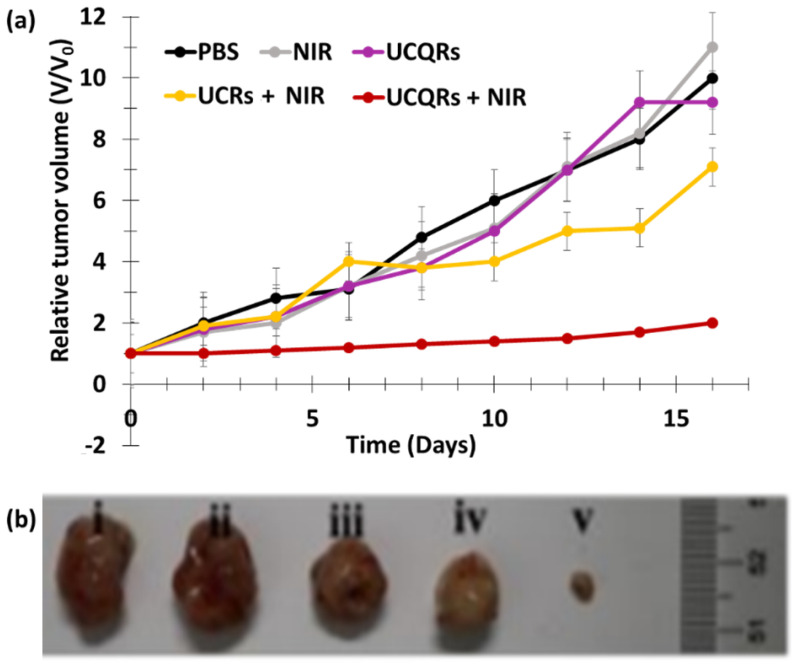
(**a**) Relative tumor volume obtained from mice with various treatments. (**b**) Representative photographs of the corresponding excised tumors (i—PBS, ii—NIR, iii—UCQRs (10 mg/mL, 40 µL), iv—UCRs + NIR (10 mg/mL, 40 µL), and v—UCQRs + NIR) (10 mg/mL, 40 µL). NIR irradiation (808 nm, 1.6 W/cm^2^, 10 min). Figure (**a**) adapted from Song et al. [37]. Figure (**b**) reprinted from Song et al. [37]. Copyright 2019 American Chemical Society.

**Figure 16 pharmaceuticals-15-01093-f016:**
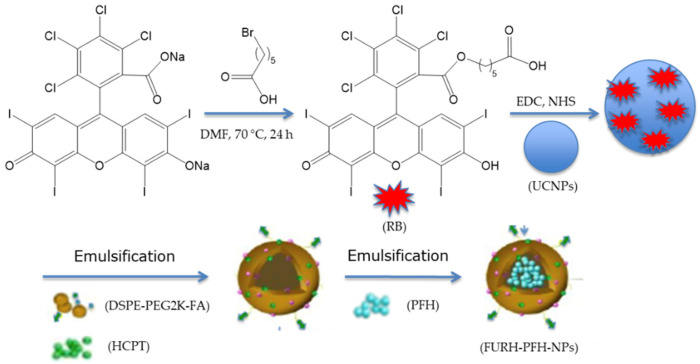
Preparation of FURH-PFH-NPs. Adapted from Wang et al. [39].

**Figure 17 pharmaceuticals-15-01093-f017:**
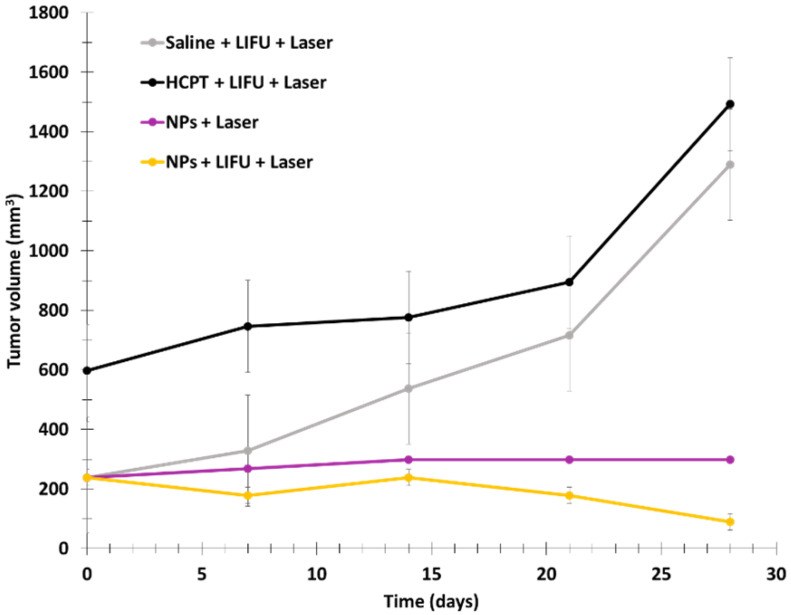
Tumor growth of four groups of SKOV3 tumor-bearing mice treated with different conditions. Laser irradiation (980 nm, 2 W/cm^2^, 2 min). Adapted from Wang et al. [39].

**Figure 18 pharmaceuticals-15-01093-f018:**
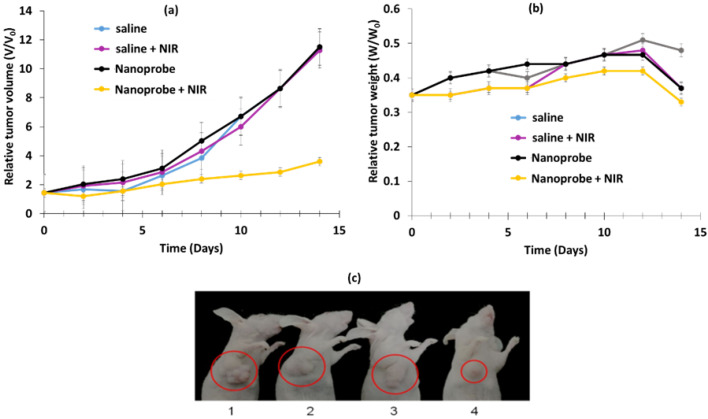
Four groups of HeLa tumor-bearing BALB/c nude mice. (**a**,**b**) Relative tumor volume, (**c**) relative tumor weight when treated with (1) saline, (2) saline + NIR, (3) nanoprobe, (4) nanoprobe + NIR. NIR laser irradiation (808 nm, 1.0 W/cm^2^, 40 min). The nanoprobe was MUCNPs/800WC/RB/Cy3/pep-QSY7. [Nanoprobe] = 300 µg/mL. Figures (**a**,**b**) adapted from Li et al. [40]. Figure (**c**) reprinted with permission from Li et al. [40]. Copyright 2020 American Chemical Society.

**Figure 19 pharmaceuticals-15-01093-f019:**
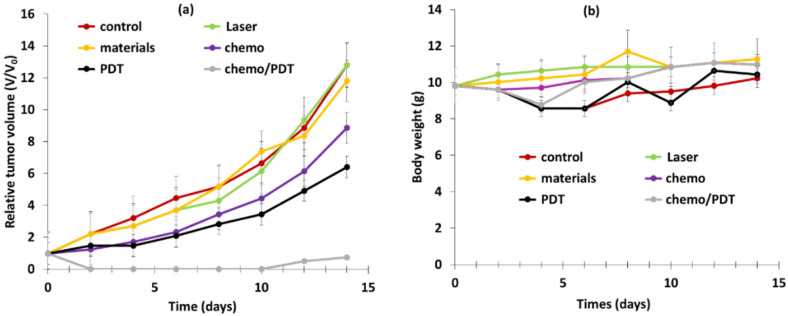
(**a**) Relative tumor volume and (**b**) body relative weight of 6 groups of HeLa tumor-bearing mice under various conditions: control, laser (808 nm irradiation and 0.5 W/cm^2^ for 10 min), DOX-RB-NH2-UCMS-BA-CD chemotherapy, PDT, chemotherapy + PDT. Adapted from Han et al. [41].

**Figure 20 pharmaceuticals-15-01093-f020:**
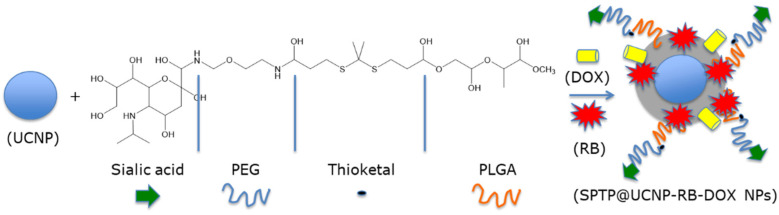
Illustration of the synthesis of the SPTP@UCNP-RB-DOX NPs. Adapted from Jin et al. [42].

**Figure 21 pharmaceuticals-15-01093-f021:**
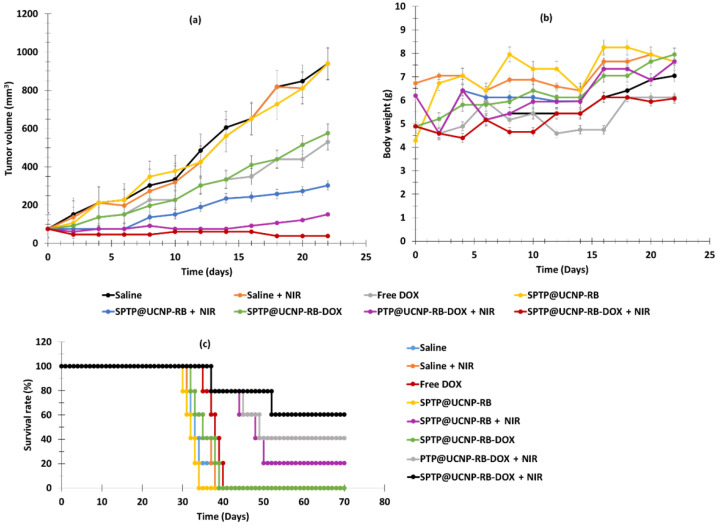
In vivo antitumor effect. (**a**) Tumor volume for different treatment groups (*n* = 6). (**b**) Change in body weight of mice over 23 days. (**c**) Changes in survival rate in different groups. Light irradiation at a wavelength of 980 nm, 0.5 W/cm^2^, for 5–10 min (SPTP: 50 μg/mL, DOX: 2.5 μg/mL and RB: 2.5 μg/mL). Adapted from Jin et al. [42].

**Figure 22 pharmaceuticals-15-01093-f022:**
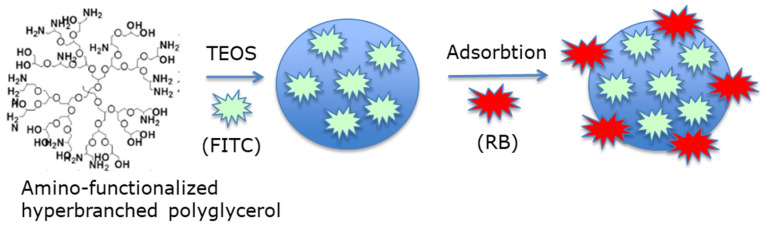
Preparation of polyglycerol MSNs with adsorbed fluorescein isothiocyanate (FITC) and RB. Adapted from Liu et al. [53].

**Figure 23 pharmaceuticals-15-01093-f023:**
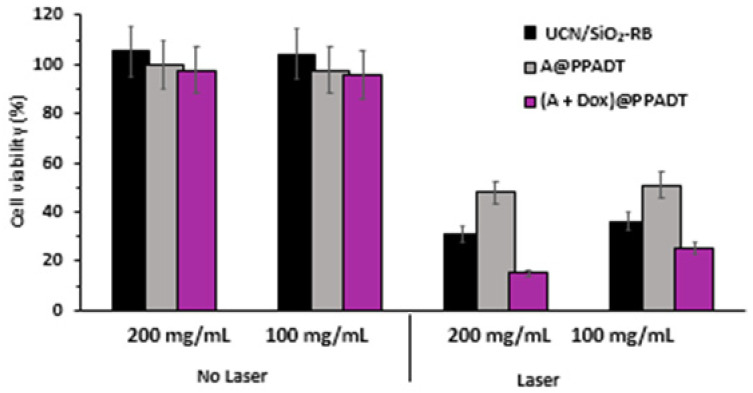
Viability of HeLa cells incubated with UCN/SiO_2_-RB, A@PPADT or (A + DOX)@PPADT) (100 and 200 µg/mL) with or without laser irradiation (980 nm, 1.0 W/cm^2^, 20 min). Adapted from Zhou et al. [54].

**Figure 24 pharmaceuticals-15-01093-f024:**
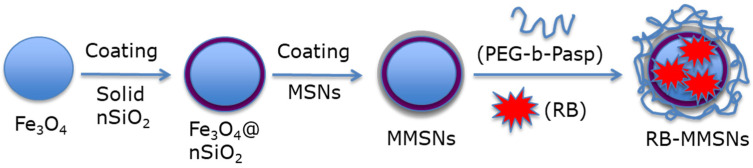
Preparation of the synthesis of the core–shell RB-MMSNs. Adapted from Zhan et al. [56].

**Figure 25 pharmaceuticals-15-01093-f025:**
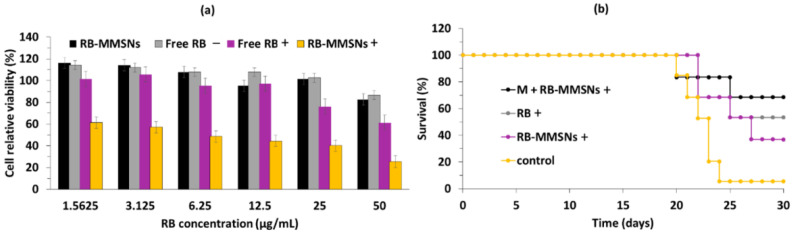
(**a**) Relative cell viability of B16 cells treated with various concentrations of MMSNs, RB–MMSNs, and RB with (+) or without (−) green light irradiation (532 nm, 25 mW/cm^2^, 3 min). (**b**) Survival curves of groups with different treatments (532 nm, 25 mW/cm^2^, 3 min) (control, RB+, RB–MMSNs+ and M + RB–MMSNs+). Adapted from Zhan et al. [56].

**Figure 26 pharmaceuticals-15-01093-f026:**
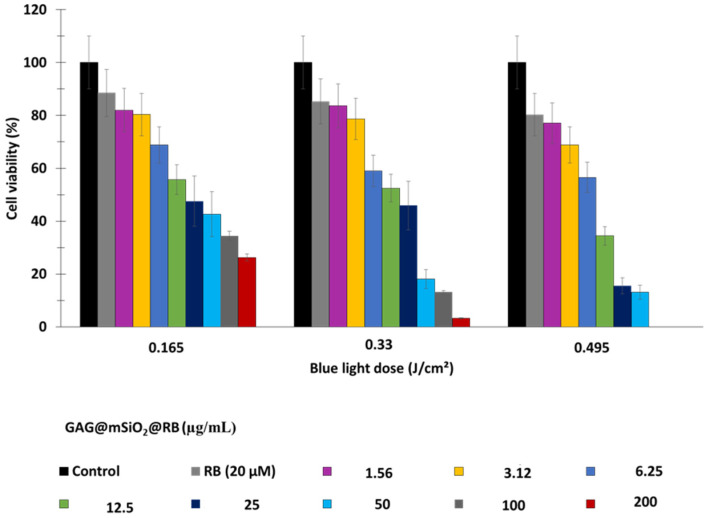
Cell viability of MDA-MB-231 cells upon exposure to irradiation (470 nm, 20 mW/cm^2^) for 15 min, 30 min and 45 min corresponding to energy densities of 0.165, 0.33, and 0.495 J/cm^2^, respectively. Adapted from Jain et al. [60].

**Figure 27 pharmaceuticals-15-01093-f027:**
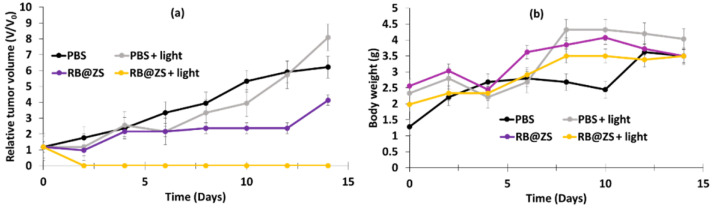
(**a**) Relative tumor volume and (**b**) body weight of the tumor-bearing mice in different treatment groups (*n* = 6) during post-treatment with PBS or RB@ZS aqueous solution (30 mg/mL). Laser irradiation at 532 nm (500 mW/cm^2^) for 10 min. Adapted from Hu et al. [61].

**Figure 28 pharmaceuticals-15-01093-f028:**
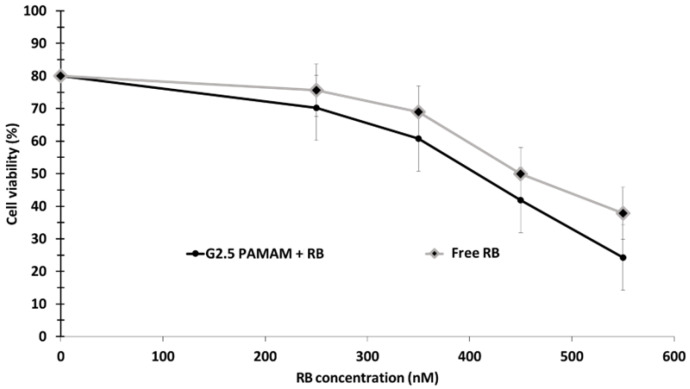
DLA cell viability following incubation with G2.5 PAMAM + RB and free RB upon exposure to a 150 W xenon lamp (540 nm, 10 min). Adapted from Karthikeyan et al. [69].

**Figure 29 pharmaceuticals-15-01093-f029:**
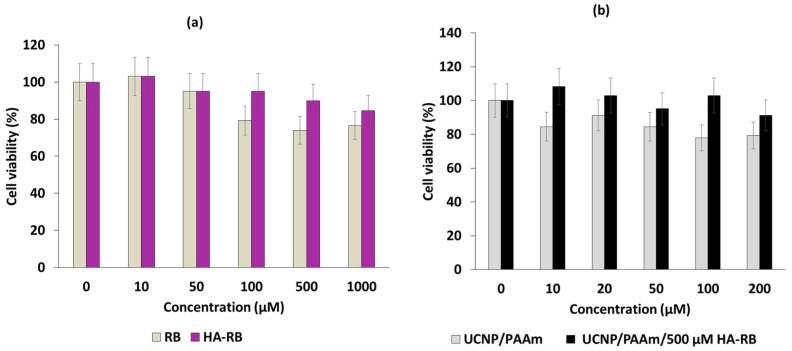
NIH 3T3 cell viability following incubation for 24 h with various concentrations of (**a**) RB and HA-RB and (**b**) UCNP/PAH and UCNP/HA-RB after exposure to laser irradiation (980 nm, 1.5 mW/cm^2^, 30 min). Adapted from Han et al. [72].

**Figure 30 pharmaceuticals-15-01093-f030:**
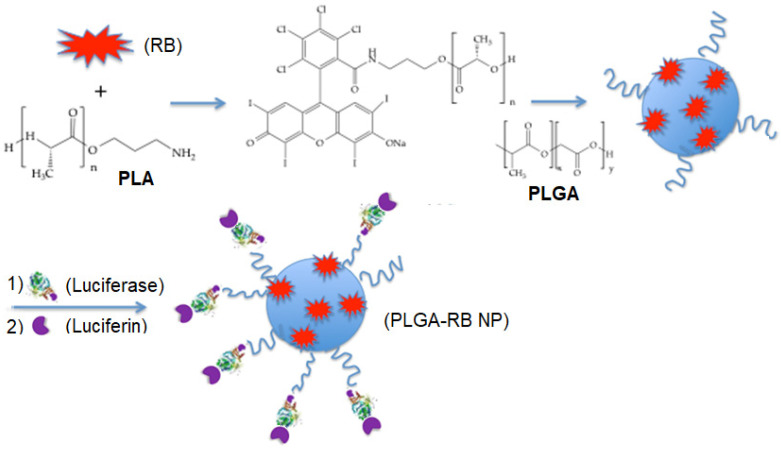
Preparation of PLGA-RB NPs for BRET-PDT. Adapted from Yang et al. [73].

**Figure 31 pharmaceuticals-15-01093-f031:**
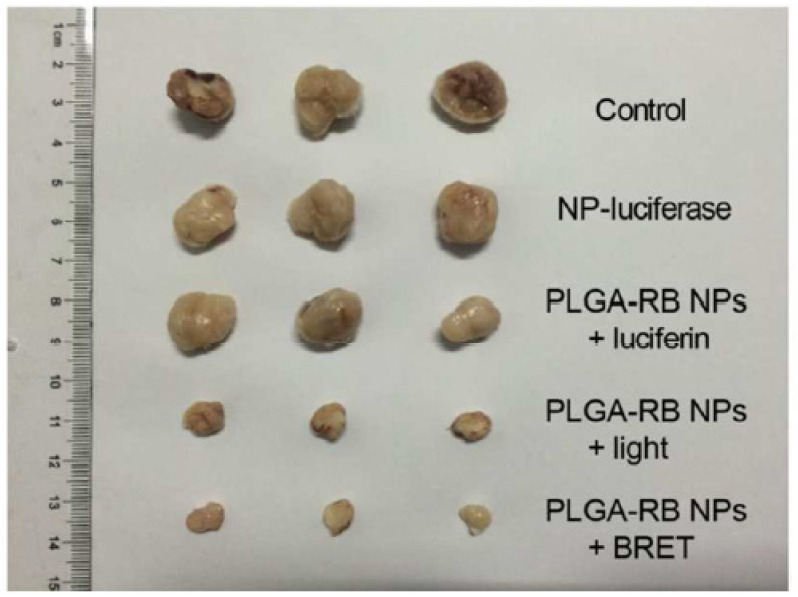
Photographs of tumors excised from the BRET-PDT treatment group and other groups on day 14 following laser irrdiation (520 nm, 200 mW/cm^2^, 30 min). Reprinted with the permission from Yang et al. [73]. Copyright 2018 American Chemical Society.

**Figure 32 pharmaceuticals-15-01093-f032:**
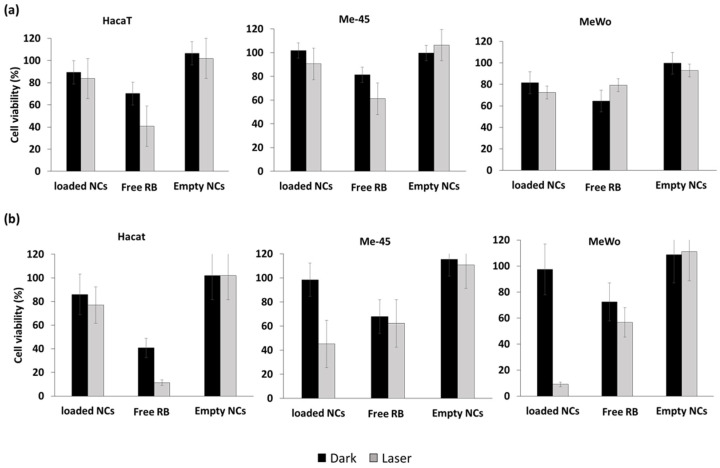
Cell viability of Me-45, MeWo, and HaCaT in the dark or upon exposure to laser irradiation (980 nm, 10 J/cm^2^, 6.2 W/cm^2^, 5 min) after (**a**) 24 h and (**b**) 48 h. Adapted from Bazylińska et al. [75].

**Figure 33 pharmaceuticals-15-01093-f033:**
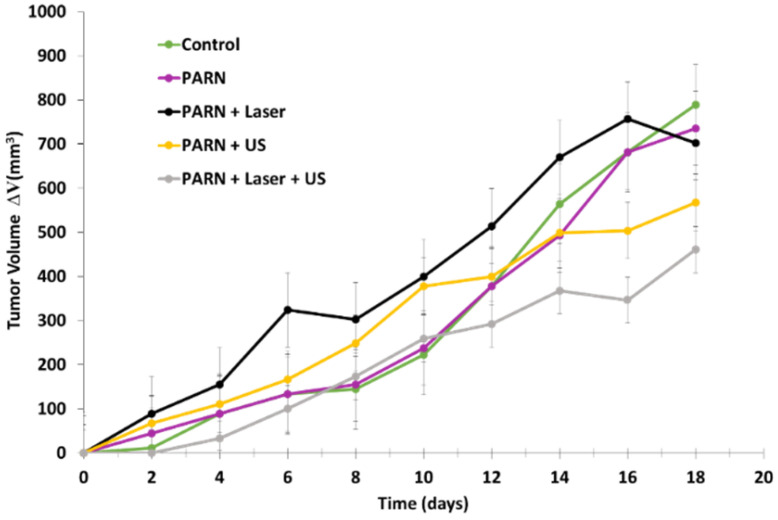
Variations in tumor volume in HeLa-xenografted nude mice under various treatment conditions. Laser irradiation (808 nm, 1 W/cm^2^, 3 min), US (frequency: 1.0 MHz; duty cycle: 50%; power density: 1.0 W/cm^2^). Adapted from Liu et al. [77].

**Figure 34 pharmaceuticals-15-01093-f034:**
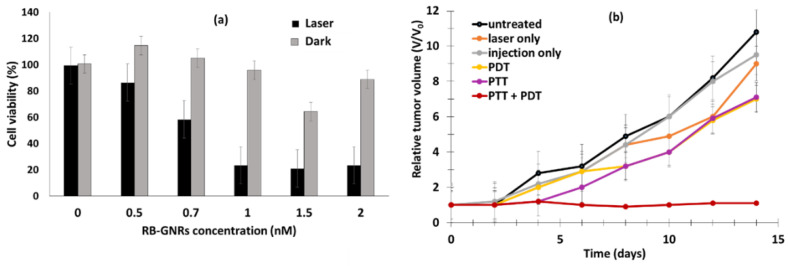
(**a**) CaI-27 cell viability in the dark or upon exposure to green laser light irradiation (532 nm, 1.76. W/cm^2^, 10 min). (**b**) Relative tumor volume under different treatment conditions (PDT: 532 nm, 1.76 W/cm^2^, 10 min; PTT: 810 nm 8.16 W/cm^2^, 5 min). Adapted from Wang et al. [83].

**Figure 35 pharmaceuticals-15-01093-f035:**
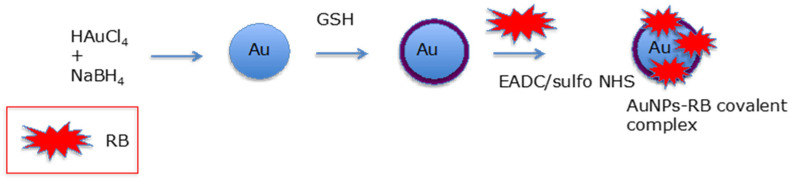
Preparation of Au NPs covalently coupled to RB. Adapted from Prasanna et al. [84].

**Figure 36 pharmaceuticals-15-01093-f036:**
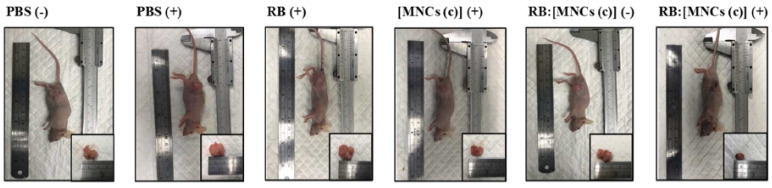
Representative digital images of mouse tumor treated with PBS, RB, MNCs and RB:MNCs (124 μg and 0.25 mg per mouse of RB and Fe_3_O_4_ NPs, respectively) with (+) and without (−) green laser light irradiation (532 nm, 100 mW/cm^2^, 5 min) for mice after 38 days under different treatment conditions. Reprinted with permission from Yeh et al. [97]. Copyright 2018 American Chemical Society.

**Figure 37 pharmaceuticals-15-01093-f037:**
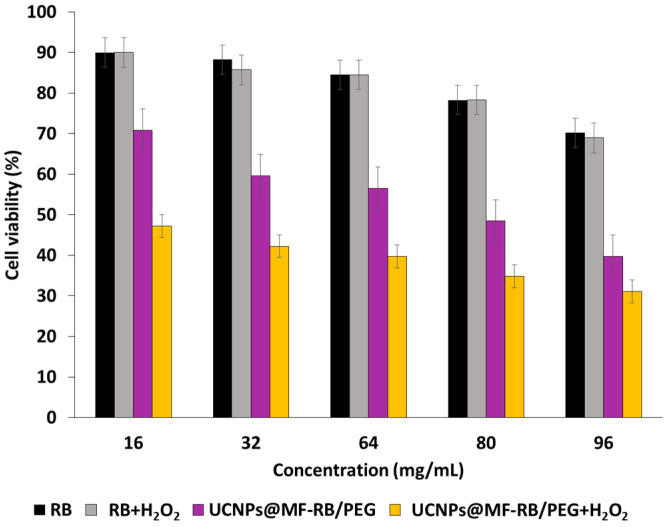
Cell viability of HeLa cells incubated with various samples under laser irradiation (980 nm, 0.5 mW/cm^2^, 15 min). Adapted from Su et al. [101].

**Figure 38 pharmaceuticals-15-01093-f038:**
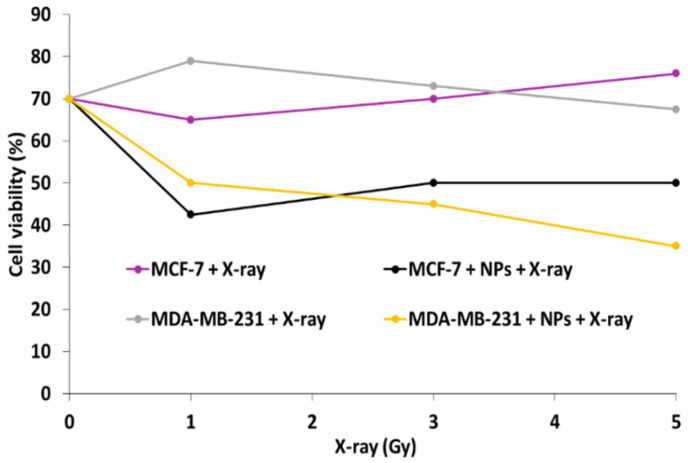
MDA-MB-231 and MCF-7 cells incubated (50 µg/mL) with core−shell−shell ScNP-PAH-RB-PEG-FA (NPs) for 24 h followed by X-ray radiation with doses of 1, 3, 5 Gy of. Adapted from Hsu et al. [105].

**Figure 39 pharmaceuticals-15-01093-f039:**
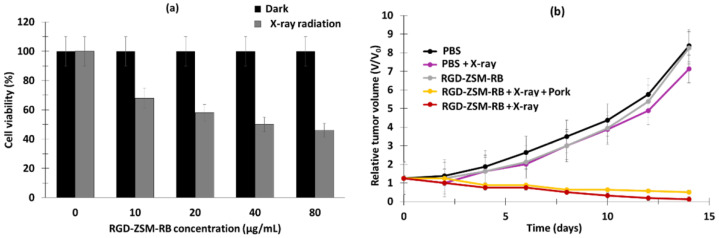
(**a**) In vitro evaluation of X-PDT. Cell viability of U87MG cells incubated with different concentrations of RGD-ZSM-RB with or without X-ray irradiation (50 kV, 1 Gy, 10 min). (**b**) In vivo evaluation of X-PDT. Relative tumor volume obtained from mice under different treatment conditions (X-ray irradiation (1.0 Gy, RGD-ZSM-RB: 20 mg/kg) at different times (1, 2, 4, 6, 8 and 12 h) after injection). Adapted from Sun et al. [106].

**Figure 40 pharmaceuticals-15-01093-f040:**
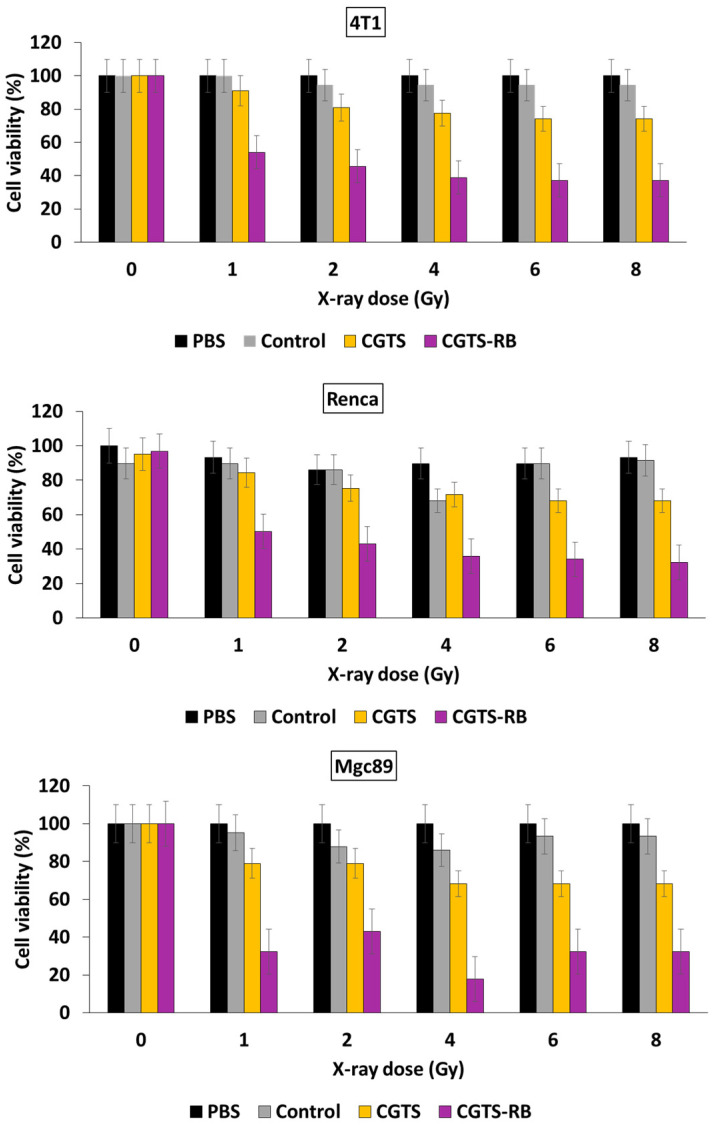
Cell viability of 4T1, Renca, and Mgc89 cells incubated for 24 h with PBS, CGTS and CGTS-RB (50 μg/mL) under different X-ray doses (1 Gy/min). Adapted from Ahmad et al. [107].

**Figure 41 pharmaceuticals-15-01093-f041:**
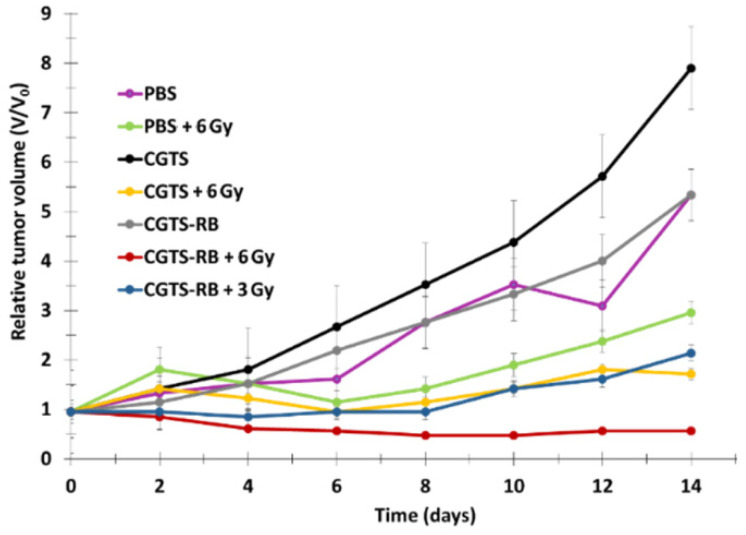
Relative tumor volume under different treatment conditions (2 mg/mL, 200 μL, X-ray irradiation (160 kV, 25 μA, 1 Gy/min)). Adapted from Ahmad et al. [107].

**Figure 42 pharmaceuticals-15-01093-f042:**
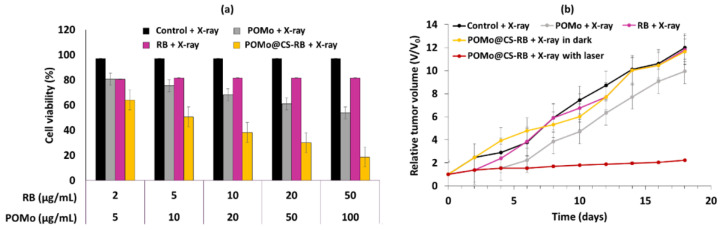
(**a**) In vitro cell viability of 4T1 cells incubated for 24 h with various concentrations of RB, POMo and POMo@CS-RB under X-ray irradiation (2 Gy). (**b**) Relative tumor volume under different treatment conditions (X-ray irradiation (2 Gy), 25 mg/Kg). Adapted from Maiti et al. [109].

**Figure 43 pharmaceuticals-15-01093-f043:**
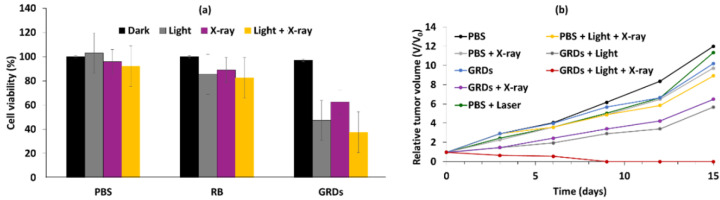
(**a**) Cell viability of 4T1 cells incubated for 24 h with PBS, RB and GRDs (4.0 μg/mL) in the dark or following laser and/or X-ray irradiation (laser: 532 nm, 30 mW/cm^2^, 5 min; and X-ray: 1 Gy). (**b**) Relative tumor volume under different treatment conditions (10 mg/kg). (Laser: 532 nm, 140 mW/cm^2^, 15 min; X-ray: 1Gy). Adapted from Sun et al. [110].

**Figure 44 pharmaceuticals-15-01093-f044:**
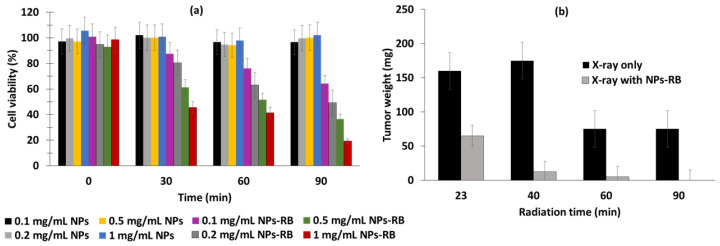
Antitumor effect of NPs and NPs-RB on nude mice bearing HepG2 tumor cells subcutaneously. NPs or NPs-RB were injected intratumorally. (**a**) HepG2 cell viability after 90 min irradiation, 1.5 Gy, 24 h incubation. (**b**) Tumor weight with different irradiation times. Dissected tumor tissues were obtained from nude mice incubated for 24 h with 10 mg/mL NPs-RB. The X-ray tube was set at 80 kV, 0.5 mA and 1.17 Gy/h. Adapted from Zhang et al. [111].

**Figure 45 pharmaceuticals-15-01093-f045:**
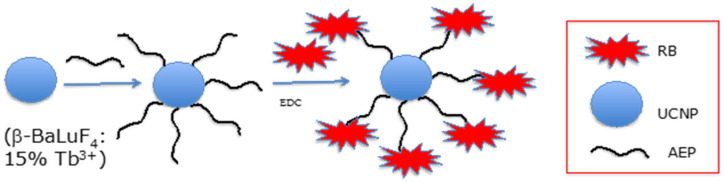
Preparation of XLNPs-RB. Adapted from Zhang et al. [112].

**Figure 46 pharmaceuticals-15-01093-f046:**
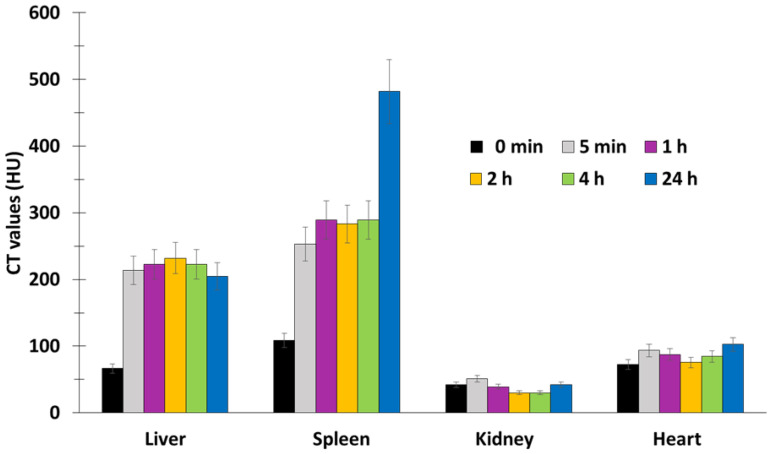
CT values in the liver, spleen, kidney, and heart on BALB/c 5 min post injection of NPs (200 µL of PEG-capped GdF_3_:Tb^3+^ quantified with 20 mg Gd) with different radiation times (X-ray; 136 mGy/4 min). Adapted from Polozhentsev et al. [113].

**Figure 47 pharmaceuticals-15-01093-f047:**
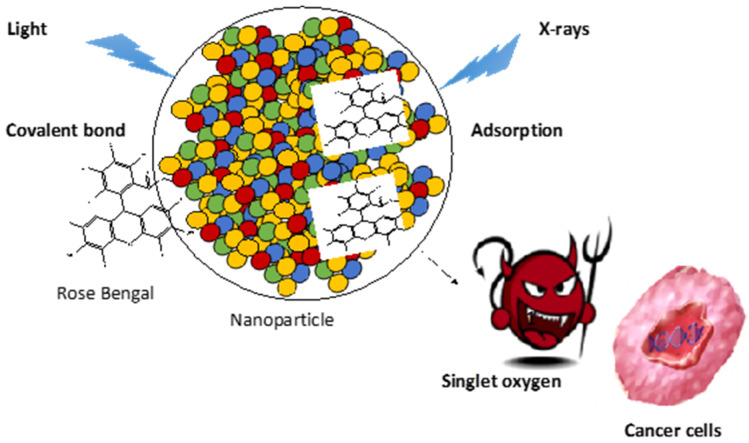
Illustration of nanoparticle coupled to RB for PDT and PDT-X.

**Table 1 pharmaceuticals-15-01093-t001:** Table summarizing all references dealing with UCNP@RB, including type of NPs, along with their size, excitation wavelength, type of detection of ^1^O_2_, type of coupling between the NPs and the RB, and an indication of the type of biological tests performed.

PDT UCNPs									
References	Type of NPs	Size of NPs (nm)	Irradiation Conditions	Type of ^1^O_2_ Detection	Type of Coupling NP-RB	Biological Tests/In Vitro		Biological Test/In Vivo	
						Cells Used	Results	Mice Used	Results
[20]	UCNPs@RB@FA-(NaYF_4_:Yb^3+^, Er^3+^)	20 without RB	980 nm, 1.5 W/cm^2^, 10 min	DPBF	Covalent	JAR NIH 3T3	Cytotoxicity (+)	nd	nd
[25]	UCNPs-RB	25 without RB	In vitro: 0.8 W/cm^2^, 8 min In vivo: 808 nm, 520 mW/cm^2^	DPBF	Covalent	4T1	Cytotoxicity (+)	4T1 breast cancer bearing mice	PDT (+)
[21]	UCNP@BSA-RB&IR825	60	980 nm, 0.4 W/cm^2^ in vitro: 10 min in vivo: 30 min	SOSG	Encapsulation	4T1	Cytotoxicity (+)	4T1 tumor-bearing mice	PDT (+)
[43]	Er@Nd-RB Upconversion	216	808 nm laser (in vitro: 1 W/cm^2^, in vivo: 1.25 W/cm^2^), 5 min	DCFH-DA	Encapsulation	4T1	Cytotoxicity (+)	4T1 tumor-bearing mice- Healthy BALB/c	PDT (+)
[37]	UCNP@BSA-RB&IR825	16	In vitro: 808 nm, 0.75 W/cm^2^, 5 min In vivo: 808 nm, 0.95 W/cm^2^, 15 min	ABDA	Encapsulation	HeLa HCT116	Cytotoxicity (+)	4T1 tumor-bearing mice	PDT (+)
[29]	UCNPs-RB&ZnPc	24–32	980 nm, 0.25 W/cm^2^, 15 min	DPBF	Encapsulation	A549	Cytotoxicity (+)	Hepa1-6 tumor-bearing mice	PDT (+)
[30]	Nd^3+^/Yb^3^ UCNPs with Ag_2_Se QD	120	980 nm, 2.0 W/cm^2^, 10 min	ABDA	Covalent	HeLa	Cytotoxicity (+)	nd	nd
[28]	NaYF_4_: 20%Yb^3+^, 2% Er3+@ NaYF_4_	29	980 nm, 0.5 W/cm^2^, 10 min	DPBF	Covalent	NET	Cytotoxicity (+)	TT-tumor-bearing mice	PDT (+)
[27]	UCN@mSiO_2_-(Azo + RB)	35.9	808 nm, 6 W/cm^2^, 5 min	DPBF	Covalent	A2780 A2780cisR	Cytotoxicity (+)	nd	nd
[26]	PNBMA-PEG@ RB@AB3@KE108	29	980 nm, 1.5 W/cm^2^, 30 min	DPBF	Adsorption	HT-29	Cytotoxicity (+)	Mouse anti-human CD326 (EpCAM) (clone HEA-125)	PDT (+)
[22]	RB-UPPLVs	40	980 nm, 2.5 W/cm^2^, 20 min	ABDA	Encapsulation	HeLa	Cytotoxicity (+)	nd	nd
[23]	NaYF_4_:Yb/Ho@NaYF_4_:Nd@NaYF_4_	0.4 to 4.2	808 nm, 0.67 W/cm^2^, 10 min	DPBF	Covalent	HeLa	Cytotoxicity (+)	nd	nd
[42]	SPTP@UCNP-RB-DOX	54.3	980 nm, 0.5 W/cm^2^, 5–10 min	DPBF	Adsorption	4T1	Cytotoxicity (+)	TNBC	PDT (+)
[35]	CSGUR-MSGG/5FU@RB@FA	72.3	980 nm, 30 min	DPBF	Covalent	HT-29	Cytotoxicity (+)	nd	nd
[40]	STPT@UCNP-RB-DOX	56.1 ± 1.0	808 nm, 1.5 W/cm^2^, 40 min	DPBF	Covalent	HeLa MCF-7 NIH 3T3	Cytotoxicity (+)	BALB/c nude mice with HeLa tumor	PDT (+)
[39]	FA/UCNPs-RB/HCPT/PFH@Lipid	223.6 ± 68.8	980 nm, 0.2 W/cm^2^, 2 min	DPBF	Covalent	SKOV-3	Cytotoxicity (+)	SKOV3 tumor-bearing mice	PDT (+)
[41]	NaYF_4_:Gd@NaYF_4_:Er,Yb@NaYF_4_:Nd,Yb NPs @ RB, Cy3 and pep-QSY7	130	808 nm, 0.5 W/cm^2^, 10 min	ABDA	Covalent	HeLa	Cytotoxicity (+)	HeLa tumor-bearing mice	PDT (+)
[37]	FA/UCNPs-RB/HCPT/PFH@Lipid	62.57 without RB	808 nm, 1.6 W/cm^2^, 10 min	ABDA	Encapsulation	HeLa	Cytotoxicity (+)	Lewis lung carcinoma (LLC) tumor-bearing mice	PDT (+)
[36]	BT@PAH/RB/PAH, BT–RB	70	532 nm, 30 mW/cm^2^, 3 min	DPBF	Loading	HeLa	Cytotoxicity (+)	nd	nd
[34]	Nd^3+^/Yb^3^ UCNPs) with Ag2Se QDs, RB	150	808 nm, 0.5 W/cm^2^, 10 min	DPBF	Encapsulation	HeLa	Cytotoxicity (+)	tumor-bearing BALB/c mice	PDT (+)
[33]	Barium titanate NPs	23.7 without RB	980 nm, 200 mW/cm^2^, 6 min	ABDA	Covalent	SK-BR-3	Cytotoxicity (+)	nd	nd
[32]	NaGdF_4_:Yb^3+^/Er^3+^@BSA@RB	78	980 nm, 13 W/cm^2^, 10 min	DPBF	Covalent	A549	Cytotoxicity (+)	nd	nd
[31]	NaYF_4_:Yb,Er,Gd@NaYF_4_@ SiO_2_-RB	91	980 nm, 2.5 W/cm^2^, 20 min	DPBF	Adsorption	HeLa	Cytotoxicity (+)	nd	nd
[44]	NaGdF_4_:Yb^3+^/Er^3+^ @ RB	41.1	980 nm, 2 W/cm^2^, 10 min	DPBF	Covalent	4T1	Cytotoxicity (+)	nd	nd
[45]	PEG-block-Polymer/C18RB/NaYF_4_:Yb^3+^/Er^3+^	8.7	980 nm, 2W/cm^2^, 30 min	DPBF	Encapsulate	Colon-26	Cytotoxicity (+)	nd	nd
[32]	NaYF_4_:2%Er,20%Yb)@SiO_2_-RB	78	980 nm, 13 W/cm^2^, 10 min	DPBF	Covalent	A549	Cytotoxicity (+)	nd	nd
[24]	UCNP@OS@RB	27	980 nm, 0.5 W/cm^2^, 10 min	DPBF	Adsorption	HT1299	Affinity and Cytotoxicity (+)	nd	nd

nd: not determined; cytotoxicity (+): induced tumor cell death in vitro; PDT (+): induced tumor death in vivo.

**Table 2 pharmaceuticals-15-01093-t002:** Table summarizing the references dealing with silica NPs@RB, including the type of NPs, as well as their size, excitation wavelength, type of detection of ^1^O_2_, type of coupling between the NPs and the RB, and the type of biological experiments performed.

PDT Silica NPs									
References	Type of NPs	Size of NPs (nm)	Irradiation Conditions	Type of ^1^O_2_ Detection	Type of Coupling between NPs and RB	Biological Tests/In Vitro		Biological Test/In Vivo	
						Cells Used	Results	Mice Used	Results
[63]	Ln:ZrO_2_@SiO_2_@RB	20–25 thick	633 nm, 4 mW He-Ne laser, 10 min	DPBF	Covalent	HeLa	Cytotoxicity (+)	nd	nd
[50]	Mesoporous silica NPs@RB	160–180	green light, 5 min	nd	Covalent	SK-MEL-28	Cytotoxicity (+)	nd	nd
[59]	Silica coated (Gd_3_Al_5_O_12_:Ce^3+^)-RB	74.13 ± 9.04	470 nm, 20 mW/cm^2^, 15–30–45 min	DPBF	Loaded	MDA-MB-231	Cytotoxicity (+)	nd	nd
[58]	MSN@DOX@RB	200	532 nm, 0.5 W/cm^2^, 5 min	nd	Covalent	MCF-7	Cytotoxicity (+)	nd	nd
[57]	MSN@C-dots/RB	104	540 nm, 300 mW/cm^2^, 5 min	DPBF	Encapsulation	H1299	Cytotoxicity (+)	nd	nd
[54]	UCN/SiO_2_-RB DOX)@PPADT	38.2 ± 3.6	980 nm, 1.0 W/cm^2^, 20 min	ABDA	Covalent	HeLa	Cytotoxicity (+)	nd	nd
[55]	SiNP-AAP-RB	5–15	410 nm, 1.6 J/cm^2^, 15 min	Direct	Covalent	A549	Cytotoxicity (+)	nd	nd
[52]	Silica capped rMSN-ts	Lenght 33 ± 27 and width 103 ± 15	5 mV green laser, 30 s	Direct	Covalent	MCF7 L269	Cytotoxicity (+)	nd	nd
[53]	FITC-PGSN-RB	100	480 nm, 100 W xenon lamp, 10 min	ABDA	Adsorption	HeLa	Cytoxicity (+)	nd	nd
[51]	RB-MSNs NPs	150–180	nd	Direct	Encapsulation	nd	nd	nd	nd
[62]	RB-DOX@HMSNs	170	532 nm, 10 mW/cm^2^, 5 min	DPBF	Encapsulation	4T1	Cytotoxicity (+)	nd	nd
[61]	RB@ZIF-8@SiO_2_@RB NPs (drug loaded MOFs@SiO_2_ NPs)	93.8	532 nm, 200 mW/cm^2^, 2 min or 100 mW/cm^2^, 7 min	DPBF	Encapsulation	4T1	Cytotoxicity (+)	BALB/c mice bearing AT1 tumors	PDT (+)
[56]	RB–MMSNs	190	535 nm, 25 mW/cm^2^, 3 min	Uric acid	Encapsultation	B16 HeLa L929	Cytotoxicity (+)	C57BL/6J mice	PDT (+)
[64]	RB-PEG5000-MSNs	50	518 nm, 10 J/cm^2^, 5 min	Direct	Covalent	HeLa	Cytotoxicity (+)	nd	nd
[49]	RB-SiNP	electrostatic = 35 ± 5 covalent = 50 ± 5	480 nm, 1 h (fluence not cited)	SOSG	Covalent/Electrotatic	MCF-7 4451	Cytoxicity (+)	nd	nd

nd: not determined; cytotoxicity (+): induced tumor cell death in vitro; PDT (+): induced tumor death in vivo.

**Table 3 pharmaceuticals-15-01093-t003:** References dealing with organic NPs@RB, including the type of NPs, as well as their size, excitation wavelength, type of detection of ^1^O_2_, the type of coupling between the NPs and the RB, and the type of biological experiments performed.

PDT Organic NPs									
References	Type of NPs	Size of NPs (nm)	Irradiation Conditions	Type of ^1^O_2_ Detection	Type of Coupling between NPs and RB	Biological Tests/In Vitro		Biological Test/In Vivo	
						Cells Used	Results	Mice Used	Results
[72]	UCNP/PAH/HA-RB	30.34	980 nm, 1.5 mW/cm^2^, 30 min	nd	Covalent	NIH3T3	Cytotoxicity (+)	nd	nd
[78]	POEGMA-b-P(HPMA-co-RBMA)	52	590–595 nm, 0.2 mW/cm^2^, 15 min	ADPA	Covalent	HCT116 K562	Cytotoxicity (+)	nd	nd
[70]	RB-BTSA	109 ± 5	490 nm, 30–135 J/cm^2^, 5–30 min	TMP–OH	Encapsulation	HeLa	Cytotoxicity (+)	nd	nd
[73]	PLGA-NP-RB	28	520 nm, 200 mW/cm^2^, 30 min	nd	Encapsulation	MCF-7 HeLa	Cytotoxicity (+)	H22 tumor-bearing mice	PDT (+)
[71]	PVK@C153@RB	340	No biological test	2-chlorophenol	Encapsulation	nd	nd	nd	nd
[75]	NaYF_4_:Er^3+^,Yb^3+^ NPs@SPAN-20 micelles@RB@ TOPO@Cremophor@PLGA copolymer	150	980 nm, 10 J/cm^2^, 6.2 W/cm^2^, 5 min	nd	Encapsulation	Me-45 MeWo HaCaT	Cytotoxicity (+)	Melanoma cells innoculated on monoclonal mouse	PDT (+)
[74]	HBNCs@RB	220	632 nm, 15 mW/cm^2^, 60 min	DCFH-DA,	Encapsulation	Tramp-C1	Cytotoxicity (+)	nd	nd
[69]	PAPAM dendrimers G 2.5-RB	20	150 W xenon lamp 540 nm, 10 min	Iodure reaction	Covalent	DLA	Cytotoxicity (+)	nd	nd
[77]	polypeptide C18GR7RGDS	17.28	808 nm, 1.5 W/cm^2^, 3 min	DPBF	Encapsulation	B16 HeLa	Cytotoxicity (+)	nd	nd
[76]	BP-CDPNP-RB	280	Two photons (810 nm, 245 mW/cm^2^, 20 min	ABDA	Encapsulation	MCF-7	Cytotoxicity (+)	nd	nd
[77]	PARN@RB	17.28	808 nm, 1.5 W/cm^2^, 3 min	DPBF DCF-DA	Encapsulation	B16	Cytotoxicity (+)	HeLa tumor-bearing mice	PDT (+)

nd: not determined; cytotoxicity (+): induced tumor cell death in vitro; PDT (+): induced tumor death in vivo.

**Table 4 pharmaceuticals-15-01093-t004:** References dealing with gold NPs@RB, including the type of NPs, along with their size, excitation wavelength, type of detection of ^1^O_2_, the type of coupling between the NPs and the RB, and the type of biological experiments performed.

PDT Gold NPs									
References	Type of NPs	Size of NPs (nm)	Irradiation Conditions	Type of ^1^O_2_ Detection	Type of Coupling between NPs and RB	Biological Tests/In Vitro		Biological Test/In Vivo	
						Cells Used	Results	Mice Used	Results
[84]	AuNPs-RB	AuNPs-RB complexes: Electrostatic (53) Covalent (49)	500 nm, 5 J/cm^2^	nd	Covalent	Vero HeLa	Affinity (+)	nd	nd
[85]	Gold NPs@RB@DOX	124.6	532 nm, 14.3 mW/cm^2^, 6 min	SOSG	Encapsulation	HCT116 MCF-7 CCD 841 CoN	Cytotoxicity (+)	nd	nd
[86]	CTO-RB-AuNR	0.273	980 nm, 1.5 W/cm^2^, 6 min	DPBF	Covalent	Hep G2	Cytotoxicity (+)	nd	nd
[83]	AuNRs@RB	13 ± 2	PDT: 532 nm, 1.76 W/cm^2^, 10 min PTT: 810 nm NIR light, 8.16 W/cm^2^, 5 min	ABDA	Covalent	Cal-27	Cytotoxicity (+)	Male Syrian Golden hamsters with oral carcinomas	PDT (+)

nd: not determined; affinity (+): induced the affinity to tumor cells; cytotoxicity (+): induced tumor cell death in vitro; PDT (+): induced tumor death in vivo.

**Table 5 pharmaceuticals-15-01093-t005:** References dealing with quantum dots NPs@RBs, including the type of NPs, along with their size, excitation wavelength, type of detection of ^1^O_2_, the type of coupling between the NPs and the RB, and the type of biological tests performed.

PDT Polymer Dots NPs									
References	Type of NPs	Size of NPs (nm)	Irradiation Conditions	Type of ^1^O_2_ Detection	Type of Coupling between NPs and RB	Biological Tests/In Vitro		Biological Test/In Vivo	
						Cells Used	Results	Mice Used	Results
[89]	Pdots-RB with: PEG350-PE, PEG2000-PE, PEG5000-PE	PEG350-PE: 43 ± 9 PEG2000-PE: 77 ± 0.3 PEG5000-PE: 60 ± 0.15	473 nm, 0.15 mJ/cm^2^, between 1 h and 4 h	DMA	Encapsulated in the coating of Pdots	MCF-7	Cytotoxicity (+)	nd	nd
[90]	CQDS-RB	CQDs-RB: 33.1 ± 8.7 CQDs: 2.1 ± 0.3	532 nm, 20 mW/cm^2^, 30 min	SOSG	Covalent	MCF-7	Cytotoxicity (+)	nd	nd

nd: not determined; cytotoxicity (+): induced tumor cell death in vitro; PDT (+): induced tumor death in vivo.

**Table 6 pharmaceuticals-15-01093-t006:** References dealing with nanogel@RBs, nanogels, nanohybrid NPs, pH-sensitive NPs, magnetic NPs, nanocomplexes, nanocapsules, including type of NPs, along with their size, excitation wavelength, type of detection of ^1^O_2_, the type of coupling between the NPs and the RB, and the type of biological tests performed.

PDT Nanogels, Nanohybrid NPs, pH-Sensitive NPs, Magnetic NPs, Nanocomplexes, Nanocapsules										
References	Type of NPs		Size of NPs (nm)	Irradiation Conditions	Type of ^1^O_2_ Detection	Type of Coupling NPs-RB	Biological Tests/In Vitro		Biological Test/In Vivo	
							Cells Used	Results	Mice Used	Results
[92]	Ca^2+^/Gd^3+^-RB nanocapsules/RGD		300	550 nm, 100 mW/cm^2^, 15 min	DPBF	Encapsulation	HepG2	Cytotoxicity (+)	nd	nd
[94]	BP-RB Bis(pyrene)-RB		70	100 W xenon lamp (810 nm, 20 min)	ABDA	Encapsulation	MCF-7	Cytotoxicity (+)	NIH mice (female)	PDT (+)
[97]	(MNCs)-RB		108.6	632 nm, 15 mW/cm^2^ or 532 nm, 100 mW/cm^2^ for 5 min	DCFH-DA	Electrostatic	MCF-7 Tramp-C1 SKOV-3	Cytotoxicity (+)	BALB/c nude mice bearing MCF-7/MDR	PDT (+)
[99]	Cs-Na/RB, Cs-Na-RBD	Cs-Na/RB: 50 Cs-Na/RBD: 60		518 nm, 4.0 W/cm^2^, 10 min	DCFH-DA	Encapsulation	PC9	Cytotoxicity (+)	nd	nd
[101]	UCNPs@MF-RB/PEG		75	980 nm, 0.5 mW/cm^2^, 15 min	DPBF	Covalent	HeLa	Cytotoxicity (+)	nd	nd
[103]	RB loaded in nanogel		218	11 W LEDs (400–700 nm), 2 min	ABDA	Encapsulation	HT-29	Cytotoxicity (+)	nd	nd

nd: not determined; cytotoxicity (+): induced tumor cell death in vitro; PDT (+): induced tumor death in vivo.

**Table 7 pharmaceuticals-15-01093-t007:** References dealing with silica NPs@RB under X-rays, including the type of NPs, along with their size, excitation wavelength, type of detection of ^1^O_2_, the type of coupling between the NPs and the RB, and the type of biological tests performed.

X-PDT Silica NPs									
References	Tpe of NPs	Size of NPs (nm)	Irradiation Conditions	Type of ^1^O_2_ Detection	Type of Coupling NPs-RB	Biological Tests/In Vitro		Biological Test/In Vivo	
						Cells Used	Results	Mice Used	Results
[104]	LaF_3_:Tb@SiO_2_-RB (ScNP-RB)	45	No biological test	DPBF	Covalent	nd	nd	nd	nd
[107]	CeF_3_:Tb^3+^, Gd^3+^ coated with MSN, loading with RB	89.2	160 kV (1 Gy/min)	Iodure reaction	Covalent	4T1 Renca Mgc893	Toxicity (+)	BALB/c, 4T1	PDT (+)
[105]	CSS (NaLuF_4_:Gd,Eu@NaLuF_4_:Gd@NaLuF_4_:Gd,Tb)- CSS@MSiO_2_@PAH-RB	27.5	160 kV, 25 mA, 1, 3, 5 Gy	nd	Adsorption	MCF-7/MDA-MB-231	Cytotoxicity (+)	nd	nd
[60]	Gd_2.98_Ce_0.02_Al_5_O_12_ NPs (GAG) coated with mSiO_2_ and loaded with RB, GAG@mSiO_2_@RB	147	Blue light (470 nm) 20 mW/cm^2^ (55 kV)	nd	Adsorption	MDA-MB-231	Cytotoxicity (+)	nd	nd
[106]	RB-ZSM-RGD	80.8 ± 3.5	50 kV, 1.0 Gy, 10 min	SOSG	Covalent	U87MG	Cytotoxicity (+)	Nude mice U98 MG	PDT (+)

nd: not determined; cytotoxicity (+): induced tumor cell death in vitro; PDT (+): induced tumor death in vivo.

**Table 8 pharmaceuticals-15-01093-t008:** References dealing with polymer NPs@RB under X-rays, including the type of NPs, along with their size, excitation wavelength, type of detection of ^1^O_2_, the type of coupling between the NPs and the RB, and the type of biological tests performed.

X-PDT Polymer NPs									
References	Type of NPs	NPs Size (nm)	Irradiation Conditions	Type of ^1^O_2_ Detection	Type of Coupling NPs-RB	Biological Tests/In Vitro		Biological Test/In Vivo	
						Cells	Results	Mice Used	Results
[110]	Gadolinium (Gd)–RB coordination polymer nanodots (GRDs)	3.3 ± 0.8	In vitro: 532 nm, 30 mW/cm^2^, 5 min; 1 Gy In vivo: 532 nm, 140 mW/cm^2^, 15 min; 1 Gy	SOSG	Coordination	4T1	Cytotoxicity (+)	BALB/c mice, 4T1-tumor-bearing mice	PDT (+)
[109]	POMo@CS–RB	80	2 Gy/min (160 kV)	SOSG	Adsorption	4T1	Cytotoxicity (+)	BALB/c mice, 4T1-tumor-bearing mice	PDT (+)
[108]	Ln NPs-PEG-RB	4	2 Gy (150 kV)	nd	Covalent	nd	nd	B16 murine mela noma cells	PDT (+)

nd: not determined; cytotoxicity (+): induced tumor cell death in vitro; PDT (+): induced tumor death in vivo.

**Table 9 pharmaceuticals-15-01093-t009:** References dealing with nanocomposites@RB under X-ray, including the type of NPs, along with their size, excitation wavelength, type of detection of ^1^O_2_, type of coupling between the NPs and the RB, and the types of biological tests performed.

X-PDT Nanocomposites									
References	Type of NPs	Size of NPs (nm)	Irradiation Conditions	Type of ^1^O_2_ Detection	Type of Coupling NPs-RB	Biological Tests/In Vitro		Biological Test/In Vivo	
						Cells Used	Results	Mice Used	Results
[112]	β-NaGdF_4_:Tb^3+^-RB	25.6	80 kV, 0.5 mA (0,05–2 Gy), 20 min	DPBF	Covalent	HepG2	Cytotoxicity (+)	Nude Mice with subcutaneous HepG2 xenografts	PDT (+)
[111]	β-NaGdF_4_:Tb^3+^-RB	10	80 kV, 0.5 mA and 45 cm distance (1.17 Gy/h)	DPBF	Covalent	HepG2	Cytotoxicity (+)	Female BALB/c nude mice	PDT (+)
[113]	PEG-capped GdF_3_:Tb^3+^@RB nanocomposites	Length: 250 Width: 60	136 mGy/4 min	nd	Adsorption	nd	nd	BALB/c (ca. 3 months age, weight 34–35 g) male mice	PDT (+)

nd: not determined; cytotoxicity (+): induced tumor cell death in vitro; PDT (+): induced tumor death in vivo.

**Table 10 pharmaceuticals-15-01093-t010:** References dealing with lanthanide@MOF, nanoprobes@RB, nanophosphors@RB and ScNPs@RB under X-ray, including the type of NPs, along with their size, excitation wavelength, type of detection of ^1^O_2_, type of coupling between the NPs and the RB, and the type of biological tests performed.

X-PDT Lanthanide@MOF, Nanoprobes, Nanophosphors, ScNPs@RB									
Reference	Type of NPs	Size of NPs (nm)	Irradiation Conditions	Type of ^1^O_2_ Detection	Type of Coupling NP-RB	Biological Tests/In Vitro		Biological Test/In Vivo	
						Cells Used	Results	Mice Used	Results
[118]	SNPs@Zr-MOF@RB- SNPs	30	45 kV (0, 3, 6, 9, 12, 15 min).	DPBF	Covalent	4T1	Cytotoxicity (+)	4T1 tumor-bearing BALB/c mice)	PDT (+)
[114]	ScNPs: LaF_3_:Tb-RB	38.9	75 kV, 2.0 mA	DPBF	Encapsulation	nd	nd	nd	nd
[117]	NaGdF_4_@NaGdF_4_:Tb@NaYF_4_ (CSS)@RB@cRGD	18.8	4 Gy/min	DPBF	Covalent	U87MG HepG2 NIH-3T3	Cytotocity (+)	nd	nd

nd: not determined; cytotoxicity (+): induced tumor cell death in vitro; PDT (+): induced tumor death in vivo.

## Data Availability

Data sharing not applicable.

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
