# Peer review of "Importance of Rose Bengal Loaded with Nanoparticles for Anti-Cancer Photodynamic Therapy"

_pharmaceuticals, 2022, doi:10.3390/ph15091093_

Round 1

Reviewer 1 Report (Previous Reviewer 3)

I'm concerned about the last version of the your reply (pharmaceuticals-1792356, report 2), and the author's reply is very impolite. No matter my point of view or your point of view. As a review, you should have your own induction and collation, and give selective examples to highlight the key points, instead of piling up long and unfocused speeches. At present, I think there are still some areas to be improved.

1. The clarity of Figures 6,16, .... needs to be improved. 2. Many abbreviations don't seem to be reflected in the abbreviation table, such as nd, CTAB, etc. 3. The abbreviation of CTAB in the manuscript is used to represent two phrases. Please delete one of them to avoid misunderstanding.

4. The whole manuscript needs to be checked carefully. It is the third draft at present, but many problems can still be found.

Author Response

The authors have answered most of my questions, and I am happy about that. However, I still have some suggestions.

  1. I still do not think RB coupled to NPs is not an excellent approach (line 1590). So I suggest not using the extreme word “excellent”. The authors could use good instead of excellent.

Thank you for this suggestion. We changed “excellent” by “good”.

  1. There is a typo in line 1600. Please correct it.

Thank you. We changed “ if” by “is”

  1. The authors mentioned that all types of NPs presented in this review were very efficient (line 1601). Among them, which one is the best? Can authors comment on it? 

It is not so easy to answer this question because all NPs present some advantages. Anyway, we added a sentence

« To our point of view, UCPN present many advantages since their size can be small enough to allow passive targeting, can be easily fonctionalized by a vector to allow active targeting, and can be excited both by NIR or X-rays ».

Reviewer 2 Report (Previous Reviewer 2)

The authors have answered most of my questions, and I am happy about that. However, I still have some suggestions.

1.       I still do not think RB coupled to NPs is not an excellent approach (line 1590). So I suggest not using the extreme word “excellent”. The authors could use good instead of excellent.

2.       There is a typo in line 1600. Please correct it.

3.       The authors mentioned that all types of NPs presented in this review were very efficient (line 1601). Among them, which one is the best? Can authors comment on it? 

Author Response

Reviewer 2

I'm concerned about the last version of the your reply (pharmaceuticals-1792356, report 2), and the author's reply is very impolite. No matter my point of view or your point of view. As a review, you should have your own induction and collation, and give selective examples to highlight the key points, instead of piling up long and unfocused speeches. At present, I think there are still some areas to be improved.

  1. The clarity of Figures 6,16, .... needs to be improved.

We improved the clarity of figures 6, 16 and some others.

  1. Many abbreviations don't seem to be reflected in the abbreviation table, such as nd, CTAB, etc.

Thank you. We added abbreviations in the table and in the text

ABS: Acetate Buffer Solution

ATF: Amino Terminal Fragment of urokinase plasminogen activator

BFNS: bifunctional nanospheres

BP: bis-pyrene

bPEI: branched polyethylenimine

BRET: Bioluminescence resonance energy transfer

cDDP: cis-diaminedichloroplatine(II)

CQDs: carbon quantum dots

C18RB: Hydrocarbonized RB

CTAB: Cetyltrimethylammoniumbromide

DMA: 9,10-dimethylanthracene

DNA: Deoxyribonucleic acid

EADC:1-Ethyl-3-(3-dimethylaminopropyl)carbodiimide

GSH: Glutathione

HAx: Hexamethylenediamine

HSPC: Hydrogenated soy L-a-phosphatidylcholine

IR: Infrared

MMSN: Magnetic mesoporous silica NP

MNCs: Magnetic nanoclusters

MRI: Magnetic Resonance Imaging

MTT: 3-[4,5-dimethylthiazol-2-yl]-2,5 diphenyl tetrazolium bromide

Nd: Not determined

1O2: Singlet oxygen

PAMAM: Poly(amidoamine)

PBS: Phosphate Buffer Solution

PCL: PEG-block-poly(ε-caprolactone)

Pdots: Polymer dots

PE: phosphoethanolamine

PEG-b-PAsp: Polyethylene glycol-b-poly(aspartic acid)

PE-NH2: 1,2-dioleoyl-sn-glycero-phosphoethanolamine-N(hexanoylamine)

PGA: Poly(glycolic acid)

PHLA: Partially hydrolyzed α-alactalbumin

PPADT: poly-(1,4-phenyleneacetone dimethylenethioketal)

PVA: Poly(vinyl alcohol)

RET: Resonance energy transfer

ScNPs: Scintillating NPs

SHG: Second harmonic generation

SSTRs: Somatostatin receptors

TMP–OH: 2,2,6,6-tetramethyl-4-piperidinol

TUNEL: Terminal deoxynucleotidyl transferase dUTP nick end labeling

UPA: Urokinase Plasminogen Activator

UV: Ultra violet

ZIF-90: Zeolitic imidazolate framework-90

ZnPc-COOH: Zinc β-carboxyphthalocyanine

  1. The abbreviation of CTAB in the manuscript is used to represent two phrases. Please delete one of them to avoid misunderstanding

Thank you. We remove the second sentence.

Adem et al. [52] proposed MSNs namely based on silica capped surfactant (CTAB) and RB (rMSN-ts) 

  1. The whole manuscript needs to be checked carefully. It is the third draft at present, but many problems can still be found.

We checked the manuscript carefully.

This manuscript is a resubmission of an earlier submission. The following is a list of the peer review reports and author responses from that submission.

Round 1

Reviewer 1 Report

This is an extremely interesting and up-to-date overview of the latest advances in photodynamic therapy. Recent advances and trends in the development of such therapy are discussed. Of particular interest is the development of a new version of photodynamic therapy using X-rays to excite a photosensitizer. The main limitation of classical photodynamic therapy is the insignificant depth of penetration of the laser beam into biological tissue. This does not allow the therapy of deeply localized tumors. The use of X-rays irradiation together with target vectors will significantly expand the capabilities of the method.

The review will be of interest to a wide range of readers, including young scientists starting their careers. The manuscript may be accepted for publication.

Author Response

Reviewer 1

This is an extremely interesting and up-to-date overview of the latest advances in photodynamic therapy. Recent advances and trends in the development of such therapy are discussed. Of particular interest is the development of a new version of photodynamic therapy using X-rays to excite a photosensitizer. The main limitation of classical photodynamic therapy is the insignificant depth of penetration of the laser beam into biological tissue. This does not allow the therapy of deeply localized tumors. The use of X-rays irradiation together with target vectors will significantly expand the capabilities of the method.

The review will be of interest to a wide range of readers, including young scientists starting their careers. The manuscript may be accepted for publication.

Thank you very much for these nice comments

Reviewer 2 Report

In this manuscript, Dhaini et al., wrote a review paper on Rose Bengal loaded NPs for photodynamic therapy. In this review, the authors summarized the history of the use of RB-loaded NPs in the context of PDT. The authors concluded that RB-loaded NPs are an excellent photosensitizer as compared to RB alone. The authors have done an extensive study of the literature related to RB-loaded NPs. However, the authors have not given their views/perspectives.  The manuscript can be accepted after the following revisions. The suggestions for the authors are as follows:

Minor comments:

1. Please use consistent capitalization (do not mix capital and small letters) on the title of the manuscript.

2. It is suggested to include the authors’ department.

3. I think it would be better to write the caption of Figure 1 in sequential order.

4. In Figure caption 37, it should be Figure 38. Please correct it. In addition to that, Figure 38 should be Figure 39.

5. Figure 50 should be Figure 49 (Figure 49 is not there).

6. The authors mentioned ‘in the dark’ on line 28. Could the authors make it clear?

7. I think it would be better to mention the last name of the first author. For example, it is good to write Wang et al. [20] rather than Sheng Wang et al. [20] (line 108). I am just giving one example. The authors used the first and last names of the first author throughout the manuscript. Also, I think month is not required. For example,   Novembre 2014, Sheng Wang et al. [20] (line 108). The authors have included month throughout the manuscript when referring to the literature.

8. There are minor typos. The authors should be consistent. For example, there is no space between 70 and %, but there are spaces between 10 and % and 2 and % (line 199). Please be consistent. Also, please double-check on line 200.

9.  Period is missing (line 1363).

Major comments:

11. The main issue of this review paper is the lack of authors’ views. After studying this manuscript, it seems that RB-loaded NPs are the best candidate, which is not true. Why don’t the authors mention some limitations of the RB-loaded NPs in the context of PDT? The authors should also mention how those limitations could be solved (at least the authors can give their views). The authors should give their honest perspectives to the best of their knowledge. 

12. The review article is unnecessarily lengthy. Can the authors make it a little bit concise?

Author Response

Reviewer 2

In this manuscript, Dhaini et al., wrote a review paper on Rose Bengal loaded NPs for photodynamic therapy. In this review, the authors summarized the history of the use of RB-loaded NPs in the context of PDT. The authors concluded that RB-loaded NPs are an excellent photosensitizer as compared to RB alone. The authors have done an extensive study of the literature related to RB-loaded NPs. However, the authors have not given their views/perspectives.  The manuscript can be accepted after the following revisions. The suggestions for the authors are as follows:

Minor comments:

  1. Please use consistent capitalization (do not mix capital and small letters) on the title of the manuscript.

Thank you for this comment. It has been corrected

  1. It is suggested to include the authors’ department.

Thank you for this comment. Authors’ department have been added

  1. I think it would be better to write the caption of Figure 1 in sequential order.

Thank you for this comment. It has been corrected

  1. In Figure caption 37, it should be Figure 38. Please correct it. In addition to that, Figure 38 should be Figure 39.

Thank you for this comment. It has been corrected

  1. Figure 50 should be Figure 49 (Figure 49 is not there).

Thank you for this comment. It has been corrected

  1. The authors mentioned ‘in the dark’ on line 28. Could the authors make it clear?

Thank you for this comment. We change te sentence “The PS is not toxic in the dark but induces the production of highly reactive oxygen species (ROS) such as singlet oxygen upon light illumination »

  1. I think it would be better to mention the last name of the first author. For example, it is good to write Wang et al. [20] rather than Sheng Wang et al. [20] (line 108). I am just giving one example. The authors used the first and last names of the first author throughout the manuscript. Also, I think month is not required. For example,   Novembre 2014, Sheng Wang et al. [20] (line 108). The authors have included month throughout the manuscript when referring to the literature.

Thank you for this comment. We removed the first name, the month and the date everywhere in the text

  1. There are minor typos. The authors should be consistent. For example, there is no space between 70 and %, but there are spaces between 10 and % and 2 and % (line 199). Please be consistent. Also, please double-check on line 200 as tu fais ? Peux tu surligner en jaune ????.

Thank you for this comment. It has been corrected

Line 199 has been changed “Cell viability decreased to 70%, 30% and 5% with increasing concentration of A2780cisR (respectively 16, 80 and 400 µg/mL)”.

  1.  Period is missing (line 1363).

Thank you for this comment. It has been corrected

Major comments:

  1. The main issue of this review paper is the lack of authors’ views. After studying this manuscript, it seems that RB-loaded NPs are the best candidate, which is not true. Why don’t the authors mention some limitations of the RB-loaded NPs in the context of PDT? The authors should also mention how those limitations could be solved (at least the authors can give their views). The authors should give their honest perspectives to the best of their knowledge. 

Thank you for this comment. We added our point of view in the conclusion.

“RB coupled to NPs is an excellent approach for PDT applications; Indeed, RB is xanthene dye that has interesting photo- and sono-sensitive properties. RB is already used for clinical applications and cancer formulation RB known as PV-10 is currently undergoing clinical trials for different types of cancer (melanoma, breast cancer) or infection (clinicaltrials.gov). RB can be used alone but can also be a photosensitizer for PDT applications. RB present several advantages such as the ability to produce 1O2 upon light illumination and solubility on water. One disadvantage could be its absorption spectra with an absorption maximum wavelength in water of 550 nm which does not allow a great penetration of light into the tissue. Moreover, RB is not selective of cancer cells.

To improve the system, the use of NPs if an interesting strategy. Indeed, all types of NPs presented in this review were very efficient (UCNPs, Silica NPs, Organic NPs, Gold NPs, Pdots NPs, Nanocapsule, Nanocomplex, Magnetic NPs, pH sensitive NPs, Hybrid NPs, Nanogel, Nanocomposite, Nanophosphore, Lanthanide@MOF Nanoprobe). The use of NPs allows to target RB to the tumor thanks to the passive targeting due to EPR effect. It can be notice regarding all the papers that the efficiency of RB is higher when RB is coupled to NP and the best system is when RB is covalently coupled to NP better than encapsulated.

Another advantages of RB is that its absorption spectra matches with emission of lanthanide such as terbium. It is then possible to excite terbium by X-ray and after energy tranfer to RB to induce the formation of 1O2. The design of NP coupled to RB and targeted unit to target over-expressed receptors could be a nice option to treat deep tumor or melanoma for example.”

  1. The review article is unnecessarily lengthy. Can the authors make it a little bit concise?

Thank you for this comment. We removed some sentences.

Reviewer 3 Report

This manuscript reviews the recent progress in Rose Bengal loaded with nanoparticles for PDT. The following issues should be addressed to improve this work.

1. In Introduction section, there is a lack of introduction of PDT mechanism and other background. Including, but not limited to, the following documents (Coordination Chemistry Reviews 426 (2021) 213548; Coordination Chemistry Reviews 447 (2021) 214155; Biomater. Sci., 2021, 9, 7811.), and suggestions can be used as references for ideas or background knowledge during revision.

2. Many examples were listed, but lack of sufficient discussion and overview. There is a lack of mutual connection and logical relationship between each example.

3. A summary figure should be supplemented for the section of "conclusion and outlook", which would make this manuscript more attractive.

4. The list of some key information in the table is not clear. The key link of PDT is the characterization of biological activity, but the table seems to lack key information such as dosage, administration mode, cell mortality and tumor inhibition rate.

5. The figures provided in the manuscript are not representative, they are all displays of biological activity, and can't show the particularity of each example. According to the differences of different examples, it is suggested to increase the display of diagrams including preparation method, mechanism, spectrum, in vivo imaging, and so on. In addition, some figures are lack of clarity and unreasonable layout.

Author Response

Reviewer 3

This manuscript reviews the recent progress in Rose Bengal loaded with nanoparticles for PDT. The following issues should be addressed to improve this work.

Thank you for all your comments.

In Introduction section, there is a lack of introduction of PDT mechanism and other background. Including, but not limited to, the following documents (Coordination Chemistry Reviews 426 (2021) 213548 Coordination Chemistry Reviews 447 (2021) 214155; Biomater. Sci., 2021, 9, 7811. and suggestions can be used as references for ideas or background knowledge during revision.

Thank you for your suggestion.

Coordination Chemistry Reviews 426 (2021) 213548 is a review concerning the use of phtalocyanines for photothermal therapy and not PDT.

Coordination Chemistry Reviews 447 (2021) 214155 deals with the Recent advances in supramolecular activatable phthalocyanine-based photosensitizers for anti-cancer therapy. We added this reference

Biomater. Sci., 2021, 9, 7811 is a review concerning Phthalocyanine-based photoacoustic contrast agents for imaging and theranostics, not PDT

We agree we could add the different types of photosensitizers and we added some informations in the introduction. Since it is a special issue PDT and the review is already quite big , we did not add too many details.

“This PS is not toxic in the dark but induces the production of highly reactive oxygen species (ROS) such as singlet oxygen (1O2) upon light illumination [5]. Upon light illumination, the PS is activated to a singlet excited state and then to a triplet state after intersystem crossing. In its triplet state, the PS can undergo an electron or proton transfer to produce superoxide anion and hydroxyl radical, or transfer its energy to oxygen to produce 1O2. Around 20 PS have been commercialized nowadays or are in clinical trials, such as porphyrin, chlorin and phhalocyanine (Coordination Chemistry Reviews 447 (2021) 214155 et Chem Soc Rev 2021 50 4185) . There is still some improvements to be done such as a better absorption in the near IR, a better solubility and increased selectivity.

  1. Many examples were listed, but lack of sufficient discussion and overview. There is a lack of mutual connection and logical relationship between each example.

Thank you for your remark. In this review, we did not choose some explae of papers describing the used of RB and NP but we collected all the papers of the literature dealing wit RB and nanoparticles. Our choice was to classify them by type of illumination (PDT and PDTX, then by type of nanoparticles (upconversion, silica, gold, …). For each type of NH, we made the choice to classify them by chrolonogic order.

We added our strategy in the text

“In this review, we collected all the papers describing the use of RB and nanoparticles. They are divided in two parts: NP excited by light (PDT) and NP excited by W-rays (PDTW). Then we classified the papers depending on the type of nanoparticles used. Finally, for each type of nanoparticles, the articles are classified in chronologic order. »

We also modified our conclusion to give our point of view

“RB coupled to NPs is an excellent approach for PDT applications; Indeed, RB is xanthene dye that has interesting photo- and sono-sensitive properties. RB is already used for clinical applications and cancer formulation RB known as PV-10 is currently undergoing clinical trials for different types of cancer (melanoma, breast cancer) or infection (clinicaltrials.gov). RB can be used alone but can also be a photosensitizer for PDT applications. RB present several advantages such as the ability to produce 1O2 upon light illumination and solubility on water. One disadvantage could be its absorption spectra with an absorption maximum wavelength in water of 550 nm which does not allow a great penetration of light into the tissue. Moreover, RB is not selective of cancer cells.

To improve the system, the use of NPs if an interesting strategy. Indeed, all types of NPs presented in this review were very efficient (UCNPs, Silica NPs, Organic NPs, Gold NPs, Pdots NPs, Nanocapsule, Nanocomplex, Magnetic NPs, pH sensitive NPs, Hybrid NPs, Nanogel, Nanocomposite, Nanophosphore, Lanthanide@MOF Nanoprobe). The use of NPs allows to target RB to the tumor thanks to the passive targeting due to EPR effect. It can be notice regarding all the papers that the efficiency of RB is higher when RB is coupled to NP and the best system is when RB is covalently couped to NP better than encapsulated.

Another advantages of RB is that its absorption spectra matches with emission of lanthanide such as terbium. It is then possible to excite terbium by X-ray and after energy tranfer to RB to induce the formation of 1O2. The design of NP coupled to RB and targeted unit to target over-expressed receptors could be a nice option to treat deep tumor or melanoma for example.”

  1. A summary figure should be supplemented for the section of "conclusion and outlook", which would make this manuscript more attractive.

Thank you for this suggestion. We will add a summary figure.

  1. The list of some key information in the table is not clear. The key link of PDT is the characterization of biological activity, but the table seems to lack key information such as dosage, administration mode, cell mortality and tumor inhibition rate

Thank you for your suggestion

The tables are already very full and it is not possible for us to add the information. The review is already quite big.

  1. The figures provided in the manuscript are not representative, they are all displays of biological activity, and can't show the particularity of each example. According to the differences of different examples, it is suggested to increase the display of diagrams including preparation method, mechanism, spectrum, in vivo imaging, and so on. In addition, some figures are lack of clarity and unreasonable layout.

Thank you for your suggestion

For the same reasons, we can not add diagrams including preparation method, mechanism, spectrum, in vivo imaging, and so on since the review is already quite big.

We hope that if some researches are interested in synthesizing nanoparticle coupled to RB, this exhaustive review will help them to make their choice and they can find all the details in the publications.

Round 2

Reviewer 3 Report

At present, the manuscript has not completely solved my question. The current review is too long because there are too many inserted figures and most of them are not representative. It is suggested that the corresponding figures should be selected according to the characteristics of different examples, and the selected figures should be representative, including but not limited to preparation methods, spectral data, activity characterization, etc. Instead of randomly stacking pictures together, it doesn't make much sense.

Author Response

Dear "reviewer 3",

Reviewer 1 accepts our publication .
Reviewer 2 accepts our publication after some major modifications
You accept all our modifications but still ask us to remove figures, focus on few publications and add details.
It is of course your point of view but it is not our point of view. Our strategy was to give all the papers of the literature and not to focus on few papers. That's the way we see a review.

I think the editor will decide

Best regards

Céline Frochot